# Combating Bilateral Edge Noise
# for Robust Link Prediction

**Zhanke Zhou**[1]    **Jiangchao Yao**[2,3†]    **Jiaxu Liu**[4]    **Xiawei Guo**[4]    **Quanming Yao**[5]
**Li He**[4]    **Liang Wang**[4]    **Bo Zheng**[4]    **Bo Han**[1†]

[1]Hong Kong Baptist University    [2]CMIC, Shanghai Jiao Tong University
[3]Shanghai AI Laboratory    [4] Alibaba Group    [5] Tsinghua Unversity

{cszkzhou, bhanml}@comp.hkbu.edu.hk    sunarker@sjtu.edu.cn    qyaoaa@tsinghua.edu.cn
{liujiaxu.ljx, guoxiawei, heli, wangliang}@taobao.com    bozheng@alibaba-inc.com

## Abstract

Although link prediction on graphs has achieved great success with the development of graph neural networks (GNNs), the potential robustness under the edge noise is still less investigated. To close this gap, we first conduct an empirical study to disclose that the edge noise bilaterally perturbs both input *topology* and target *label*, yielding severe performance degradation and representation collapse. To address this dilemma, we propose an information-theory-guided principle, Robust Graph Information Bottleneck (RGIB), to extract reliable supervision signals and avoid representation collapse. Different from the basic information bottleneck, RGIB further decouples and balances the mutual dependence among graph topology, target labels, and representation, building new learning objectives for robust representation against the bilateral noise. Two instantiations, RGIB-SSL and RGIB-REP, are explored to leverage the merits of different methodologies, *i.e.*, self-supervised learning and data reparameterization, for implicit and explicit data denoising, respectively. Extensive experiments on six datasets and three GNNs with diverse noisy scenarios verify the effectiveness of our RGIB instantiations. The code is publicly available at: `https://github.com/tmlr-group/RGIB`.

## 1 Introduction

Link prediction [25] is the process of determining whether two nodes in a graph are likely to be connected. As a fundamental task in graph learning, it has attracted growing interest in real-world applications, *e.g.*, drug discovery [18], knowledge graph completion [3], and question answering [17].

While recent advances in graph neural networks (GNNs) [21, 12, 22, 48, 55] have achieved superior performances, the poor robustness under the edge noise is still a practical bottleneck to the current deep graph models [11, 10, 36, 8]. Early works explore improving the robustness of GNNs under the node label noise [7, 24] by means of the smoothing effect of neighboring nodes. Other methods achieve a similar goal via randomly removing edges [30] or actively selecting the informative nodes or edges and pruning the task-irrelevant ones [50, 27]. However, when applying these noise-robust methods to the link prediction with noise, only marginal improvements are achieved (Sec. 5). The attribution is that different from the label noise, the edge noise here is *bilateral*: it can naturally deteriorate both the *input* graph topology and the *target* edge labels, as illustrated in Fig. 1. Such a bilateral noise is practical in real-world graph data [11, 36]. Nonetheless, previous works that only consider unilateral noise in input space or label space cannot effectively deal with such a coupled scenario. This raises a new challenge of tackling the bilateral edge noise for robust link prediction.

---

[†]Correspondence to Bo Han (bhanml@comp.hkbu.edu.hk) and Jiangchao Yao (sunarker@sjtu.edu.cn).

37th Conference on Neural Information Processing Systems (NeurIPS 2023).

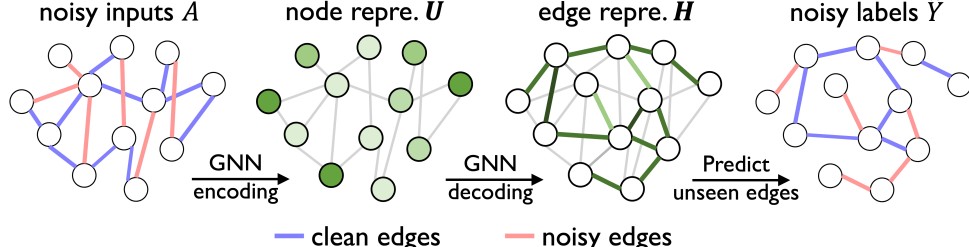

clean edges      noisy edges

Figure 1: Link prediction with bilateral edge noise. The GNN takes the graph $\mathcal{G} = (\tilde{A}, X)$ as inputs, generates the edge representation $\boldsymbol{H}$, and then predicts the existence of unseen edges with labels $\tilde{Y}$.

In this paper, we first disclose that the bilateral edge noise leads to the severe edge representation collapse behind the performance degradation (Sec. 3.1). Note the collapsed representation is reflected by much lower alignment and poorer uniformity (Sec. 3.2), which are used for quantitive and qualitative analysis of edge noise from representation perspective [35]. To solve this, we propose an information-theory-guided principle named Robust Graph Information Bottleneck (RGIB) (Sec. 4.1).

Conceptually, the RGIB principle is with new learning objectives that decouple the mutual information (MI) among noisy inputs, noisy labels, and representation. As illustrated in Fig. 2, RGIB generalizes the basic GIB [39] to learn a robust representation that is resistant to the bilateral edge noise. Technically, we provide two instantiations of RGIB based on different methodologies: (1) the RGIB-SSL utilizes contrastive pairs with automatically augmented views to form the informative regularization in a self-supervised learning manner, which benefits from

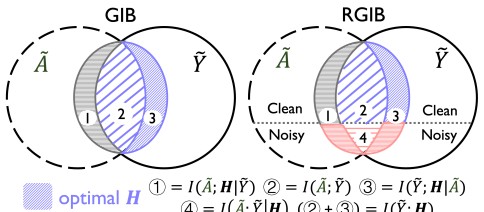

$\textcircled{1} = I(\tilde{A}; \boldsymbol{H}|\tilde{Y})$   $\textcircled{2} = I(\tilde{A}; \tilde{Y})$   $\textcircled{3} = I(\tilde{Y}; \boldsymbol{H}|\tilde{A})$
$\textcircled{4} = I(\tilde{A}; \tilde{Y}|\boldsymbol{H})$   $(\textcircled{2}+\textcircled{3}) = I(\tilde{Y}; \boldsymbol{H})$

Figure 2: The basic GIB and the proposed RGIB. Note RGIB balances signals $\textcircled{1}, \textcircled{3}, \textcircled{4}$ to capture the optimal $\boldsymbol{H}$ and discard noisy signals in red.

its intrinsic robustness to label noise (Sec. 4.2); (2) the RGIB-REP explicitly purifies the graph topology and supervision targets with the reparameterization mechanism, which enables further modeling of the mutual patterns of noise from both input and label spaces (Sec. 4.3). Both instantiations are equipped with adaptive designs, aiming to effectively estimate and balance the informative terms in a tractable manner, *i.e.*, the hybrid augmentation algorithm and self-adversarial alignment loss for RGIB-SSL; the relaxed information constraints on topology space and label space for RGIB-REP. Empirically, the two instantiations of RGIB work effectively under extensive noisy scenarios and can be seamlessly integrated with various GNNs (Sec. 5). Our contributions are summarized as follows.

- To our best knowledge, we are the first to study the robustness problem of link prediction under bilateral edge noise. We reveal that the bilateral noise can bring severe representation collapse and performance degradation, and such negative impacts are general to common datasets and GNNs.

- We propose a general learning framework, RGIB, with new representation learning objectives to promote GNNs' robustness. Two instantiations, RGIB-SSL and RGIB-REP, are proposed upon different methodologies that are equipped with adaptive designs and extensive theoretical analysis.

- Without modifying the GNN architectures, the RGIB achieves state-of-the-art results on 3 GNNs and 6 datasets under various noisy scenarios, obtaining up to $12.9\%$ promotion in AUC. The distribution of learned representations is notably recovered and more robust to the bilateral noise.

## 2 Preliminaries

**Notations.** We denote $\mathcal{V} = \{v_i\}_{i=1}^{N}$ as the set of nodes and $\mathcal{E} = \{e_{ij}\}_{ij=1}^{M}$ as the set of edges. With adjacent matrix $A$ and node features $X$, a graph is denoted as $\mathcal{G} = (A, X)$, where $A_{ij} = 1$ means there is an edge $e_{ij}$ between $v_i$ and $v_j$. $X_{[i,:]} \in \mathbb{R}^D$ is the $D$-dimension node feature of $v_i$. The link prediction task is to indicate the existence of query edges that are not observed in $A$. The binary labels of these query edges are denoted as $Y$, where 0 for negative edges and 1 for positive edges. The $I(\cdot; \cdot)$ indicates the mutual information between two variables. Detailed notations are in Tab. 9.

**GNNs for Link Prediction.** Here, we follow the encode-decode framework (Fig. 1) for link prediction [22], where the GNN architecture can be GCN [21], GAT [33], or SAGE [15]. Given a $L$-layer GNN $f_{\boldsymbol{w}}(\cdot)$ with learnable weights $\boldsymbol{w}$, the node representation $\boldsymbol{U} \in \mathbb{R}^{|\mathcal{V}| \times D}$ for each node $v_i \in \mathcal{V}$ are obtained by a $L$-layer message propagation as the encoding process. For decoding, the logits $\boldsymbol{\phi}_{e_{ij}}$ of each query edge $e_{ij}$ are computed by projecting the edge representation $\boldsymbol{h}_{ij} = \boldsymbol{u}_i \odot \boldsymbol{u}_j$ to probability. Note that the representation of all query edges is denoted as $\boldsymbol{H}$ and namely $f_{\boldsymbol{w}}(A, X) = \boldsymbol{H}$. Finally, the optimization objective is to minimize the binary classification loss with logits $\boldsymbol{\phi}_{e_{ij}}$ and labels $Y_{ij}$.

**Topological denoising approaches.** A natural way to tackle the input edge noise is to directly clean the noisy graph. Sampling-based methods, such as DropEdge [30], NeuralSparse [50], and PTDNet [27], are proposed to remove the task-irrelevant edges. Besides, as GNNs can be easily fooled by adversarial networks with only a few perturbed edges [4, 52, 9, 6], defending methods GCN-jaccard [38] and GIB [39] are designed for pruning adversarial edges from the poisoned graphs.

**Label-noise-resistant techniques.** To tackle the general problem of noisy labels, sample-selection-based methods such as Co-teaching [16] let two neural networks teach each other with small-loss samples based on the memorization effect [2]. Besides, peer loss function [26] pairs independent peer examples for supervision and works within the standard empirical risk minimization framework without noise priors. For tackling label noise on graphs, label propagation techniques [7, 24] are designed to propagate the reliable signals from clean nodes to noisy ones, which are nonetheless entangled with the node annotations and node classification task that cannot be directly applied here.

# 3 An Empirical Study of Bilateral Edge Noise

In this section, we attempt to figure out how GNNs behave when learning with the edge noise and what are the latent mechanisms behind it. We first present an empirical study in Sec. 3.1, and then investigate the negative impact of noise through the lens of representation distribution in Sec. 3.2.

## 3.1 How GNNs perform under bilateral edge noise?

To quantitatively study the impact of edge noise, we simulate different levels of perturbations properly on a range of GNNs benchmark datasets, as elaborated in Def. 3.1. Note the data split manner adopt by most relevant works [22, 48, 55] randomly divides partial edges as observations and the others as prediction targets. Hence, the noisy edges will be distributed to input $\tilde{A}$ and labels $\tilde{Y}$. With the bilateral noise defined in Def. 3.1, we then conduct a comprehensive empirical study. A further discussion on the edge noise is in Appendix B.1, and full evaluations can be found in Appendix D.1.

**Definition 3.1** (Bilateral edge noise). *Given a clean training data, i.e., observed graph $\mathcal{G} = (A, X)$ and labels $Y \in \{0, 1\}$ of query edges, the noisy adjacence $\tilde{A}$ is generated by directly adding edge noise to the original adjacent matrix $A$ while keeping the node features $X$ unchanged. The noisy labels $\tilde{Y}$ are similarly generated by adding edge noise to the labels $Y$. Specifically, given a noise ratio $\varepsilon_a$, the noisy edges $A'$ ($\tilde{A} = A + A'$) are generated by flipping the zero element in $A$ as one with the probability $\varepsilon_a$. It satisfies that $A' \odot A = O$ and $\varepsilon_a = {}^{|\mathtt{nonzero}(\tilde{A})| - |\mathtt{nonzero}(A)|} / {}_{|\mathtt{nonzero}(A)|}$. Similarly, noisy labels are generated and added to the original labels, where $\varepsilon_y = {}^{|\mathtt{nonzero}(\tilde{Y})| - |\mathtt{nonzero}(Y)|} / {}_{|\mathtt{nonzero}(Y)|}$.*

**Observation 3.1.** As illustrated in Fig. 3, *the bilateral edge noise causes a significant drop in performance, and a larger noise ratio generally leads to greater degradation.* It means that the standardly trained GNNs are vulnerable to the bilateral edge noise, yielding a severe robustness problem. As will be shown in Sec. 5, *the performance drop brought by the bilateral noise is much greater than that of the unilateral input noise or label noise.* However, none of the existing methods can effectively tackle such bilateral noise, since they only consider unilateral noise either in input space or label space. Thus, it is necessary to devise a robust method accordingly. To this end, we further investigate the underneath noise effect as follows.

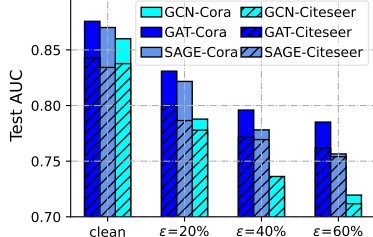

Figure 3: Link prediction performance in AUC with the bilateral edge noise. The bilateral noise ratio $\epsilon = 40\%$ means $\epsilon_a = \epsilon_y = 40\%$ with both noisy $\tilde{A}$ and $\tilde{Y}$.

Table 1: Mean values of alignment, which are calculated as the L2 distance of representations of two randomly perturbed graphs $\tilde{A}_1^i, \tilde{A}_2^i$, *i.e.*, $\texttt{Align} = \frac{1}{N} \sum_{i=1}^{N} \|\boldsymbol{H}_1^i - \boldsymbol{H}_2^i\|_2$. Representation $\boldsymbol{H}_1^i = f_{\boldsymbol{w}}(\tilde{A}_1^i, X)$ and $\boldsymbol{H}_2^i = f_{\boldsymbol{w}}(\tilde{A}_2^i, X)$.

| dataset | Cora | Citeseer |
|---|---|---|
| clean | .616 | .445 |
| $\varepsilon = 20\%$ | .687 | .586 |
| $\varepsilon = 40\%$ | .695 | .689 |
| $\varepsilon = 60\%$ | .732 | .696 |

(a) Clean data  (b) $\varepsilon = 20\%$  (c) $\varepsilon = 40\%$  (d) $\varepsilon = 60\%$

Figure 4: Uniformity distribution on Cora dataset. Representations of query edges in the test set are mapped to unit circle of $\mathbb{R}^2$ with normalization followed by the Gaussian kernel density estimation as [35]. Both positive and negative edges are expected to be uniformly distributed.

## 3.2 Understanding the noise impact via inspecting the representation distribution

Recall that the edge representation is extracted by a forward pass as $\boldsymbol{H} = f_{\boldsymbol{w}}(\tilde{A}, X)$ and the model is optimized by stochastic gradient descent as $\boldsymbol{w} := \boldsymbol{w} - \eta \nabla_{\boldsymbol{w}} \mathcal{L}(\boldsymbol{H}, \tilde{Y})$. When encountering noise within $\tilde{A}$ and $\tilde{Y}$, the edge representation $\boldsymbol{H}$ can be directly influenced, since the training neglects the adverse effects of data corruption. Besides, the GNN readouts the edge logit $\phi_{e_{ij}}$ based on node representations $\boldsymbol{h}_i$ and $\boldsymbol{h}_j$, which are possibly collapsed under the bilateral edge noise [14, 28].

To analyze the edge noise from the perspective of representation $\boldsymbol{H}$, we use two recent concepts [35] in representation learning: (1) *alignment* quantifies the stability of GNN when encountering edge noise in the testing phase. It is computed as the distance of representations between two randomly augmented graphs. A higher alignment means being more resistant to input perturbations; (2) *uniformity* qualitatively measures the denoising effects of GNN when learning with edge noise in the training phase. Overall, a greater uniformity implies that the representation of query edges are more uniformly distributed on the unit hypersphere, preserving more information about the original data.

**Observation 3.2.** As can be seen from Tab. 1 and Fig. 4, *a poorer alignment and a worse uniformity are brought by a severer edge noise.* As the noise ratio $\epsilon$ increases, the learned GNN $f_{\boldsymbol{w}}(\cdot)$ is more sensitive to input perturbations as the alignment values are sharply increased, and the learned edge representations tend to be less uniformly distributed and gradually collapse into individual subregions. That is, the learned edge representation $\boldsymbol{H}$ is *severely collapsed* under the bilateral edge noise, which is strongly correlated with the observation in Sec. 3.1 and reflects an undesirable property of GNNs.

# 4 Robust Graph Information Bottleneck

Without prior knowledge like the noise ratio, without the assistance of auxiliary datasets, and even without modifying GNN architectures, how can representations be resistant to the bilateral noise? Here, we formally build a method to address the above problems from the perspective of robust graph information bottleneck (Sec. 4.1) and design its two practical instantiations (Sec. 4.2 and Sec. 4.3).

## 4.1 The principle of RGIB

Recall in Sec. 3.2, the edge representation $\boldsymbol{H}$ is degraded. To robustify $\boldsymbol{H}$, one can naturally utilize the information constraint based on the graph information bottleneck (GIB) [39, 46], *i.e.*, solving

$$\min \text{GIB} \triangleq -I(\boldsymbol{H}; \tilde{Y}), \text{ s.t. } I(\boldsymbol{H}; \tilde{A}) < \gamma, \tag{1}$$

where the hyper-parameter $\gamma$ constrains the MI $I(\boldsymbol{H}; \tilde{A})$ to avoid $\boldsymbol{H}$ from capturing excess task-irrelevant information from $\tilde{A}$. The basic GIB can effectively defend the input perturbation [39]. However, it is intrinsically vulnerable to label noise since it entirely preserves the label supervision and maximizes the noisy supervision $I(\boldsymbol{H}; \tilde{Y})$. Our empirical results in Sec. 5 show that optimization only with Eq. 1 is ineffective when learning with the bilateral edge noise defined in Def. 3.1.

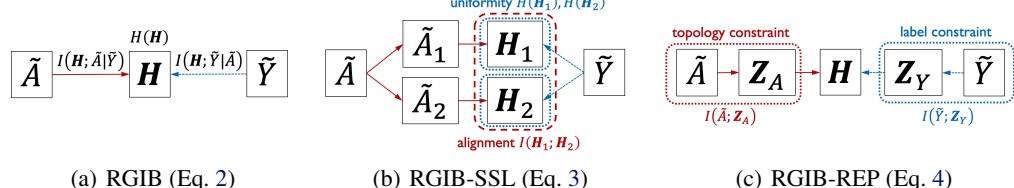

(a) RGIB (Eq. 2)  (b) RGIB-SSL (Eq. 3)  (c) RGIB-REP (Eq. 4)

Figure 5: Digrams of the RGIB principle (a) and its two instantiations (b,c). RGIB-SSL utilizes the automatically augmented views $\tilde{A}_1, \tilde{A}_2$ in a contrastive manner to be resistant to input noise. RGIB-SSL is intrinsically robust to label noise due to its self-supervised nature. Besides, RGIB-REP explicitly purifies the input graph's topology and target labels with the jointly reparameterized $\boldsymbol{Z}_A$ and $\boldsymbol{Z}_Y$. It enables to modeling the mutual patterns of edge noise from both input and label spaces.

**Deriving the RGIB principle.** The basic GIB of Eq. 1 decreases $I(\boldsymbol{H}; \tilde{A}|\tilde{Y})$ by directly constraining $I(\boldsymbol{H}; \tilde{A})$ to handle the input noise. Symmetrically, the label noise can be hidden in $I(\boldsymbol{H}; \tilde{Y}|\tilde{A})$, but trivially constraining $I(\boldsymbol{H}; \tilde{Y})$ to regularize $I(\boldsymbol{H}; \tilde{Y}|\tilde{A})$ is not ideal, since it conflicts with Eq. 1 and also cannot tackle the noise within $I(\tilde{A}; \tilde{Y})$. Thus, it is crucial to further decouple the dependence among $\tilde{A}, \tilde{Y}$, and $\boldsymbol{H}$, while the noise can exist in areas of $I(\boldsymbol{H}; \tilde{Y}|\tilde{A})$, $I(\boldsymbol{H}; \tilde{A}|\tilde{Y})$, and $I(\tilde{A}; \tilde{Y}|\boldsymbol{H})$. Analytically, $I(\tilde{A}; \tilde{Y}|\boldsymbol{H}) = I(\tilde{A}; \tilde{Y}) + I(\boldsymbol{H}; \tilde{Y}|\tilde{A}) + I(\boldsymbol{H}; \tilde{A}|\tilde{Y}) - H(\boldsymbol{H}) + H(\boldsymbol{H}|\tilde{A}, \tilde{Y})$, where $I(\tilde{A}; \tilde{Y})$ is a constant and redundancy $H(\boldsymbol{H}|\tilde{A}, \tilde{Y})$ can be minimized. Thus, the $I(\tilde{A}; \tilde{Y}|\boldsymbol{H})$ can be approximated by $H(\boldsymbol{H})$, $I(\boldsymbol{H}; \tilde{Y}|\tilde{A})$ and $I(\boldsymbol{H}; \tilde{A}|\tilde{Y})$. Since the two later terms are also noisy, a balance of these three informative terms can be a solution to the problem of bilateral edge noise.

**Definition 4.1** (Robust Graph Information Bottleneck). *Based on the above analysis, we propose a new learning objective to balance informative signals regarding $\boldsymbol{H}$, as illustrated in Fig. 5(a), i.e.,*

$$\min RGIB \triangleq -I(\boldsymbol{H}; \tilde{Y}), \;\; s.t. \; \gamma_H^- < H(\boldsymbol{H}) < \gamma_H^+, I(\boldsymbol{H}; \tilde{Y}|\tilde{A}) < \gamma_Y, \; I(\boldsymbol{H}; \tilde{A}|\tilde{Y}) < \gamma_A. \quad (2)$$

*Specifically, constraints on $H(\boldsymbol{H})$ encourage a diverse $\boldsymbol{H}$ to prevent representation collapse ($> \gamma_H^-$) and also limit its capacity ($< \gamma_H^+$) to avoid over-fitting. Another two MI terms, $I(\boldsymbol{H}; \tilde{Y}|\tilde{A})$ and $I(\boldsymbol{H}; \tilde{A}|\tilde{Y})$, mutually regularize posteriors to mitigate the negative impact of bilateral noise on $\boldsymbol{H}$. The complete derivation of RGIB and a further comparison of RGIB and GIB are in Appendix B.2.*

Note that MI terms, *e.g.*, $I(\boldsymbol{H}; \tilde{A}|\tilde{Y})$ are usually intractable. Therefore, we introduce two practical implementations of RGIB, *i.e.*, RGIB-SSL and RGIB-REP, based on different methodologies. RGIB-SSL *explicitly* optimizes the representation $\boldsymbol{H}$ with the self-supervised regularization, while RGIB-REP *implicitly* optimizes $\boldsymbol{H}$ by purifying the noisy $\tilde{A}$ and $\tilde{Y}$ with the reparameterization mechanism.

## 4.2  Instantiating RGIB with self-supervised learning

Recall that the edge representation has deteriorated with the *supervised* learning paradigm. Naturally, we modify it into a *self-supervised* counterpart by explicitly regularizing the representation $\boldsymbol{H}$ (see Fig. 5(b)) to avoid collapse and to implicitly capture reliable relations among noisy edges as

$$\min \text{RGIB-SSL} \triangleq - \underbrace{\lambda_s (I(\boldsymbol{H}_1; \tilde{Y}) + I(\boldsymbol{H}_2; \tilde{Y}))}_{\text{supervision}} - \underbrace{\lambda_u (H(\boldsymbol{H}_1) + H(\boldsymbol{H}_2))}_{\text{uniformity}} - \underbrace{\lambda_a I(\boldsymbol{H}_1; \boldsymbol{H}_2)}_{\text{alignment}}, \quad (3)$$

where margins $\lambda_s, \lambda_u, \lambda_a$ balance one supervised and two self-supervised regularization terms. When $\lambda_s \equiv 1, \lambda_u \equiv 0$, the RGIB-SSL can be degenerated to the basic GIB. Note $\boldsymbol{H}_1$ and $\boldsymbol{H}_2$ are the representations of two augmented views $\tilde{A}_1$ and $\tilde{A}_2$, namely, $\boldsymbol{H}_1 = f_{\boldsymbol{w}}(\tilde{A}_1, X)$ and $\boldsymbol{H}_2 = f_{\boldsymbol{w}}(\tilde{A}_2, X)$.

**Instantiation.** To achieve a tractable approximation of the MI terms in Eq. 2, we adopt the contrastive learning technique [5], and contrast pair of samples, *i.e.*, perturbed $\tilde{A}_1, \tilde{A}_2$ that are sampled from the augmentation distribution $\mathbb{P}(\tilde{A})$. Based on which we approximate the expected supervision term by $\mathbb{E}[I(\boldsymbol{H}; \tilde{Y}|\tilde{A})] \leq \mathbb{E}[I(\boldsymbol{H}; \tilde{Y})] = \mathbb{E}_{\tilde{A}_s \sim \mathbb{P}(\tilde{A})}[I(\boldsymbol{H}_s; \tilde{Y})] \approx 1/2(\mathcal{L}_{cls}(\boldsymbol{H}_1; \tilde{Y}) + \mathcal{L}_{cls}(\boldsymbol{H}_2; \tilde{Y}))$. Note the approximation also supports multiple samples. Similarly, we approximate the entropy term $\mathbb{E}[H(\boldsymbol{H})]$ by $1/2(H(\boldsymbol{H}_1) + H(\boldsymbol{H}_2))$, where a higher uniformity leads to a high entropy $H(\boldsymbol{H})$ as proved in Prop. 4.2. Lastly, the $I(\boldsymbol{H}; \tilde{A}|\tilde{Y})$ is estimated by the alignment term $I(\boldsymbol{H}_1; \boldsymbol{H}_2)$ (refer to Prop. 4.3).

**Proposition 4.2.** *A higher information entropy $H(\boldsymbol{H})$ of edge representation $\boldsymbol{H}$ indicates a higher uniformity [35] of the representation's distribution on the unit hypersphere. Proof. See Appendix A.3.*

**Proposition 4.3.** *A lower alignment $I(\boldsymbol{H}_1; \boldsymbol{H}_2)$ indicates a lower $I(\boldsymbol{H}; \tilde{A}|\tilde{Y})$. Since $I(\boldsymbol{H}; \tilde{A}|\tilde{Y}) \leq I(\boldsymbol{H}; \tilde{A}) \leq \frac{1}{2}\big(I(\boldsymbol{H}_1; \boldsymbol{H}_2) + I(\tilde{A}_1; \tilde{A}_2)\big) = \frac{1}{2}\big(I(\boldsymbol{H}_1; \boldsymbol{H}_2) + c\big)$, a constrained alignment estimated by $I(\boldsymbol{H}_1; \boldsymbol{H}_2)$ can bound a lower $I(\boldsymbol{H}; \tilde{A}|\tilde{Y})$ and $I(\boldsymbol{H}; \tilde{A})$. Proof. See Appendix A.4.*

However, directly applying existing contrastive methods like [5, 20] can be easily suboptimal, since they are not originally designed for graph data and neglect the internal correlation between topology $\tilde{A}$ and target $\tilde{Y}$. Here, we propose the following two designs to further improve robust learning.

**Hybrid graph augmentation.** To encourage more diverse views with lower $I(A_1; A_2)$ and to avoid manual selection of augmentation operations, we propose a hybrid augmentation method with four augmentation operations as predefined candidates and ranges of their corresponding hyper-parameters. In each training iteration, two augmentation operators and their hyper-parameters are *automatically* sampled from the search space. Then, two augmented graphs are obtained by applying the two operators on the original graph adjacency $A$. The detailed algorithm is elaborated in Appendix C.

**Self-adversarial loss terms.** With edge representations $\boldsymbol{H}_1$ and $\boldsymbol{H}_2$ from two augmented views, we build the alignment objective by minimizing the representation similarity of the positive pairs $(\boldsymbol{h}_{ij}^1, \boldsymbol{h}_{ij}^2)$ and maximizing that of the randomly sampled negative pairs $(\boldsymbol{h}_{ij}^1, \boldsymbol{h}_{mn}^2)$, $e_{ij} \neq e_{mn}$. The proposed self-adversarial alignment loss is $\mathcal{R}_{align} = \sum_{i=1}^{N} \mathcal{R}_i^{pos} + \mathcal{R}_i^{neg}$. [2] Importantly, softmax functions $p^{pos}(\cdot)$ and $p^{neg}(\cdot)$ aim to mitigate the inefficiency problem [32] that aligned pairs are not informative. Besides, the uniformity loss is $\mathcal{R}_{unif} = \sum_{ij,mn}^{K} e^{-\left\|\boldsymbol{h}_{ij}^1 - \boldsymbol{h}_{mn}^1\right\|_2^2} + e^{-\left\|\boldsymbol{h}_{ij}^2 - \boldsymbol{h}_{mn}^2\right\|_2^2}$ with the Gaussian potential kernel, where edges $e_{ij}$ and $e_{mn}$ are respectively sampled from $\mathcal{E}^{pos}$ and $\mathcal{E}^{neg}$.

**Optimization.** Regarding Eq. 3, the objective of RGIB-SSL is $\mathcal{L} = \lambda_s \mathcal{L}_{cls} + \lambda_a \mathcal{R}_{align} + \lambda_u \mathcal{R}_{unif}$.

**Remark 4.1.** *The collapsed representation comes from trivially minimizing the noisy supervision $I(\boldsymbol{H}; \tilde{Y})$. The alignment and uniformity terms in Eq. 3 can alleviate such noise effects (see Sec. 5).*

### 4.3 Instantiating RGIB with data reparameterization

Another realization is by reparameterizing the graph data on both topology space and label space jointly to preserve clean information and discard noise (as Fig. 5(c)). We propose RGIB-REP that explicitly models the reliability of $\tilde{A}$ and $\tilde{Y}$ via latent variables $\boldsymbol{Z}$ to learn a noise-resistant $\boldsymbol{H}$, *i.e.*,

$$\min \text{RGIB-REP} \triangleq -\underbrace{\lambda_s I(\boldsymbol{H}; \boldsymbol{Z}_Y)}_{\text{supervision}} + \underbrace{\lambda_A I(\boldsymbol{Z}_A; \tilde{A})}_{\text{topology constraint}} + \underbrace{\lambda_Y I(\boldsymbol{Z}_Y; \tilde{Y})}_{\text{label constraint}}, \tag{4}$$

where latent variables $\boldsymbol{Z}_Y$ and $\boldsymbol{Z}_A$ are clean signals extracted from noisy $\tilde{Y}$ and $\tilde{A}$. Their complementary parts $\boldsymbol{Z}_{Y'}$ and $\boldsymbol{Z}_{A'}$ are considered as noise, satisfying $\tilde{Y} = \boldsymbol{Z}_Y + \boldsymbol{Z}_{Y'}$ and $\tilde{A} = \boldsymbol{Z}_A + \boldsymbol{Z}_{A'}$. When $\boldsymbol{Z}_Y \equiv \tilde{Y}$ and $\boldsymbol{Z}_A \equiv \tilde{A}$, the RGIB-REP can be degenerated to the basic GIB. Here, the $I(\boldsymbol{H}; \boldsymbol{Z}_Y)$ measures the supervised signals with selected samples $\boldsymbol{Z}_Y$, where the classifier takes $\boldsymbol{Z}_A$ (*i.e.*, a subgraph of $\tilde{A}$) as input instead of the original $\tilde{A}$, *i.e.*, $\boldsymbol{H} = f_{\boldsymbol{w}}(\boldsymbol{Z}_A, X)$. Constraints $I(\boldsymbol{Z}_A; \tilde{A})$ and $I(\boldsymbol{Z}_Y; \tilde{Y})$ aim to select the cleanest and most task-relevant information from $\tilde{A}$ and $\tilde{Y}$.

**Instantiation.** For deriving a tractable objective regarding $\boldsymbol{Z}_A$ and $\boldsymbol{Z}_Y$, a parameterized sampler $f_{\boldsymbol{\phi}}(\cdot)$ sharing the same architecture and weights as $f_{\boldsymbol{w}}(\cdot)$ is adopted here. $f_{\boldsymbol{\phi}}(\cdot)$ generates the probabilistic distribution of edges that include both $\tilde{A}$ and $\tilde{Y}$ by $\boldsymbol{P} = \sigma(\boldsymbol{H}_{\boldsymbol{\phi}} \boldsymbol{H}_{\boldsymbol{\phi}}^\top) \in (0, 1)^{|\mathcal{V}| \times |\mathcal{V}|}$, where representation $\boldsymbol{H}_{\boldsymbol{\phi}} = f_{\boldsymbol{\phi}}(\tilde{A}, X)$. Bernoulli sampling is then used to obtain high-confidence edges, *i.e.*, $\boldsymbol{Z}_A = \texttt{SAMP}(\boldsymbol{P}|\tilde{A})$ and $\boldsymbol{Z}_Y = \texttt{SAMP}(\boldsymbol{P}|\tilde{Y})$, where $|\boldsymbol{Z}_A| \leq |\tilde{A}|$ and $|\boldsymbol{Z}_Y| \leq |\tilde{Y}|$.

**Proposition 4.4.** *Given the edge number $n$ of $\tilde{A}$, the marginal distribution of $\boldsymbol{Z}_A$ is $\mathbb{Q}(\boldsymbol{Z}_A) = \mathbb{P}(n) \prod_{\tilde{A}_{ij}=1}^{n} \boldsymbol{P}_{ij}$. $\boldsymbol{Z}_A$ satisfies $I(\boldsymbol{Z}_A; \tilde{A}) \leq \mathbb{E}[KL(\mathbb{P}_{\boldsymbol{\phi}}(\boldsymbol{Z}_A|A) || \mathbb{Q}(\boldsymbol{Z}_A))] = \sum_{e_{ij} \in \tilde{A}} \boldsymbol{P}_{ij} \log \frac{\boldsymbol{P}_{ij}}{\tau} + (1 - \boldsymbol{P}_{ij}) \log \frac{1 - \boldsymbol{P}_{ij}}{1 - \tau} = \mathcal{R}_A$, where $\tau$ is a constant. The topology constraint $I(\boldsymbol{Z}_A; \tilde{A})$ in Eq. 4 is bounded by $\mathcal{R}_A$, and the label constraint is similarly bounded by $\mathcal{R}_Y$. Proof. See Appendix A.5.*

---

[2] $\mathcal{R}_i^{pos} = p^{pos}(\boldsymbol{h}_{ij}^1, \boldsymbol{h}_{ij}^2) \cdot \left\|\boldsymbol{h}_{ij}^1 - \boldsymbol{h}_{ij}^2\right\|_2^2$, where $p^{pos}(\boldsymbol{h}_{ij}^1, \boldsymbol{h}_{ij}^2) = \exp(\left\|\boldsymbol{h}_{ij}^1 - \boldsymbol{h}_{ij}^2\right\|_2^2)/\sum_{i=1}^{N} \exp(\left\|\boldsymbol{h}_{ij}^1 - \boldsymbol{h}_{ij}^2\right\|_2^2)$. $\mathcal{R}_i^{neg} = p^{neg}(\boldsymbol{h}_{ij}^1, \boldsymbol{h}_{mn}^2) \cdot (\gamma - \left\|\boldsymbol{h}_{ij}^1 - \boldsymbol{h}_{mn}^2\right\|_2^2), p^{neg}(\boldsymbol{h}_{ij}^1, \boldsymbol{h}_{mn}^2) = \exp(\alpha - \left\|\boldsymbol{h}_{ij}^1 - \boldsymbol{h}_{mn}^2\right\|_2^2)/\sum_{i=1}^{N} \exp(\alpha - \left\|\boldsymbol{h}_{ij}^1 - \boldsymbol{h}_{mn}^2\right\|_2^2)$.

Table 2: Method comparison with GCN under bilateral noise, *i.e.*, both the input and label noise exist. The **boldface** mean the best results in AUC, while the underlines indicate the second-best results.

| method | Cora | | | Citeseer | | | Pubmed | | | Facebook | | | Chameleon | | | Squirrel | | |
|---|---|---|---|---|---|---|---|---|---|---|---|---|---|---|---|---|---|---|
| | 20% | 40% | 60% | 20% | 40% | 60% | 20% | 40% | 60% | 20% | 40% | 60% | 20% | 40% | 60% | 20% | 40% | 60% |
| Standard | .8111 | .7419 | .6970 | .7864 | .7380 | .7085 | .8870 | .8748 | .8641 | .9829 | .9520 | .9438 | .9616 | .9496 | .9274 | .9432 | .9406 | .9386 |
| DropEdge | .8017 | .7423 | .7303 | .7635 | .7393 | .7094 | .8711 | .8482 | .8354 | .9811 | .9682 | .9473 | .9568 | .9548 | .9407 | .9439 | .9377 | .9365 |
| NeuralSparse | .8190 | .7318 | .7293 | .7765 | .7397 | .7148 | .8908 | .8733 | .8630 | .9825 | .9638 | .9456 | .9599 | .9497 | .9402 | .9494 | .9309 | .9297 |
| PTDNet | .8047 | .7559 | .7388 | .7795 | .7423 | .7283 | .8872 | .8733 | .8623 | .9725 | .9674 | .9485 | .9607 | .9514 | .9424 | .9485 | .9326 | .9304 |
| Co-teaching | .8197 | .7479 | .7030 | .7533 | .7238 | .7131 | .8943 | .8760 | .8638 | .9820 | .9526 | .9480 | .9595 | .9516 | .9483 | .9461 | .9352 | .9374 |
| Peer loss | .8185 | .7468 | .7018 | .7423 | .7345 | .7104 | .8961 | .8815 | .8630 | .9807 | .9536 | .9430 | .9543 | .9533 | .9267 | .9457 | .9345 | .9286 |
| Jaccard | .8143 | .7498 | .7024 | .7473 | .7324 | .7107 | .8872 | .8803 | .8512 | .9794 | .9579 | .9428 | .9503 | .9538 | .9344 | .9443 | .9327 | .9244 |
| GIB | .8198 | .7485 | .7148 | .7509 | .7388 | .7121 | .8899 | .8729 | .8544 | .9773 | .9608 | .9417 | .9554 | .9561 | .9321 | .9472 | .9329 | .9302 |
| VIB | .8208 | .7810 | .7218 | .7701 | .8120 | .7185 | .8927 | .8825 | .8501 | .9697 | .9637 | .9500 | .9529 | .9561 | .9487 | .9431 | .9399 | .9288 |
| PRI | .7976 | .7330 | .6981 | .7567 | .7452 | .7018 | .8898 | .8801 | .8487 | .9601 | .9619 | .9507 | .9513 | .9499 | .9490 | .9382 | .9413 | .9301 |
| SupCon | .8240 | .7819 | .7490 | .7554 | .7458 | .7299 | .8853 | .8718 | .8525 | .9588 | .9508 | .9297 | .9561 | .9531 | .9467 | .9473 | .9348 | .9301 |
| GRACE | .7872 | .6940 | .6929 | .7632 | .7242 | .6844 | .8922 | .8749 | .8588 | .8899 | .8865 | .8315 | .8978 | .8987 | .8949 | .9394 | .9380 | .9363 |
| **RGIB-REP** | .8313 | .7966 | .7591 | .7875 | .7519 | .7312 | .9017 | .8834 | .8652 | **.9832** | **.9770** | .9519 | **.9723** | **.9621** | **.9519** | **.9509** | **.9455** | **.9434** |
| **RGIB-SSL** | **.8930** | **.8554** | **.8339** | **.8694** | **.8427** | **.8137** | **.9225** | **.8918** | **.8697** | .9829 | .9711 | **.9643** | .9655 | .9592 | .9500 | .9499 | .9426 | .9425 |

**Proposition 4.5.** *The supervision term $I(\boldsymbol{H}; \boldsymbol{Z}_Y)$ in Eq. 4 can be empirically reduced to the classification loss, i.e., $I(\boldsymbol{H}; \boldsymbol{Z}_Y) \geq \mathbb{E}_{\boldsymbol{Z}_Y, \boldsymbol{Z}_A}[\log \mathbb{P}_{\boldsymbol{w}}(\boldsymbol{Z}_Y | \boldsymbol{Z}_A)] \approx -\mathcal{L}_{cls}(f_{\boldsymbol{w}}(\boldsymbol{Z}_A), \boldsymbol{Z}_Y)$, where $\mathcal{L}_{cls}$ is the standard cross-entropy loss. Proof. See Appendix A.6.*

**Optimization.** A relaxation is then conducted on the three MI terms in Eq. 4. With derived bounds in Prop. 4.4 (*i.e.*, regularization $\mathcal{R}_A$ and $\mathcal{R}_Y$) and Prop. 4.5 (*i.e.*, $\mathcal{L}_{cls}$), the final optimization objective is formed as $\mathcal{L} = \lambda_s \mathcal{L}_{cls} + \lambda_A \mathcal{R}_A + \lambda_Y \mathcal{R}_Y$, and the corresponding analysis Thm. 4.6 is as follows.

**Theorem 4.6.** *Assume the noisy training data $D_{train} = (\tilde{A}, X, \tilde{Y})$ contains a potentially clean subset $D_{sub} = (\boldsymbol{Z}_A^*, X, \boldsymbol{Z}_Y^*)$. The $\boldsymbol{Z}_Y^*$ and $\boldsymbol{Z}_A^*$ are the optimal solutions of Eq. 4 that $\boldsymbol{Z}_Y^* \approx Y$, based on which a trained GNN predictor $f_{\boldsymbol{w}}(\cdot)$ satisfies $f_{\boldsymbol{w}}(\boldsymbol{Z}_A^*, X) = \boldsymbol{Z}_Y^* + \epsilon$. The random error $\epsilon$ is independent of $D_{sub}$ and $\epsilon \to 0$. Then, for arbitrary $\lambda_s, \lambda_A, \lambda_Y \in [0, 1]$, $\boldsymbol{Z}_A = \boldsymbol{Z}_A^*$ and $\boldsymbol{Z}_Y = \boldsymbol{Z}_Y^*$ minimizes the RGIB-REP of Eq. 4. Proof. See Appendix A.7.*

**Remark 4.2.** *Note that the methodologies take by RGIB-SSL and RGIB-REP, i.e., self-supervised learning and data reparameterization are not considered in the original GIB [39]. In Sec. 5, we justify that they are effective in instantiating the RGIB to handle the bilateral edge noise. A detailed comparison of the two instantiations is conducted in Appendix B.3. More importantly, it is possible that new instantiations based on other methodologies are inspired by the general RGIB principle.*

## 5  Experiments

**Setup.** 6 popular datasets and 3 types of GNNs are taken in the experiments. The edge noise is generated based on Def. 3.1 after the commonly used data split where 85% edges are randomly selected for training, 5% as the validation set, and 10% for testing. The AUC is used as the evaluation metric as in [48, 55]. The software framework is the Pytorch [29], while the hardware is one NVIDIA RTX 3090 GPU. We repeat all experiments five times, and full results can be found in Appendix E.

**Baselines.** As existing robust methods separately deal with input noise or label noise, both kinds of methods can be considered as baselines. For tackling *input noise*, three sampling-based approaches are used for comparison, *i.e.*, DropEdge [30], NeuralSparse [50], and PTDNet [27]. Besides, we also include Jaccard [38], GIB [39], VIB [31], and PRI [47], which are designed for pruning adversarial edges. Two generic methods are selected for *label noise*, *i.e.*, Co-teaching [16] and Peer loss [26]. Besides, two contrastive learning methods are taken into comparison, *i.e.*, SupCon [20] and GRACE [54]. The implementation details of the above baselines are summarized in Appendix C.

### 5.1  Main results

**Performance comparison.** As shown in Tab. 2, the RGIB achieves the best results in all 6 datasets under the bilateral edge noise with various noise ratios, especially on challenging datasets, *e.g.*, Cora and Citeseer, where a 12.9% AUC promotion can be gained compared with the second-best methods. As for the unilateral noise settings shown in Tab. 3, RGIB still consistently surpasses all the baselines ad hoc for unilateral input noise or label noise by a large margin. We show that the bilateral edge noise can be formulated and solved by a unified learning framework, while the previous denoising methods can only work for one specific noise pattern, *e.g.*, the unilateral input noise or label noise.

Table 3: Method comparison with GCN under unilateral input noise (upper) or label noise (lower). The **boldface** numbers mean the best results, while the underlines indicate the second-best results.

| input noise | Cora | | | Citeseer | | | Pubmed | | | Facebook | | | Chameleon | | | Squirrel | | |
|---|---|---|---|---|---|---|---|---|---|---|---|---|---|---|---|---|---|---|
| | 20% | 40% | 60% | 20% | 40% | 60% | 20% | 40% | 60% | 20% | 40% | 60% | 20% | 40% | 60% | 20% | 40% | 60% |
| Standard | .8027 | .7856 | .7490 | .8054 | .7708 | .7583 | .8854 | .8759 | .8651 | .9819 | .9668 | .9622 | .9608 | .9433 | .9368 | .9416 | .9395 | .9411 |
| DropEdge | .8338 | .7826 | .7454 | .8025 | .7730 | .7473 | .8682 | .8456 | .8376 | .9803 | .9685 | .9531 | .9567 | .9433 | .9432 | .9426 | .9376 | .9358 |
| NeuralSparse | .8534 | .7794 | .7637 | .8093 | .7809 | .7468 | .8931 | .8720 | .8649 | .9712 | .9691 | .9583 | .9609 | .9540 | .9348 | .9469 | .9403 | .9417 |
| PTDNet | .8433 | .8214 | .7770 | .8119 | .7811 | .7638 | .8903 | .8776 | .8609 | .9725 | .9668 | .9493 | .9610 | .9457 | .9360 | .9469 | .9400 | .9379 |
| Jaccard | .8200 | .7838 | .7617 | .8176 | .7776 | .7725 | .8987 | .8764 | .8639 | .9784 | .9702 | .9638 | .9507 | .9436 | .9364 | .9388 | .9345 | .9240 |
| GIB | .8002 | .8099 | .7741 | .8070 | .7717 | .7798 | .8932 | .8808 | .8618 | .9796 | .9647 | .9650 | .9605 | .9521 | .9416 | .9390 | .9406 | .9397 |
| VIB | .8603 | **.8590** | .8008 | .8497 | .8399 | .7703 | .8910 | .8829 | .8519 | .9800 | .9710 | .9536 | .9499 | .9558 | .9312 | .9416 | .9402 | .9297 |
| PRI | .8307 | .7990 | .7736 | .8105 | .7852 | .7558 | .8801 | .8790 | .8551 | .9691 | .9698 | .9529 | .9528 | .9501 | .9435 | .9397 | .9419 | .9318 |
| SupCon | .8349 | .8301 | .8025 | .8076 | .7767 | .7655 | .8867 | .8739 | .8558 | .9647 | .9517 | .9401 | .9606 | .9536 | .9468 | .9372 | .9343 | .9305 |
| GRACE | .7877 | .7107 | .6975 | .7615 | .7151 | .6830 | .8810 | .8795 | .8593 | .9015 | .8833 | .8395 | .8994 | .9007 | .8964 | .9392 | .9378 | .9363 |
| **RGIB-REP** | .8624 | .8313 | .8158 | .8299 | .7996 | .7771 | .9008 | .8822 | .8687 | **.9833** | **.9723** | **.9682** | **.9705** | **.9604** | .9480 | **.9495** | **.9432** | .9405 |
| **RGIB-SSL** | **.9024** | .8577 | **.8421** | **.8747** | **.8461** | **.8245** | **.9126** | **.8889** | **.8693** | .9821 | .9707 | .9668 | .9658 | .9570 | **.9486** | .9479 | .9429 | **.9429** |

| label noise | Cora | | | Citeseer | | | Pubmed | | | Facebook | | | Chameleon | | | Squirrel | | |
|---|---|---|---|---|---|---|---|---|---|---|---|---|---|---|---|---|---|---|
| | 20% | 40% | 60% | 20% | 40% | 60% | 20% | 40% | 60% | 20% | 40% | 60% | 20% | 40% | 60% | 20% | 40% | 60% |
| Standard | .8281 | .8054 | .8060 | .7965 | .7850 | .7659 | .9030 | .9039 | .9070 | .9882 | .9880 | .9886 | .9686 | .9580 | .9362 | .9720 | .9720 | .9710 |
| Co-teaching | .8446 | .8209 | .8157 | .7974 | .7877 | .7913 | .9315 | .9291 | .9319 | .9762 | .9797 | .9638 | .9642 | .9650 | .9533 | .9675 | .9641 | .9655 |
| Peer loss | .8325 | .8036 | .8069 | .7991 | .7990 | .7751 | .9126 | .9101 | .9210 | .9769 | .9750 | .9734 | .9621 | .9501 | .9569 | .9636 | .9694 | .9696 |
| Jaccard | .8289 | .8064 | .8148 | .8061 | .7887 | .7689 | .9098 | .9135 | .9096 | .9702 | .9725 | .9758 | .9603 | .9659 | .9557 | .9529 | .9512 | .9501 |
| GIB | .8337 | .8137 | .8157 | .7986 | .7852 | .7649 | .9037 | .9114 | .9064 | .9742 | .9703 | .9771 | .9651 | .9582 | .9469 | .9641 | .9628 | .9601 |
| VIB | .8406 | .8296 | .8036 | .8068 | .8034 | .7739 | .9088 | .9042 | .8981 | .9705 | .9790 | .9713 | .9610 | .9629 | .9518 | .9568 | .9722 | .9690 |
| PRI | .8298 | .8139 | .7960 | .7902 | .7881 | .7725 | .8939 | .8917 | .8832 | .9733 | .9802 | .9620 | .9414 | .9593 | .9599 | .9658 | .9698 | .9533 |
| SupCon | .8491 | .8275 | .8256 | .8024 | .7983 | .7807 | .9131 | .9108 | .9162 | .9647 | .9567 | .9553 | .9584 | .9580 | .9477 | .9516 | .9595 | .9511 |
| GRACE | .8531 | .8237 | .8193 | .7909 | .7630 | .7737 | .9234 | .9252 | .9255 | .8913 | .8972 | .8887 | .9053 | .9074 | .9075 | .9171 | .9174 | .9166 |
| **RGIB-REP** | .8554 | .8318 | **.8297** | .8083 | .7846 | .7945 | .9357 | .9343 | **.9332** | **.9884** | **.9883** | **.9889** | **.9785** | **.9797** | **.9785** | **.9735** | **.9733** | **.9737** |
| **RGIB-SSL** | **.9314** | **.9224** | **.9241** | **.9204** | **.9218** | **.9250** | **.9594** | **.9604** | **.9613** | .9857 | .9881 | .9857 | .9730 | .9752 | .9744 | .9727 | .9729 | .9726 |

Table 4: Combating with adversarial attacks on Cora and Citeseer datasets.

| adversarial attacks | Cora | | | | Citeseer | | | |
|---|---|---|---|---|---|---|---|---|
| | clean | $\epsilon_{adv}=20\%$ | $\epsilon_{adv}=40\%$ | $\epsilon_{adv}=60\%$ | clean | $\epsilon_{adv}=20\%$ | $\epsilon_{adv}=40\%$ | $\epsilon_{adv}=60\%$ |
| Standard | .8686 | .7971 | .7671 | .7014 | .8317 | .8139 | .7736 | .7481 |
| RGIB-SSL | **.9260** | .8296 | **.8095** | **.8052** | **.9148** | **.8656** | **.8347** | **.8022** |
| RGIB-REP | .8758 | **.8408** | .7918 | .7611 | .8415 | .8382 | .8107 | .7893 |

**Remark 5.1.** *The two instantiations of RGIB can be generalized to different scenarios with their own priority according to the intrinsic graph properties. Basically, the RGIB-SSL is more adaptive to sparser graphs, e.g., Cora and Citeseer, where the edge noise results in greater performance degradation. The RGIB-REP can be more suitable for denser graphs, e.g., Facebook and Chameleon.*

**Combating with adversarial attacks.** Adversarial attacks on graphs can be generally divided into poisoning attacks that perturb the graph in training time and evasion attacks that perturb the graph in testing time. Here, we conduct the poisoning attacks based on Nettack [57] that only perturbs the graph structure. Notes that Nettack generates perturbations by modifying graph structure or node attributes such that perturbations maximally destroy downstream GNN's predictions. Here, we apply Nettack on Cora and Citeseer datasets as representatives. As shown in Tab. 4, the adversarial attack that adds noisy edges to the input graph also significantly degenerates the GNN's performance. And comparably, the brought damage is more severe than randomly added edges [57, 53, 42]. Crucially, we empirically justify that RGIB-SSL and RGIB-REP can also promote the robustness of GNN against adversarial attacks on the graph structure, remaining the adversarial robustness of the GIB.

**The distribution of learned representation.** We justify that the proposed methods can effectively alleviate the representation collapse. Compared with the standard training, both RGIB-REP and RGIB-SSL bring significant improvements to the alignment with much lower values, as in Tab. 5. At the same time, the uniformity of learned representation is also enhanced: it can be seen from Fig. 6 that the various query edges tend to be more uniformly distributed on the unit circle, especially for the negative edges. It shows that the distribution of edge representation is effectively recovered. Besides, as RGIB-SSL explicitly constrains the representation, its recovery power on representation distribution is naturally stronger than RGIB-REP, resulting in much better alignment and uniformity.

**Learning with clean data.** Here, we study how the robust methods behave when learning with clean data, *i.e.*, no edge noise exists. As shown in Tab. 7, the proposed two instantiations of RGIB can also boost the performance when learning on clean graphs, and outperforms other baselines in most cases.

Table 5: Comparison of alignment. Here, std. is short for *standard training*, and SSL/REP is short for RGIB-SSL/RGIB-REP, respectively.

| dataset | Cora | | | Citeseer | | |
|---|---|---|---|---|---|---|
| method | std. | REP | SSL | std. | REP | SSL |
| clean | .616 | .524 | **.475** | .445 | .439 | **.418** |
| $\varepsilon=20\%$ | .687 | .642 | **.543** | .586 | .533 | **.505** |
| $\varepsilon=40\%$ | .695 | .679 | **.578** | .689 | .623 | **.533** |
| $\varepsilon=60\%$ | .732 | .704 | **.615** | .696 | .647 | **.542** |

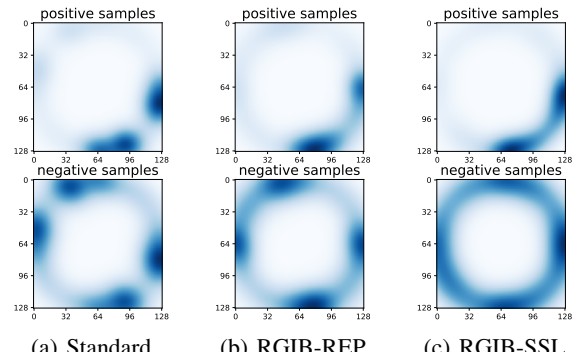

(a) Standard    (b) RGIB-REP    (c) RGIB-SSL

Figure 6: Uniformity distribution on Citeseer with $\varepsilon=40\%$.

Table 6: Comparison on different schedulers. SSL/REP are short for RGIB-SSL/RGIB-REP. Experiments are performed with a 4-layer GAT and $\epsilon=40\%$ mixed edge noise.

| dataset | Cora | | Citeseer | | Pubmed | |
|---|---|---|---|---|---|---|
| method | SSL | REP | SSL | REP | SSL | REP |
| $constant$ | .8398 | **.7927** | **.8227** | .7742 | .8596 | **.8416** |
| $linear(\cdot)$ | .8427 | .7653 | .8167 | .7559 | **.8645** | .8239 |
| $sin(\cdot)$ | **.8436** | .7924 | .8132 | .7680 | .8637 | .8275 |
| $cos(\cdot)$ | .8334 | .7833 | .8088 | .7647 | .8579 | .8372 |
| $exp(\cdot)$ | .8381 | .7815 | .8085 | .7569 | .8617 | .8177 |

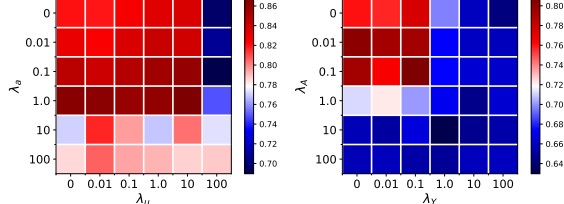

Figure 7: Grid search of hyper-parameter with RGIB-SSL (left) and RGIB-REP (right) on Cora dataset with bilateral noise $\epsilon = 40\%$. As can be seen, neither too large nor too small value can bring a good solution.

Table 7: Method comparison with a 4-layer GCN trained on the clean data.

| method | Cora | Citeseer | Pubmed | Facebook | Chameleon | Squirrel |
|---|---|---|---|---|---|---|
| Standard | .8686 | .8317 | .9178 | .9870 | .9788 | **.9725** |
| DropEdge | .8684 | .8344 | .9344 | .9869 | .9700 | .9629 |
| NeuralSparse | .8715 | .8405 | .9366 | .9865 | **.9803** | .9635 |
| PTDNet | .8577 | .8398 | .9315 | .9868 | .9696 | .9640 |
| Co-teaching | .8684 | .8387 | .9192 | .9771 | .9698 | .9626 |
| Peer loss | .8313 | .7742 | .9085 | .8951 | .9374 | .9422 |
| Jaccard | .8413 | .8005 | .8831 | .9792 | .9703 | .9610 |
| GIB | .8582 | .8327 | .9019 | .9691 | .9628 | .9635 |
| SupCon | .8529 | .8003 | .9131 | .9692 | .9717 | .9619 |
| GRACE | .8329 | .8236 | .9358 | .8953 | .8999 | .9165 |
| **RGIB-REP** | .8758 | .8415 | .9408 | **.9875** | .9792 | .9680 |
| **RGIB-SSL** | **.9260** | **.9148** | **.9593** | .9845 | .9740 | .9646 |

**Optimization schedulers.** To reduce the search cost of coefficients in the objectives of RGIB-SSL and RGIB-REP, we set a unified optimization framework that can be formed as $\mathcal{L} = \alpha\mathcal{L}_{cls} + (1-\alpha)\mathcal{R}_1 + (1-\alpha)\mathcal{R}_2$. Here, we attempt 5 different schedulers to tune the only hyper-parameter $\alpha \in [0, 1]$, including (1) constant, $\alpha \equiv c$; (2) linear, $\alpha_t = k \cdot t$, where $t$ is the normalized time step; (3) sine, $\alpha_t = \sin(t \cdot \pi/2)$; (4) cosine, $\alpha_t = \cos(t \cdot \pi/2)$; and (5) exponential, $\alpha_t = e^{k \cdot t}$. As empirical results summarized in Tab. 6, the selection of optimization schedulers greatly affects the final results. Although there is no golden scheduler to always perform the best, the *constant* and *sine* schedulers are generally better than others among the 5 above candidates. A further hyper-parameter study with grid search of the multiple $\lambda$ in Eq. 3 of RGIB-SSL and Eq. 4 of RGIB-REP are illustrated in Fig. 7.

## 5.2 Ablation study

**Hybrid augmentation.** As shown in Tab. 8, RGIB-SSL benefits from the hybrid augmentation algorithm that automatically generates graphs of high diversity for contrastive learning. *e.g.*, compared with the fixed augmentation, the hybrid manner brings a 3.0% average AUC promotion on Cora.

**Self-adversarial alignment loss.** Randomly-sampled pairs with hierarchical information to be contrasted and learned from. The proposed re-weighting technique further enhances high-quality pairs and decreases low-quality counterparts. It refines the signal and brings up to 2.1% promotion.

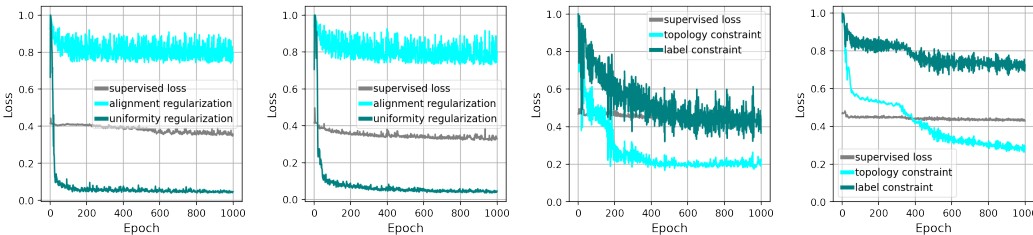

(a) RGIB-SSL on Cora  (b) RGIB-SSL on Citeseer  (c) RGIB-REP on Cora  (d) RGIB-REP on Citeseer
Figure 8: Learning curves of RGIB-SSL and RGIB-REP with $\varepsilon = 40\%$ bilateral noise.

Table 8: Ablation study for RGIB-SSL and RGIB-REP with a 4-layer SAGE. Here, $\epsilon = 60\%$ indicates the $60\%$ bilateral noise, while $\epsilon_a$ and $\epsilon_y$ represent ratios of unilateral input and label noise.

| variant | Cora | | | Chameleon | | |
|---|---|---|---|---|---|---|
| | $\epsilon = 60\%$ | $\epsilon_a = 60\%$ | $\epsilon_y = 60\%$ | $\epsilon = 60\%$ | $\epsilon_a = 60\%$ | $\epsilon_y = 60\%$ |
| RGIB-SSL (full) | .8596 | .8730 | .8994 | .9663 | .9758 | .9762 |
| - w/o hybrid augmentation | .8150 (5.1%↓) | .8604 (1.4%↓) | .8757 (2.6%↓) | .9528 (1.3%↓) | .9746 (0.1%↓) | .9695 (0.6%↓) |
| - w/o self-adversarial | .8410 (2.1%↓) | .8705 (0.2%↓) | .8927 (0.7%↓) | .9655 (0.1%↓) | .9732 (0.2%↓) | .9746 (0.1%↓) |
| - w/o supervision ($\lambda_s = 0$) | .7480 (12.9%↓) | .7810 (10.5%↓) | .7820 (13.0%↓) | .8626 (10.7%↓) | .8628 (11.5%↓) | .8512 (12.8%↓) |
| - w/o alignment ($\lambda_a = 0$) | .8194 (4.6%↓) | .8510 (2.5%↓) | .8461 (5.9%↓) | .9613 (0.5%↓) | .9749 (0.1%↓) | .9722 (0.4%↓) |
| - w/o uniformity ($\lambda_u = 0$) | .8355 (2.8%↓) | .8621 (1.2%↓) | .8878 (1.3%↓) | .9652 (0.1%↓) | .9710 (0.4%↓) | .9751 (0.1%↓) |
| RGIB-REP (full) | .7611 | .8487 | .8095 | .9567 | .9706 | .9676 |
| - w/o edge selection ($Z_A \equiv \tilde{A}$) | .7515 (1.2%↓) | .8199 (3.3%↓) | .7890 (2.5%↓) | .9554(0.1%↓) | .9704 (0.1%↓) | .9661 (0.1%↓) |
| - w/o label selection ($Z_Y \equiv \tilde{Y}$) | .7533 (1.0%↓) | .8373 (1.3%↓) | .7847 (3.0%↓) | .9484(0.8%↓) | .9666 (0.4%↓) | .9594 (0.8%↓) |
| - w/o topology constraint ($\lambda_A = 0$) | .7355 (3.3%↓) | .7699 (9.2%↓) | .7969 (1.5%↓) | .9503(0.6%↓) | .9658 (0.4%↓) | .9635 (0.4%↓) |
| - w/o label constraint ($\lambda_Y = 0$) | .7381 (3.0%↓) | .8106 (4.4%↓) | .8032 (0.7%↓) | .9443(1.2%↓) | .9665 (0.4%↓) | .9669 (0.1%↓) |

**Information constraints.** Label supervision contributes the most among the three informative terms, even though it contains label noise. As can be seen from Tab. 8, degenerating RGIB-SSL to a pure self-supervised manner without supervision (*i.e.*, $\lambda_s = 0$) leads to an average 11.9% drop in AUC.

**Edge / label selection.** The two selection methods adopted in RGIB-REP are nearly equally important. Wherein, the edge selection is more important for tackling the input noise, as a greater drop will come when removed; while the label selection plays a dominant role in handling the label noise.

**Topological / label constraint.** Tab. 8 also shows that the selection mechanisms should be regularized by the related constraints, otherwise sub-optimal cases with degenerated effects of denoising will be achieved. Besides, the topological constraint is comparably more sensitive than the label constraint.

**Learning curves.** We draw the learning curves with constant schedulers in Fig. 8, where the learning processes are generally stable across datasets. A detailed analysis with more cases is in Appendix E.1.

## 6 Conclusion

In this work, we study the problem of link prediction with the bilateral edge noise and reveal that the edge representation is severely collapsed under such a bilateral noise. Based on the observation, we introduce the Robust Graph Information Bottleneck (RGIB) principle, aiming to extract reliable signals via decoupling and balancing the mutual information among inputs, labels, and representation to enhance the robustness and avoid collapse. Regarding the instantiations of RGIB, the self-supervised learning technique and data reparameterization mechanism are utilized to establish the RGIB-SSL and RGIB-REP, respectively. Extensive studies on 6 common datasets and 3 popular GNNs verify the denoising effect of the proposed RGIB methods under different noisy scenarios.

## Acknowledgements

ZKZ and BH were supported by the NSFC Young Scientists Fund No. 62006202, NSFC General Program No. 62376235, Guangdong Basic and Applied Basic Research Foundation No. 2022A1515011652, Alibaba Innovative Research Program, HKBU Faculty Niche Research Areas No. RC-FNRA-IG/22-23/SCI/04, and HKBU CSD Departmental Incentive Scheme. JCY was supported by the National Key R&D Program of China (No. 2022ZD0160703), 111 plan (No. BP0719010), and the National Natural Science Foundation of China (No. 62306178). QMY was in part sponsored by the National Natural Science Foundation of China (No. 92270106).

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

# Appendix

# A Theoretical justification

## A.1 Notations

As an expansion of the notations in Sec. 2, we summarize the frequently used notations in Tab. 9.

Table 9: The most frequently used notations in this paper.

| notations | meanings |
|---|---|
| $\mathcal{V} = \{v_i\}_{i=1}^N$ | the set of nodes |
| $\mathcal{E} = \{e_{ij}\}_{ij=1}^M$ | the set of edges |
| $A \in \{0,1\}^{N \times N}$ | the adjacent matrix with binary elements |
| $X \in \mathbb{R}^{N \times D}$ | the node features |
| $\mathcal{G} = (A, X)$ | a graph (the observation input of a GNN) |
| $\mathcal{E}_{query} = \{e_{ij}\}_{ij=1}^K$ | the query edges to be predicted |
| $Y$ | the labels of the query edges |
| $\boldsymbol{h}_{ij}$ | representation of a query edge $e_{ij}$ |
| $\boldsymbol{H}$ | representation of all the query edges |
| $\boldsymbol{u}_i$ | representation of node $v_i$ |
| $\boldsymbol{U}$ | representation of all nodes |
| $I(X;Y)$ | the mutual information of $X$ and $Y$ |
| $I(X;Y|Z)$ | the conditional mutual information of $X$ and $Y$ when observing $Z$ |

## A.2 Preliminaries for mutual information

**Definition A.1** (Informational Divergence). *The informational divergence (also called relative entropy or Kullback-Leibler distance) between two probability distributions $p$ and $q$ on a finite space $\mathcal{X}$ (i.e., a common alphabet) is defined as*

$$D(p||q) = \sum_{x \in \mathcal{X}} p(x) \log \frac{p(x)}{q(x)} = \mathbb{E}_p\left[\log \frac{p(X)}{q(X)}\right]. \tag{5}$$

**Remark A.1.** *$D(p||q)$ measures the distance between $p$ and $q$. However, $D(\cdot||\cdot))$ is not a true metric, and it does not satisfy the triangular inequality. $D(p||q)$ is non-negative and zero if and only if $p = q$.*

**Definition A.2** (Mutual Information). *Given two discrete random variables $X$ and $Y$, the mutual information (MI) $I(X;Y)$ is the relative entropy between the joint distribution $p(x,y)$ and the product of the marginal distributions $p(x)p(y)$, namely,*

$$\begin{aligned} I(X;Y) &= D(p(x,y)||p(x)p(y)) \\ &= \sum_{x \in X, y \in Y} p(x,y) \log\left(\frac{p(x,y)}{p(x)p(y)}\right) \\ &= \sum_{x \in X, y \in Y} p(x,y) \log\left(\frac{p(x|y)}{p(x)}\right). \end{aligned} \tag{6}$$

**Remark A.2.** *$I(X;Y)$ is symmetrical in $X$ and $Y$, i.e., $I(X;Y) = H(X) - H(X|Y) = H(Y) - H(Y|X) = I(Y;X)$.*

**Proposition A.3** (Chain Rule for Entropy). *$H(X_1, X_2, \cdots, X_n) = \sum_{i=1}^n H(X_i|X_1, X_2, \cdots, X_{i-1})$.*

**Proposition A.4** (Chain Rule for Conditional Entropy). *$H(X_1, X_2, \cdots, X_n|Y) = \sum_{i=1}^n H(X_i|X_1, X_2, \cdots, X_{i-1}, Y)$.*

**Proposition A.5** (Chain Rule for Mutual Information). $I(X_1, X_2, \cdots, X_n; Y) = \sum_{i=1}^{n} I(X_i; Y | X_1, X_2, \cdots, X_{i-1})$.

**Corollary A.6.** $\forall A, Z_i, Z_j, I(A; Z_i, Z_j) \geq \max \big( I(A; Z_i), I(A; Z_j) \big)$.

*Proof.* As $I(A; Z_i | Z_j) \geq 0$, $I(A; Z_i, Z_j) = I(A; Z_i) + I(A; Z_i | Z_j) \geq I(A; Z_i)$. Similarly, $I(A; Z_i, Z_j) \geq I(A; Z_j)$ can be obtained. Thus, we have $I(A; Z_i, Z_j) \geq \max \big( I(A; Z_i), I(A; Z_j) \big)$. □

**Proposition A.7** (Chain Rule for Conditional Mutual Information). $I(X_1, \cdots, X_n; Y | Z) = \sum_{i=1}^{n} I(X_i; Y | X_1, \cdots, X_{i-1}, Z)$.

### A.3 Proof for Proposition 4.2

*Proof.* As introduced in Sec. 4.2, the uniformity loss on two augmented graphs is formed as $\mathcal{R}_{unif} = \sum_{ij,mn}^{K} e^{-\|\boldsymbol{h}_{ij}^1 - \boldsymbol{h}_{mn}^1\|_2^2} + e^{-\|\boldsymbol{h}_{ij}^2 - \boldsymbol{h}_{mn}^2\|_2^2}$. For simplicity, we can reduce it to one graph, *i.e.*, $\mathcal{R}_{unif} = \mathbb{E}_{ij,mn \sim p_{\mathcal{E}}}[e^{-\|\boldsymbol{h}_{ij} - \boldsymbol{h}_{mn}\|_2^2}]$. Next, we obtain its upper bound by the following derivation.

$$
\begin{aligned}
\log \mathcal{R}_{unif} &= \log \mathbb{E}_{ij,mn \sim p_{\mathcal{E}}}[e^{-\|\boldsymbol{h}_{ij} - \boldsymbol{h}_{mn}\|_2^2}] \\
&= \log \mathbb{E}_{ij,mn \sim p_{\mathcal{E}}}[e^{2t \cdot \boldsymbol{h}_{ij}^T \boldsymbol{h}_{mn} - 2t}] \\
&\leq \mathbb{E}_{ij \sim p_{\mathcal{E}}}\big[ \log \mathbb{E}_{mn \sim p_{\mathcal{E}}}[e^{2t \cdot \boldsymbol{h}_{ij}^T \boldsymbol{h}_{mn}}] \big] \\
&= \frac{1}{N} \sum_{ij}^{N} \big[ \log \sum_{mn}^{N} [e^{2t \cdot \boldsymbol{h}_{ij}^T \boldsymbol{h}_{mn}}] \big] \\
&= \frac{1}{N} \sum_{ij}^{N} \log \hat{p}_{\text{vMF-KDE}}(\boldsymbol{h}_{ij}) + \log Z_{\text{vMF}} \\
&\triangleq -\hat{H}(\boldsymbol{H}) + \log Z_{\text{vMF}}.
\end{aligned}
\tag{7}
$$

Here, $\hat{p}_{\text{vMF-KDE}}(\cdot)$ is a von Mises-Fisher (vMF) kernel density estimation based on $N$ training samples, and $Z_{\text{vMF}}$ is the vMF normalization constant. $\hat{H}(\cdot)$ is the resubstitution entropy estimator of $\boldsymbol{H} = f_{\boldsymbol{w}}(\cdot)$ [1]. As the uniformity loss $\mathcal{L}_{unif}$ can be approximated by entropy $\hat{H}(\boldsymbol{H})$, a higher entropy $H(\boldsymbol{H})$ indicates a lower uniformity loss, *i.e.*, a higher uniformity. □

### A.4 Proof for Proposition 4.3

*Proof.* Recall the hybrid graph augmentation technique is adopted by RGIB-SSL for contrasting, we can approximate the expectation $\mathbb{E}[I(\boldsymbol{H}; \tilde{A} | \tilde{Y})]$ by $^1/_N \sum_{i=1}^{N} I(\boldsymbol{H}_i; \tilde{A}_i | \tilde{Y})$ with $N$ augmented graphs. Based on this, we drive the upper bound of $I(\boldsymbol{H}; \tilde{A} | \tilde{Y})$ as follows.

$$
\begin{aligned}
I(\boldsymbol{H}; \tilde{A} | \tilde{Y}) \leq I(\boldsymbol{H}; \tilde{A}) &= ^1/_N \sum_{i=1}^{N} \big( I(\boldsymbol{H}_i; \tilde{A}_i) \big) \\
&\approx ^1/_2 \big( I(\boldsymbol{H}_1; \tilde{A}_1) + I(\boldsymbol{H}_2; \tilde{A}_2) \big) \\
&= ^1/_2 \big( H(\boldsymbol{H}_1) + H(\tilde{A}_1) + H(\boldsymbol{H}_2) + H(\tilde{A}_2) - H(\boldsymbol{H}_1, \tilde{A}_1) - H(\boldsymbol{H}_2, \tilde{A}_2) \big) \\
&\leq ^1/_2 \big( H(\boldsymbol{H}_1) + H(\tilde{A}_1) + H(\boldsymbol{H}_2) + H(\tilde{A}_2) - H(\boldsymbol{H}_1, \boldsymbol{H}_2) - H(\tilde{A}_1, \tilde{A}_2) \big) \\
&= ^1/_2 \big( I(\boldsymbol{H}_1; \boldsymbol{H}_2) + I(\tilde{A}_1; \tilde{A}_2) \big) \\
&= ^1/_2 \big( I(\boldsymbol{H}_1; \boldsymbol{H}_2) + c \big).
\end{aligned}
\tag{8}
$$

According to the definition of interaction information w.r.t three variables, we know that $I(X; Y; Z) = I(X; Y) - I(X; Y | Z) \geq 0$ under the condition of Markov chain $X \rightarrow Z \rightarrow Y$. Thus, we can obtain $I(X; Y) \geq I(X; Y | Z)$.

Specifically, the first inequality, *i.e.*, $\frac{1}{N}\sum_{i=1}^{N}I(\boldsymbol{H}_i;\tilde{A}_i) \approx \frac{1}{2}(I(\boldsymbol{H}_1;\tilde{A}_1) + I(\boldsymbol{H}_2;\tilde{A}_2))$, is to approximate the expectation of $N$ views by 2 representative views in practice. As these two views are generated by random and unbias perturbation in each training iteration, the approximation error induced here is expected to be reduced.

As for the second inequality, *i.e.*,

$$\frac{1}{2}(H(\boldsymbol{H}_1) + H(\tilde{A}_1) + H(\boldsymbol{H}_2) - H(\boldsymbol{H}_1,\tilde{A}_1) - H(\boldsymbol{H}_2,\tilde{A}_2))$$
$$\leq \frac{1}{2}(H(\boldsymbol{H}_1) + H(\tilde{A}_1) + H(\boldsymbol{H}_2) - H(\boldsymbol{H}_1,\boldsymbol{H}_2) - H(\tilde{A}_1,\tilde{A}_2)), \tag{9}$$

is due to $H(\boldsymbol{H}_1,\tilde{A}_1) + H(\boldsymbol{H}_2,\tilde{A}_2) \geq H(\boldsymbol{H}_1,\boldsymbol{H}_2) + H(\tilde{A}_1,\tilde{A}_2)$.

Besides, $I(\boldsymbol{H}_1;\boldsymbol{H}_2) \geq I(\boldsymbol{H}_1;\tilde{A}_1)$ and $I(\tilde{A}_1;\tilde{A}_2) \geq I(\boldsymbol{H}_1;\tilde{A}_1)$.

That is, the informative variables located in the same representation space are naturally more correlated.

Then, we have

$$H(\boldsymbol{H}_1,\tilde{A}_1) + H(\boldsymbol{H}_2,\tilde{A}_2)$$
$$= H(\boldsymbol{H}_1) + H(\boldsymbol{H}_2) + H(\tilde{A}_1) + H(\tilde{A}_2) - I(\boldsymbol{H}_1;\tilde{A}_1) - I(\boldsymbol{H}_2;\tilde{A}_2)$$
$$\geq H(\boldsymbol{H}_1) + H(\boldsymbol{H}_2) + H(\tilde{A}_1) + H(\tilde{A}_2) - I(\boldsymbol{H}_1;\boldsymbol{H}_2) - I(\tilde{A}_1;\tilde{A}_2) \tag{10}$$
$$= H(\boldsymbol{H}_1,\boldsymbol{H}_2) + H(\tilde{A}_1,\tilde{A}_2).$$

Thus, we can obtain $H(\boldsymbol{H}_1,\tilde{A}_1) + H(\boldsymbol{H}_2,\tilde{A}_2) \geq H(\boldsymbol{H}_1,\boldsymbol{H}_2) + H(\tilde{A}_1,\tilde{A}_2)$. And finally, a lower $I(\boldsymbol{H}_1;\boldsymbol{H}_2)$ can upper bounded as lower $I(\boldsymbol{H};\tilde{A})$ and $I(\boldsymbol{H};\tilde{A}|\tilde{Y})$. □

## A.5 Proof for Proposition 4.4

*Proof.* From the perspective of data processing, the collected noisy edges in the training set can be randomly split into the observed graph (*i.e.*, the input of GNN) or the predictive graph (*i.e.*, the query edges for the GNN to predict). Considering the noisy edges might come from similar sources, *e.g.*, biases from human annotation, the corresponding noise patterns can also be similar between these two kinds of noise that are naturally coupled together to some extent. We treat the label constraint and topology constraint in the same way here, without injecting any additional heuristics or assumptions. With the marginal distribution $\mathbb{Q}(\boldsymbol{Z}_A) = \sum_n \mathbb{P}_\phi(\boldsymbol{P}|n)\mathbb{P}(\tilde{A}=n) = \mathbb{P}(n)\prod_{\tilde{A}_{ij}=1}^{n}\boldsymbol{P}_{ij}$, we drive the upper bound of MI $I(\boldsymbol{Z}_A;\tilde{A})$ as:

$$I(\boldsymbol{Z}_A;\tilde{A}) = \mathbb{E}_{\boldsymbol{Z}_A,\tilde{A}}[\log(\frac{\mathbb{P}(\boldsymbol{Z}_A|\tilde{A})}{\mathbb{P}(\boldsymbol{Z}_A)})]$$
$$= \mathbb{E}_{\boldsymbol{Z}_A,\tilde{A}}[\log(\frac{\mathbb{P}_\phi(\boldsymbol{Z}_A|\tilde{A})}{\mathbb{Q}(\boldsymbol{Z}_A)})] - \text{KL}(\mathbb{P}(\boldsymbol{Z}_A)||\mathbb{Q}(\boldsymbol{Z}_A))$$
$$\leq \mathbb{E}[\text{KL}(\mathbb{P}_\phi(\boldsymbol{Z}_A|A)||\mathbb{Q}(\boldsymbol{Z}_A))] \tag{11}$$
$$= \sum_{e_{ij}\in\tilde{A}} \boldsymbol{P}_{ij}\log\frac{\boldsymbol{P}_{ij}}{\tau} + (1-\boldsymbol{P}_{ij})\log\frac{1-\boldsymbol{P}_{ij}}{1-\tau} = \mathcal{R}_A,$$

where the KL divergence on two given distribution $\mathbb{P}(x)$ and $\mathbb{Q}(x)$ is defined as $\text{KL}(\mathbb{P}(x)||\mathbb{Q}(x)) = \sum_x \mathbb{P}(x)\log \mathbb{P}(x)/\mathbb{Q}(x)$. Thus, we obtain the upper bound of $I(\boldsymbol{Z}_A;\tilde{A})$ as Eq. 11. Similarly, the label constraint is bounded as $I(\boldsymbol{Z}_Y;\tilde{Y}) \leq \sum_{e_{ij}\in\tilde{Y}} \boldsymbol{P}_{ij}\log\frac{\boldsymbol{P}_{ij}}{\tau} + (1-\boldsymbol{P}_{ij})\log\frac{1-\boldsymbol{P}_{ij}}{1-\tau} = \mathcal{R}_Y$. □

## A.6 Proof for Proposition 4.5

*Proof.* We derive the lower bound of $I(\boldsymbol{H}; \boldsymbol{Z}_Y)$ as follows.

$$
\begin{aligned}
I(\boldsymbol{H}; \boldsymbol{Z}_Y) =& I(f_{\boldsymbol{w}}(\boldsymbol{Z}_A); \boldsymbol{Z}_Y) \\
\geq& \mathbb{E}_{\boldsymbol{Z}_Y, \boldsymbol{Z}_A}[\log \frac{\mathbb{P}_{\boldsymbol{w}}(\boldsymbol{Z}_Y | \boldsymbol{Z}_A)}{\mathbb{P}(\boldsymbol{Z}_Y)}] \\
=& \mathbb{E}_{\boldsymbol{Z}_Y, \boldsymbol{Z}_A}[\log \mathbb{P}_{\boldsymbol{w}}(\boldsymbol{Z}_Y | \boldsymbol{Z}_A) - \log(\mathbb{P}(\boldsymbol{Z}_Y))] \\
=& \mathbb{E}_{\boldsymbol{Z}_Y, \boldsymbol{Z}_A}[\log \mathbb{P}_{\boldsymbol{w}}(\boldsymbol{Z}_Y | \boldsymbol{Z}_A) + H(\boldsymbol{Z}_Y)] \\
\geq& \mathbb{E}_{\boldsymbol{Z}_Y, \boldsymbol{Z}_A}[\log \mathbb{P}_{\boldsymbol{w}}(\boldsymbol{Z}_Y | \boldsymbol{Z}_A)] \\
\approx& -\frac{1}{|\boldsymbol{Z}_Y|} \sum_{e_{ij} \in \boldsymbol{Z}_Y} \mathcal{L}_{cls}(f_{\boldsymbol{w}}(\boldsymbol{Z}_A), \boldsymbol{Z}_Y).
\end{aligned}
\tag{12}
$$

Thus, the MI $I(\boldsymbol{H}; \boldsymbol{Z}_Y)$ can be lower bounded by the classification loss, and $\min -\lambda_s I(\boldsymbol{H}; \boldsymbol{Z}_Y)$ in RGIB-REP (Eq. 4) is upper bounded by $\min \lambda_s/|\boldsymbol{Z}_Y| \sum_{e_{ij} \in \boldsymbol{Z}_Y} \mathcal{L}_{cls}(f_{\boldsymbol{w}}(\boldsymbol{Z}_A), \boldsymbol{Z}_Y)$ as Eq. 12. $\qquad\square$

## A.7 Proof for Theorem 4.6

*Proof.* From the theoretical perspective, RGIB-SSL explicitly optimizes the representation $\boldsymbol{H}$ with self-supervised regularizations, *i.e.*, alignment $I(\boldsymbol{H}_1; H_2)$ and uniformity $H(\boldsymbol{H}_1), H(\boldsymbol{H}_2)$. By contrast, RGIB-REP implicitly optimizes $\boldsymbol{H}$ by purifying the noisy $\tilde{A}$ and $\tilde{Y}$ with the reparameterization mechanism to extract clean signals in the forms of latent variables $\boldsymbol{Z}_Y$ and $\boldsymbol{Z}_A$. The information constraints $I(\boldsymbol{Z}_A; \tilde{A}), I(\boldsymbol{Z}_Y; \tilde{Y})$ are directly acting on $\boldsymbol{Z}_Y$ and $\boldsymbol{Z}_A$ and indirectly regularizing the representation $\boldsymbol{H}$. The $\boldsymbol{Z}_Y^*$ and $\boldsymbol{Z}_A^*$ are the optimal solutions of RGIB-REP in Eq. 4, and that is the reason of clean labels are expected to be recovered, *i.e.*, $\boldsymbol{Z}_Y^* = Y$.

Then, with the relaxation via parametrization in A.6, we first relax the standard RGIB-REP to its upper bound as follows.

$$
\begin{aligned}
\min \text{RGIB-REP} \triangleq& -\lambda_s I(\boldsymbol{H}; \boldsymbol{Z}_Y) + \lambda_A I(\boldsymbol{Z}_A; \tilde{A}) + \lambda_Y I(\boldsymbol{Z}_Y; \tilde{Y}) \\
\leq& -\lambda_s I(\boldsymbol{Z}_A; \boldsymbol{Z}_Y) + \lambda_A I(\boldsymbol{Z}_A; \tilde{A}) + \lambda_Y I(\boldsymbol{Z}_Y; \tilde{Y}).
\end{aligned}
\tag{13}
$$

As $\boldsymbol{Z}_Y^* \approx Y$, Eq. 13 can be reduced to $\min -\lambda_s I(\boldsymbol{Z}_A; Y) + \lambda_A I(\boldsymbol{Z}_A; \tilde{A}) + \lambda_Y H(Y)$, where $I(Y; \tilde{Y}) = H(Y)$ as $Y \subseteq \tilde{Y}$. Removing the final term with constant $H(Y)$, it can be further reduced to $\min -I(\boldsymbol{Z}_A; Y) + \lambda I(\boldsymbol{Z}_A; \tilde{A})$, where $\lambda = \lambda_A/\lambda_s$. Since to minimize the $-I(\boldsymbol{Z}_A; Y) + \lambda I(\boldsymbol{Z}_A; \tilde{A})$ is equal to maximize the $I(\boldsymbol{Z}_A; Y) - \lambda I(\boldsymbol{Z}_A; \tilde{A})$, next, we conduct the following derivation:

$$
\begin{aligned}
& \max I(\boldsymbol{Z}_A; Y) - \lambda I(\boldsymbol{Z}_A; \tilde{A}) \\
=& \max \big(I(Y; \boldsymbol{Z}_A, \tilde{A}) - I(\tilde{A}; Y | \boldsymbol{Z}_A)\big) - \lambda I(\boldsymbol{Z}_A; \tilde{A}) \\
=& \max I(Y; \boldsymbol{Z}_A, \tilde{A}) - I(\tilde{A}; Y | \boldsymbol{Z}_A) - \lambda \big(I(\boldsymbol{Z}_A; \tilde{A}, Y) - I(\tilde{A}; Y | \boldsymbol{Z}_A)\big) \\
=& \max I(Y; \boldsymbol{Z}_A, \tilde{A}) - (1 - \lambda) I(\tilde{A}; Y | \boldsymbol{Z}_A) - \lambda I(\boldsymbol{Z}_A; \tilde{A}, Y) \\
=& \max I(Y; \tilde{A}) - (1 - \lambda) I(\tilde{A}; Y | \boldsymbol{Z}_A) - \lambda I(\boldsymbol{Z}_A; \tilde{A}, Y) \\
=& \max(1 - \lambda) I(\tilde{A}; Y) - (1 - \lambda) I(\tilde{A}; Y | \boldsymbol{Z}_A) - \lambda I(\boldsymbol{Z}_A; \tilde{A} | Y) \\
=& \max(1 - \lambda) c - (1 - \lambda) I(\tilde{A}; Y | \boldsymbol{Z}_A) - \lambda I(\boldsymbol{Z}_A; \tilde{A} | Y) \\
=& (1 - \lambda) c.
\end{aligned}
\tag{14}
$$

Since MI $I(\tilde{A}; Y | \boldsymbol{Z}_A) \geq 0$ and $I(\boldsymbol{Z}_A; \tilde{A} | Y) \geq 0$ are always true, the optimal $\boldsymbol{Z}_A^*$ should makes $-(1 - \lambda) I(\tilde{A}; Y | \boldsymbol{Z}_A) - \lambda I(\boldsymbol{Z}_A; \tilde{A} | Y) = 0$ to reach the optimal case. Thus, it should satisfy $I(\tilde{A}; Y | \boldsymbol{Z}_A) = 0$ and $I(\boldsymbol{Z}_A; \tilde{A} | Y) = 0$ simultaneously. Therefore, $\boldsymbol{Z}_A = \boldsymbol{Z}_A^*$ maximizes $I(\boldsymbol{Z}_A; Y) - \lambda I(\boldsymbol{Z}_A; \tilde{A})$, which is equal to minimize $-I(\boldsymbol{Z}_A; Y) + \lambda I(\boldsymbol{Z}_A; \tilde{A})$, *i.e.*, the RGIB-REP. Symmetrically, when $\boldsymbol{Z}_A \equiv A$, $\boldsymbol{Z}_Y = \boldsymbol{Z}_Y^*$ maximizes $I(A; \boldsymbol{Z}_Y) - \lambda I(\boldsymbol{Z}_Y; \tilde{Y})$ as $I(A; \boldsymbol{Z}_Y) - \lambda I(\boldsymbol{Z}_Y; \tilde{Y}) = (1 - \lambda) I(\tilde{Y}; A) - (1 - \lambda) I(\tilde{Y}; A | \boldsymbol{Z}_Y) - \lambda I(\boldsymbol{Z}_Y; \tilde{Y} | A)$. $\qquad\square$

# B  Further discussions

## B.1   Definition of the bilateral edge noise

Basically, the edge noise can exist in the form of *false positive* edges or *false negative* noise. Specifically, the false positive edges are treated as existing edges with label 1, but in fact, such edges should not exist. This is the standard noisy label learning problem. Conversely, the false negative edges are about the labels of the non-observed edges, which is usually considered as hard negative mining in GNNs. The predictive probabilities of such edges will be minimized. In terms of dropping edges, it actually will not inject noise into the graph data but only decrease the positive observations, which is beyond the research focus.

Here, we would highlight that our work focuses on the *false positive* edges as it is more practical and common in real-world scenarios since the data annotating procedure can produce such a kind of noise [37]. Thus, if the bilateral edge noise exists, it is more likely to be false positive samples, while the false negative samples are often intractable to be collected and annotated in practice. Actually, investigating the influences of the false negative samples is another line of research, such as [41, 19], which is orthogonal and complementary to our work. Learning from a clean graph can also encounter the problem of false negative samples, which is usually due to the random sampling of negative nodes/edges. Note that tackling the false negative samples is a well-known problem in the area of link prediction while handling the false positive samples is also valuable but still remains under-explored.

The input noise and label noise do look quite similar and unilateral. From the perspective of data processing, the collected noisy edges in the training set can be randomly split into the observed graph (*i.e.*, the input of GNN) or the predictive graph (*i.e.*, the query edges for the GNN to predict). Considering the noisy edges might come from similar sources, *e.g.*, biases from human annotation, the corresponding noise patterns can also be similar between these two kinds of noise that are naturally coupled together to some extent. However, we would claim and justify that *the two kinds of noise are fundamentally different.*

Although both noises can bring severe degradation to empirical performances, they actually act in different places from the model perspective. As the learnable weights are updated as $w := w - \eta \nabla_w \mathcal{L}(\boldsymbol{H}, \tilde{Y})$, the label noise $\tilde{Y}$ acting on the backward propagation can directly influence the model. By contrast, the input noise indirectly acts on the learned weights as it appears in the front end of the forward inference, *i.e.*, $\boldsymbol{H} = f_w(A, X)$. Empirically, as results shown in Tab. 2 and Tab. 3, the standard GNN (without any defenses) performs quite differently under the same proportion of input noise and label noise. Such a phenomenon can inspire one to understand the intrinsic denoising mechanism or memorization effects of the GNN, and we would leave that as further work.

More importantly, from the perspective of defending, it could be easy and trivial to defend if these two kinds of noise are the same. However, none of the existing robust methods can effectively defend such a bilateral noise. As can be seen from Table 2, only marginal improvements are achieved when applying the existing robust methods to the bilateral noise. While in Tab. 2 and Tab. 3, these robust methods work effectively in handling the unilateral noise, *i.e.*, only the structure noise or label noise exists. The reason is that the properties of bilateral noise are much more complex than unilateral noise. Both sides of the information sources, *i.e.*, $\tilde{A}$, and $\tilde{Y}$, should be considered noisy, based on which the defending mechanism could be devised.

As pointed out by recent works [11, 37], graph noise can be produced in the data generation process that requires manually annotating the data. As in [37], in general, graph noise here refers to irreducible noises in graph structures, attributes, and labels. Thus, the studied bilateral edge noise in our work is in line with the generic concept of graph noise in the literature, which is recognized as a common problem in practice but also an underexplored challenge in both academic and industrial scenes.

In this work, we focus on the edge noise on adjacency $A$ but not the node feature $X$. Specifically, we focus on the investigation of the false positive edges as the bilateral noise and design the simulation approach in Definition 3.1 as it is closer to real-world scenarios. Therefore, in the optimization objective of the proposed RGIB principle, we assume that $X$ is generally clean and attaches more importance to the adjacency $A$. Besides, the presentation $\boldsymbol{H}$ is encoded from the node feature $X$. Thus, regularizing $\boldsymbol{H}$ in RGIB objectives can indirectly balance its dependence on $X$.

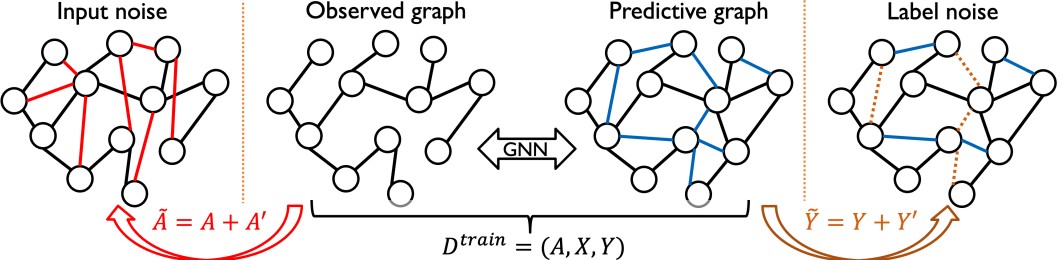

Figure 9: Illustration of input noise and label noise decomposed from the edge noise. The GNN predictor takes the observed graph as input, where the black edges are known, then predicts the latent edges (blue ones). Under the bilateral edge noise, the original training data $D^{\text{train}} = (\boldsymbol{A}, \boldsymbol{X}, \boldsymbol{Y})$ can be (i) added with noisy edges as input ($\boldsymbol{A} \to \tilde{\boldsymbol{A}}$) and (ii) perturbed by noisy supervision ($\boldsymbol{Y} \to \tilde{\boldsymbol{Y}}$).

## B.2 The difference with GIB and RGIB

In short, RGIB generalizes the GIB with improvements in both theories and methodologies to learn a robust representation more resistant to the bilateral edge noise.

Conceptually, we would clarify that GIB is intrinsically susceptive to label noise since it entirely preserves the label supervision with maximizing. Thus, GIB *cannot* provide an ideal solution to the inherent edge noise investigated in this work. Different from the GIB, the RGIB takes both noisy inputs and noisy labels into the design of its information bottleneck. RGIB decouples and balances the mutual dependence among graph topology, target labels, and representation, building new learning objectives toward robust representation.

Analytically, the GIB is intrinsically not robust to label noise since it entirely preserves the label supervision with maximizing $I(\boldsymbol{H}; \tilde{Y})$, as in Eq. 1. As illustrated in Fig. 2, the GIB decreases $I(\boldsymbol{H}; \tilde{A}|\tilde{Y})$ by directly constraining $I(\boldsymbol{H}; \tilde{A})$ to handle the input noise. Symmetrically, the label noise can be hidden in the area of $I(\boldsymbol{H}; \tilde{Y}|\tilde{A})$, but trivially constraining $I(\boldsymbol{H}; \tilde{Y})$ to regularize $I(\boldsymbol{H}; \tilde{Y}|\tilde{A})$ is not ideal, since it will conflict with Eq. 1.

Besides, it cannot tackle the noise within $I(\tilde{A}; \tilde{Y})$, where the two kinds of noise can share similar patterns as the random split manner does not change their distributions in expectation. GIB only indirectly regularizes the MI term $I(\boldsymbol{H}; \tilde{A}|\tilde{Y})$, as we introduce Sec. 4.1, which can only solve partial noisy information. By contrast, RGIB takes all the related MI terms, *i.e.*, $H(\boldsymbol{H})$, $I(\boldsymbol{H}; \tilde{Y}|\tilde{A})$ and $I(\boldsymbol{H}; \tilde{A}|\tilde{Y})$, based on which to provide a solution that balances these MI terms as a more strict information bottleneck.

Thus, it is crucial to further decouple the mutual dependence among $\tilde{A}$, $\tilde{Y}$, and $\boldsymbol{H}$. Regarding the representation $\boldsymbol{H}$, noise can exist in areas of $I(\boldsymbol{H}; \tilde{Y}|\tilde{A})$, $I(\boldsymbol{H}; \tilde{A}|\tilde{Y})$, and $I(\tilde{A}; \tilde{Y}|\boldsymbol{H})$, as shown in Fig. 2. As $I(\tilde{A}; \tilde{Y}|\boldsymbol{H}) = I(\tilde{A}; \tilde{Y}) + I(\boldsymbol{H}; \tilde{Y}|\tilde{A}) + I(\boldsymbol{H}; \tilde{A}|\tilde{Y}) - H(\boldsymbol{H}) + H(\boldsymbol{H}|\tilde{A}, \tilde{Y})$, where $\boldsymbol{I}(\tilde{A}; \tilde{Y})$ is a constant and redundancy $H(\boldsymbol{H}|\tilde{A}, \tilde{Y})$ can be regularized simultaneously. Therefore, the $I(\tilde{A}; \tilde{Y}|\boldsymbol{H})$ containing the shared noise patterns of $\tilde{A}$ and $\tilde{Y}$ can be approximated by the other three terms, *i.e.*, $H(\boldsymbol{H})$, $I(\boldsymbol{H}; \tilde{Y}|\tilde{A})$ and $I(\boldsymbol{H}; \tilde{A}|\tilde{Y})$. Since the two later terms are also with noise, a balance of these three informative terms can be a solution to the problem.

Based on the above analysis, we derive the RGIB principle in Def. 4.1 that balances the three important information terms $H(\boldsymbol{H})$, $I(\boldsymbol{H}; \tilde{Y}|\tilde{A})$ and $I(\boldsymbol{H}; \tilde{A}|\tilde{Y})$. It works as an information bottleneck to filter out the noisy signals in both $\tilde{A}$ and $\tilde{Y}$, utilizing the supervision $I(\boldsymbol{H}; \tilde{Y})$ at the same time.

From the view of methodology, we also provide two instantiations, RGIB-SSL and RGIB-REP, to approximate the aforementioned MI terms. These two instantiations benefit from different methodologies, *i.e.*, self-supervised learning and data reparameterization, for implicit and explicit data denoising, respectively. Note that these two methodologies are not considered in the original GIB. However, we show that they can be utilized to implement the specific RGIB instantiation and build effective information bottlenecks for handling the inherent noise.

Empirically, we justify the effectiveness of the two instantiations of RGIB that outperform GIB in several noisy scenarios. Besides, the GIB is highly coupled with the GAT. By contrast, RGIB does not require any modifications to GNN architecture. It can be seamlessly integrated with various existing GNNs and promote their robustness against inherent noise.

## B.3 The relationship between RGIB-SSL and RGIB-REP

Based on the empirical study of several datasets and GNNs, it is found that the two instantiations can be generalized to different scenarios. Generally, RGIB-SSL is more adaptive to sparser graphs, while RGIB-REP can be more suitable for denser graphs. More importantly, they can be complementary to each other with flexible options in practical applications, as summarized in Remark 5.1.

From the theoretical perspective, RGIB-SSL explicitly optimizes the representation $H$ with self-supervised regularizations, *i.e.*, alignment $I(H_1; H_2)$ and uniformity $H(H_1), H(H_2)$. By contrast, RGIB-REP implicitly optimizes $H$ by purifying the noisy $\tilde{A}$ and $\tilde{Y}$ with the reparameterization mechanism to extract clean signals in the forms of latent variables $Z_Y$ and $Z_A$. The information constraints $I(Z_A; \tilde{A}), I(Z_Y; \tilde{Y})$ are directly acting on $Z_Y$ and $Z_A$ and indirectly regularizing $H$.

Besides, from the perspective of methodology, both instantiations are equipped with adaptive designs for obtaining an effective information bottleneck. We also conduct a further comparison and analysis of the two instantiations that are summarized as follows.

Table 10: Comparison of RGIB-SSL and RGIB-REP.

| Instantiation | methodology | advantages | disadvantages |
|---|---|---|---|
| RGIB-SSL | self-supervised learning | automated graph augmentation; good effectiveness; can be applied in entirely self-supervised settings without labels. | with expensive calculation for the contrastive objectives, especially the uniformity; requires extra graph augmentation operations. |
| RGIB-REP | data reparame-terization | no need to do data augmentation; good efficiency; the input/output constraints do not require extra annotations for supervision and can be easily controlled. | sensitive to the hyper-parameters $\lambda$; less effective in extremely noisy cases; only applicable in fully supervised settings. |

More importantly, it is also possible that new instantiations based on other kinds of methodology are inspired by the RGIB principle. We believe that such a bidirectional IB that strictly treats the information signals on both the input and label side is helpful to noisy scenarios in practice. For example, one can generalize RGIB to the reasoning task on knowledge graphs [49, 56], where the multi-relational edges characterize more diverse and complex patterns.

## B.4 Assumption

The primary assumption of our work is that the edges of collected graph data can be potentially noisy. As in the submission, we present two ways of realizing the RGIB principle, which actually corresponds to some assumptions in building the objectives with the proper approximation.

RGIB-SSL assumes that the learned representation can be improved with higher uniformity and alignment. As RGIB-SSL directly acts on the graph representation, it is more suitable for recovering the distribution of representation, especially when encountering representation collapse due to the severe edge noise.

RGIB-REP explicitly purifies the input graph's topology and target labels with the jointly reparameterized $Z_A$ and $Z_Y$. Here, the latent variables $Z_A, Z_Y$ are expected to be more clean and informative than the noisy $\tilde{A}, \tilde{Y}$. RGIB-REP assumes that the GNN model can identify these latent variables and further benefit its learning procedure against noise.

In addition, any clean labels and adjacency are not used in training. On the contrary, the model directly learns from the noisy adjacency $\tilde{A}$ and label $\tilde{Y}$, which is practical as the collected data is potentially noisy in real-world applications. Besides, RGIB and all the baselines are run without any noise priors, *e.g.*, noise type or noise ratio.

Empirically, RGIB-SSL is more adaptive to sparser graphs, *e.g.*, Cora and Citeseer, where the edge noise results in severer representation collapse. RGIB-REP can be more suitable for denser graphs, *e.g.*, Facebook and Chameleon, where the latent variables of edge data are extracted by RGIB-REP. More importantly, the two RGIB instantiations be complementary to each other with flexible options in practical applications.

### B.5  Broader impact

In this work, we propose the RGIB framework for denoising the bilateral edge noise in graphs and for improving the robustness of link prediction. We conceptually derive the RGIB and empirically justify that it enables the various GNNs to learn the input-invariant and label-invariant edge representations, preventing representation collapse and obtaining superior performances against edge noise. Our objective is to protect and enhance the current graph models, and we do not think our work would have negative societal impacts.

Besides, A similar problem has been noticed in an early work [11], where robots tend to build connections with normal users to spread misinformation on social networks, yielding the degeneration of GNNs for robot detection. In addition, as pointed out by a recent survey [37], the bilateral noise can be produced in the data generation process that requires manually annotating the data. As introduced in [37], the bilateral noise here refers to irreducible noises in graph structures, attributes, and labels. In short, the studied edge noise in our work is in line with the generic concept of graph bilateral noise in the literature, which is recognized as a common problem in practice but also an under-explored challenge in both academic and industrial scenes.

However, several existing benchmarks, *e.g.*, Cora and Citeseer, are generally clean and without annotated noisy edges. Inevitably, there is a gap when one would like to study the influence of bilateral noise on these common benchmarks. Thus, to fill this gap, it is necessary to simulate the bilateral edge noise properly to investigate the impact of noise. If the edge noise exists, it should be false positive samples, while the false negative samples are often intractable to be collected. Thus, we focus on the investigation of the false positive edges as the noise and design the simulation approach in Def. 3.1 as it is closer to the real-world scenarios.

Besides, to our knowledge, we are the first to study the robustness problem of link prediction under the bilateral edge noise. One of our major contributions to this research problem is that we reveal that the bilateral noise can bring a severe representation collapse and performance degradation, and such negative impacts are general to common datasets and GNNs. What's more, it is also possible that new instantiations based on other kinds of methodology are inspired by the robust GIB principle. We believe that such a bidirectional information bottleneck strictly treating the information source on both the input side and label side is helpful in practice, especially for extremely noisy scenarios.

### B.6  General robustness of GNNs

Here, we provide a discussion of the robustness based on our understanding, which is in two folds.

**(1) Training-phase robustness.** Briefly, the training data is "hard" due to it can be perturbed with input or label noise or with imbalance problems, *e.g.*, long-tail sample distributions. By contrast, the testing data is comparably "easy" because it can be clean and well-balanced. Here, the robustness is usually evaluated by how the model trained on the "hard tasks" performs on "easy tasks", *i.e.*, the performance on testing data. Considering the "hard tasks" can be with different difficulties, *e.g.*, low or high noise ratios, the change in model performances is also a natural metric to quantify the robustness. A more robust model shall have lower performance changes while maintaining good performance simultaneously [13, 34].

In addition to test accuracy, the loss visualization can be utilized to establish quantitative metrics to evaluate the intrinsic robustness of the model, *e.g.*, the KL divergence between two predictions on an original input and a perturbed input [45]. It can provide more insights into the model compared with the aforementioned accuracy-based robustness estimation.

**(2) Testing-phase robustness.** Here, the training data is "easy" and the testing data can be "hard", *e.g.*, with perturbation generated by adversarial attacks, or encountering the out-of-distribution samples in testing time. The robustness is evaluated here for a trained model in the testing phase, where the property is that if a testing sample is "similar" to a training sample, *i.e.*, the in-distribution samples, then the testing error is close to the training error [51].

On the other hand, how this model performs on the out-of-distribution samples is usually hard to estimate, whether these samples are naturally collected or manually perturbed. Thus, measuring the performance of these "hard" testing samples, and be used to estimate the robustness of the model. An early work [40] derives generalization bounds for learning algorithms based on their robustness. And recently, out-of-distribution generalization on graphs has made great progress and attracted more attention from the research community [23]. Besides, adversarial training has been demonstrated to improve model robustness against adversarial attacks and out-of-distribution generalization ability.

## C    Implementation details

### C.1    GNNs for link prediction

We provide a detailed introduction forward propagation and backward update of GNNs in this part.

Formally, let $\ell = 1 \dots L$ denote the layer index, $h_i^\ell$ is the representation of the node $i$, MESS$(\cdot)$ is a learnable mapping function to transform the input feature, AGGREGATE$(\cdot)$ captures the 1-hop information from neighborhood $\mathcal{N}(v)$ in the graph, and COMBINE$(\cdot)$ is the final combination between neighbor features and the node itself. Then, the $l$-layer operation of GNNs can be formulated as $\boldsymbol{m}_v^\ell = \text{AGGREGATE}^\ell(\{\text{MESS}(\boldsymbol{h}_u^{\ell-1}, \boldsymbol{h}_v^{\ell-1}, e_{uv}) : u \in \mathcal{N}(v)\})$, where the representation of node $v$ is $\boldsymbol{h}_v^\ell = \text{COMBINE}^\ell(\boldsymbol{h}_v^{\ell-1}, \boldsymbol{m}_v^\ell)$. After $L$-layer propagation, the final node representations $\boldsymbol{h}_e^L$ of each $e \in V$ are obtained. Then, for each query edge $e_{ij} \in \mathcal{E}^{train}$ unseen from the input graph, the logit $\phi_{e_{ij}}$ is computed with the node representations $\boldsymbol{h}_i^L$ and $\boldsymbol{h}_j^L$ [3] with the readout function, *i.e.*, $\phi_{e_{ij}} = \text{READOUT}(\boldsymbol{h}_i^L, \boldsymbol{h}_j^L) \to \mathbb{R}$. Finally, the optimization objective can be defined as minimizing the binary cross-entropy loss, *i.e.*, $\min \mathcal{L}_{cls} = \sum_{e_{ij} \in \mathcal{E}^{train}} -y_{ij}\log(\sigma(\phi_{e_{ij}})) - (1-y_{ij})\log(1-\sigma(\phi_{e_{ij}}))$ where $\sigma(\cdot)$ is the sigmoid function, and $y_{ij} = 1$ for positive edges while $y_{ij} = 0$ for negative ones.

In addition, we summarize the detailed architectures of different GNNs in the following Tab. 11.

Table 11: Detailed architectures of different GNNs.

| GNN | MESS$(\cdot)$ & AGGREGATE$(\cdot)$ | COMBINE$(\cdot)$ | READOUT$(\cdot)$ |
|------|------|------|------|
| GCN | $\boldsymbol{m}_i^l = \boldsymbol{W}^l \sum_{j \in \mathcal{N}(i)} \frac{1}{\sqrt{\hat{d}_i \hat{d}_j}} \boldsymbol{h}_j^{l-1}$ | $\boldsymbol{h}_i^l = \sigma(\boldsymbol{m}_i^l + \boldsymbol{W}^l \frac{1}{\hat{d}_i} \boldsymbol{h}_i^{l-1})$ | $\phi_{e_{ij}} = \boldsymbol{h}_i^\top \boldsymbol{h}_j$ |
| GAT | $\boldsymbol{m}_i^l = \sum_{j \in \mathcal{N}(i)} \alpha_{ij} \boldsymbol{W}^l \boldsymbol{h}_j^{l-1}$ | $\boldsymbol{h}_i^l = \sigma(\boldsymbol{m}_i^l + \boldsymbol{W}^l \alpha_{ii} \boldsymbol{h}_i^{l-1})$ | $\phi_{e_{ij}} = \boldsymbol{h}_i^\top \boldsymbol{h}_j$ |
| SAGE | $\boldsymbol{m}_i^l = \boldsymbol{W}^l \frac{1}{|\mathcal{N}(i)|} \sum_{j \in \mathcal{N}(i)} \boldsymbol{h}_j^{l-1}$ | $\boldsymbol{h}_i^l = \sigma(\boldsymbol{m}_i^l + \boldsymbol{W}^l \boldsymbol{h}_i^{l-1})$ | $\phi_{e_{ij}} = \boldsymbol{h}_i^\top \boldsymbol{h}_j$ |

### C.2    Details of RGIB-SSL and RGIB-REP

As introduced in Sec. 4.2, the graph augmentation technique [43, 44] is adopted here to generate the perpetuated graphs of various views.

To avoid manually selecting and tuning the augmentation operations, we propose the hybrid graph augmentation method with the four augmentation operations as predefined candidates and the ranges of their corresponding hyper-parameters. The search space is summarized in Tab. 12, where the candidate's operations cover most augmentation approaches except for those operations modifying the number of nodes that are unsuitable for the link prediction task.

In each training iteration, two augmentation operators $\mathcal{T}^1(\cdot)$ and $\mathcal{T}^2(\cdot)$ and their hyper-parameters $\theta_1$ and $\theta_2$ are randomly sampled from the search space as elaborated in Algorithm. 1. The two operators will be performed on the observed graph $\mathcal{G}$, obtaining the two augmented graphs, namely,

---

[3]To avoid abusing notations, we use the $\boldsymbol{h}_i$ to stand for the final representation $\boldsymbol{h}_i^L$ in later contents.

$\mathcal{G}^1 = \mathcal{T}^1(\mathcal{G}|\theta_1)$ and $\mathcal{G}^2 = \mathcal{T}^2(\mathcal{G}|\theta_2)$. The edge representations are gained by $\boldsymbol{h}_{ij}^1 = f(\mathcal{G}^1, e_{ij}|\boldsymbol{w}) = \boldsymbol{h}_i^1 \odot \boldsymbol{h}_j^1$, where the node representations $\boldsymbol{h}_i^1$ and $\boldsymbol{h}_i^2$ are generated by the GNN model $f(\cdot|\boldsymbol{w})$ with learnable weights $\boldsymbol{w}$, and so it is for $\boldsymbol{h}_{ij}^2$ with $\mathcal{G}^2$.

Then, we learn the self-supervised edge representations by maximizing the edge-level agreement between the same query edge of different augmented graphs (positive pairs) and minimizing the agreement among different edges (negative pairs) with their representations. Note the $\boldsymbol{h}_{ij}$ here is the edge representation. Specifically, we minimize the representation similarity of the positive pairs $(\boldsymbol{h}_{ij}^1, \boldsymbol{h}_{ij}^2)$ and maximize the representation similarity of the randomly-sampled negative pairs $(\boldsymbol{h}_{ij}^1, \boldsymbol{h}_{mn}^2)$, where $e_{ij} \neq e_{mn}$.

Table 12: Search space $\mathcal{H}_T$ of the hybrid graph augmentation.

| Operator $\mathcal{T}_i(\cdot)$ | hyper-parameter $\theta_{T_i}$ | description |
|---|---|---|
| edge removing | $\theta_{er} \in (0.0, 0.5)$ | Randomly remove the observed edges with prob. $\theta_{er}$. |
| feature masking | $\theta_{fm} \in (0.0, 0.3)$ | Masking the columns of node features with prob. $\theta_{fm}$. |
| feature dropping | $\theta_{fd} \in (0.0, 0.3)$ | Dropping the elements of node features with prob. $\theta_{fd}$. |
| identity | NA. | Do not modify anything. |

---

**Algorithm 1** Hybrid graph augmentation.

---

**Require:** number of augmentation operators $n$;
1: initialize $\emptyset \to \mathcal{T}(\cdot)$
2: **for** $i = 1 \dots n$ **do**
3:     randomly selects one augmentation operator from the operator space $\mathcal{T}_i(\cdot) \in \mathcal{H}_T$;
4:     uniformly sample the corresponding hyper-parameter $\theta_i \sim U(\theta_{T_i})$;
5:     stores the newly sampled operator $\mathcal{T}(\cdot) \cup \{\mathcal{T}_i(\cdot|\theta_i)\} \to \mathcal{T}(\cdot)$;
6: **end for**
7: **return** the hybrid augmentation operators $\mathcal{T}(\cdot)$

---

### C.3 A further comparison of RGIB-SSL and other contrastive learning methods

Regarding the three information terms of RGIB-SSL in Eq. 3, we elaborate a detailed clarification by comparing it with other contrastive learning methods in the following three folds.

**(1) Supervision term $I(\boldsymbol{H}_1; \tilde{Y}) + I(\boldsymbol{H}_2; \tilde{Y})$.** Compared with the common manner of self-supervised contrastive learning, RGIB-SSL also considers utilizing noisy labels $\tilde{Y}$. We show that, although being potentially noisy, the labels $\tilde{Y}$ can also benefit learning. Besides, our experiments show that both the supervised contrastive learning method (*e.g.*, SupCon) and the self-supervised contrastive learning method (*e.g.*, GRACE) perform poorly when learning with the bilateral edge noise. An intuitive explanation is that SupCon entirely trusts and utilizes the noisy $\tilde{Y}$ while GRACE does not adopt $\tilde{Y}$. From the data perspective, RGIB-SSL performs a hybrid mode of supervised and self-supervised learning. It effectively extracts and utilizes the informative signals in $\tilde{Y}$ without entirely absorbing its noisy signals. This is achieved with the synergy of two self-supervision terms as follows.

**(2) Self-supervision alignment term $I(\boldsymbol{H}_1; \boldsymbol{H}_2)$.** Here, the main differences with common contrastive learning methods are data augmentation and loss function. Specifically, RGIB-SSL utilizes the hybrid augmentation algorithm and self-adversarial loss. Compared with the fixed augmentation manner of other contrastive learning methods [20, 54], *i.e.*, performs random perturbation on node features and edges, we propose a hybrid augmentation method with four augmentation operations. The motivation here is to encourage more diverse views with lower MI $I(A_1; A_2)$ and to avoid manual selection of augmentation operations. Besides, Prop. 4.2 provides a theoretical analysis of hybrid augmentation from information-theoretical perspectives. Empirically, compared with fixed augmentation, the hybrid brings a 3.0% average AUC promotion on Cora (please refer to Tab. 8). This means that a stronger augmentation scheme with more diverse views can help better deal with severer edge noise. Besides, the self-adversarial loss further enhances high-quality pairs and decreases low-quality counterparts. It refines the signal and brings up to 2.1% promotion.

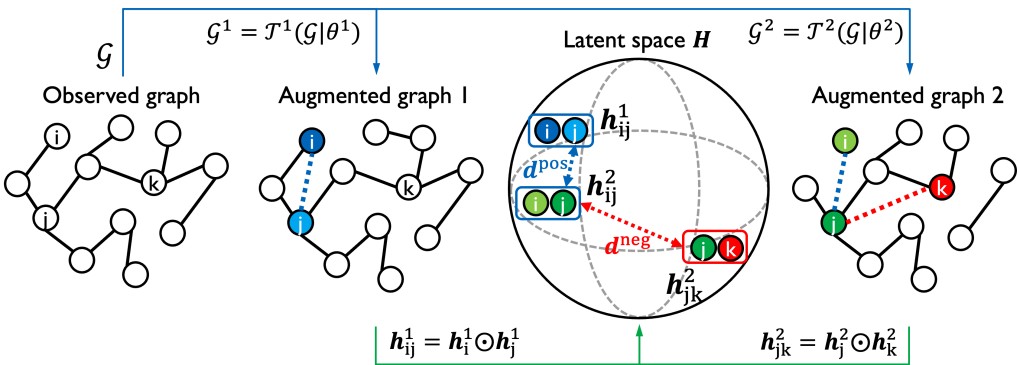

Figure 10: An illustration of the RGIB-SSL model.

**(3) Self-supervision uniformity term** $H(\boldsymbol{H}_1) + H(\boldsymbol{H}_2)$. As for the uniformity term, the understanding part in Sec. 3.2 shows that a severer edge noise brings a worse uniformity. To learn a more robust representation, we add this uniformity term to form the final loss of RGIB-SSL and adopt the Gaussian potential kernel for implementation, which is usually not considered in other contrastive learning methods. The ablation studies in Sec. 5.2 also illustrate that the uniformity term is essential, especially in dealing with label noise. Besides, the uniformity of learned representation is also enhanced (see Fig. 6), and the various query edges tend to be more uniformly distributed on the unit circle, especially for the negative edges.

### C.4 Baseline implementations

All baselines compared in this paper are based on their own original implementations. We list their source links here.

- DropEdge, `https://github.com/DropEdge/DropEdge`.
- NeuralSparse, `https://github.com/flyingdoog/PTDNet`.
- PTDNet, `https://github.com/flyingdoog/PTDNet`.
- GCN Jaccard, `https://github.com/DSE-MSU/DeepRobust`.
- GIB, `https://github.com/snap-stanford/GIB`.
- VIB, `https://github.com/RingBDStack/VIB-GSL`.
- PRI, `https://github.com/SJYuCNEL/PRI-Graphs`.
- Co-teaching, `https://github.com/bhanML/Co-teaching`.
- Peer loss functions,
  `https://github.com/weijiaheng/Multi-class-Peer-Loss-functions`.
- SupCon, `https://github.com/HobbitLong/SupContrast`.
- GRACE, `https://github.com/CRIPAC-DIG/GRACE`.

# D  A further empirical study of noise effects

This section is the extension of Sec. 3 in the main content.

## D.1  Full evaluation results

**Evaluation settings.** In this part, we provide a thorough empirical study traversing all combinations of the following settings from 5 different aspects.

- 3 GNN architectures: GCN, GAT, SAGE.

- 3 numbers of layers: $2, 4, 6$.

- 4 noise types: clean, bilateral noise, input noise, label noise.

- 3 noise ratios: $20\%, 40\%, 60\%$.

- 6 datasets: Cora, CiteSeer, PubMed, Facebook, Chameleon, Squirrel.

Then, we summarize the entire evaluation results of three kinds of GNNs as follows. As can be seen, all three common GNNs, including GCN, GAT, and SAGE, are vulnerable to bilateral edge noise.

- Tab. 27: full evaluation results with GCN.

- Tab. 28: full evaluation results with GAT.

- Tab. 29: full evaluation results with SAGE.

**Data statistics.** The statistics of the 6 datasets in our experiments are shown in Tab. 13. As can be seen, 4 homogeneous graphs (Cora, CiteSeer, PubMed, and Facebook) are with the much higher homophily values than the other 2 heterogeneous graphs (Chameleon and Squirrel).

Table 13: Dataset statistics.

| dataset | Cora | CiteSeer | PubMed | Facebook | Chameleon | Squirrel |
|---|---|---|---|---|---|---|
| # Nodes | 2,708 | 3,327 | 19,717 | 4.039 | 2,277 | 5,201 |
| # Edges | 5,278 | 4,676 | 88,648 | 88,234 | 31,421 | 198,493 |
| Homophily | 0.81 | 0.74 | 0.80 | 0.98 | 0.23 | 0.22 |

## D.2  Further investigation on the unilateral noise

**Loss distribution** We visualize the loss distribution under different scenarios to further investigate the memorization effect of GNNs. As shown in Fig. 11, two clusters are gradually separated apart with clean data, but such a learning process can be slowed down when training with noisy-input data, which confuses the model and leads to overlapped distributions. As for the label noise, the model cannot distinguish the noisy samples with a clear decision boundary to separate the clean and noisy samples apart. Besides, it is found that the model can gradually memorize the noisy edges according to the decreasing trend of the corresponding losses, where the loss distribution of noisy edges is minimized and progressively moving towards that of the clean ones.

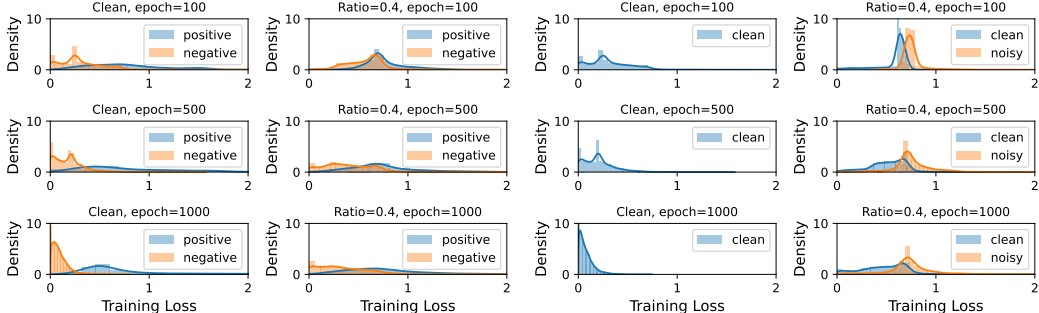

(a) Training loss of positive/negative samples with clean data (left) and 40% input-noisy data (right).

(b) Training loss of clean/noisy samples with clean data (left) and 40% label-noisy data (right).

Figure 11: Loss distribution of the standard GCN with 40% input noise (a) and 40% label noise (b).

**Visualization of edge representations** As shown in Fig. 12, the dimensionality reduction technique, T-SNE, is utilized here to measure the similarity of the learned edge representations $\boldsymbol{h}_{ij} = [\boldsymbol{h}_i, \boldsymbol{h}_j]$ with edges $e_{ij}$ in the test set $\mathcal{E}^{test}$. Compared with the representation distribution of clean data (Fig. 12(a)), which present comparatively distinct cluster boundaries, both the input noise (Fig. 12(b)) and the label noise (Fig. 12(c)) are with overlapped clusters of the positive and negative samples. Namely, the GCN model is confused by the two patterns of noise, and fails to discriminate the true or false edges with their representations.

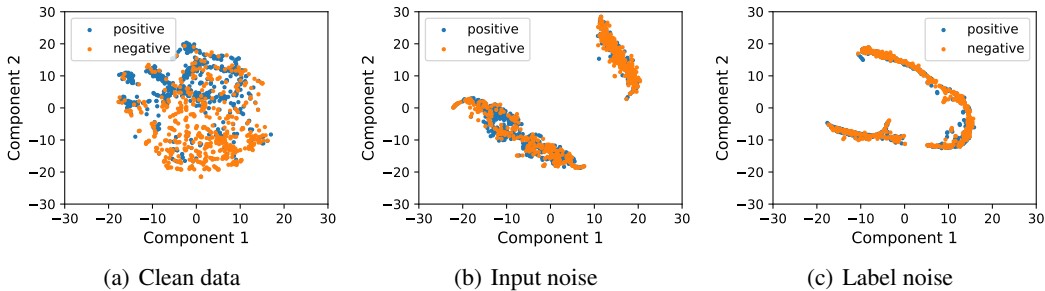

(a) Clean data        (b) Input noise        (c) Label noise

Figure 12: Visualization of the representations of predictive edges through T-SNE. The edge representations are learned on the Cora dataset by a 4-layer GCN with the standard supervised learning.

### D.3 Decoupling input noise and label noise

For a further and deeper study of the bilateral noise, we use the *edge homophily* metric to quantify the distribution of edges. Specifically, the homophily value $h_{ij}^{homo}$ of the edge $e_{ij}$ is computed as the cosine similarity of the node feature $x_i$ and $x_j$, *i.e.*, $h_{ij}^{homo} = cos(x_i, x_j)$. As shown in Fig. 13, the distributions of edge homophily are nearly the same for label noise and structure noise, where the envelope of the two distributions is almost overlapping. Here, we justify that the randomly split label noise and structure noise are indeed coupled together.

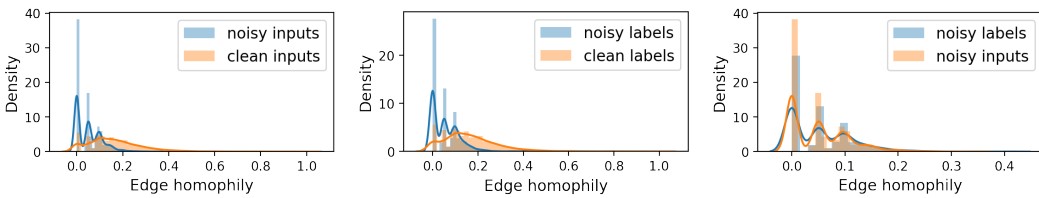

(a) Homophily distributions of input edges

(b) Homophily distributions of output edges

(c) Homophily distributions of noisy edges

Figure 13: Distributions of edge homophily with random split manner.

Based on the measurement of edge homophily, we then decouple these two kinds of noise in simulation. Here, we consider two cases that separate these two kinds of noise apart, *i.e.*, (case 1) input noise with high homophily and label noise with low homophily shown in Fig. 14, and (case 2) input noise with low homophily and label noise with high homophily in Fig. 15.

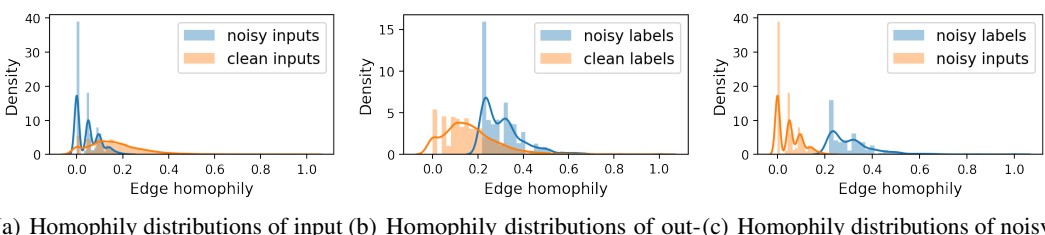

(a) Homophily distributions of input edges

(b) Homophily distributions of output edges

(c) Homophily distributions of noisy edges

Figure 14: Distributions of edge homophily for unilateral noise (case 1).

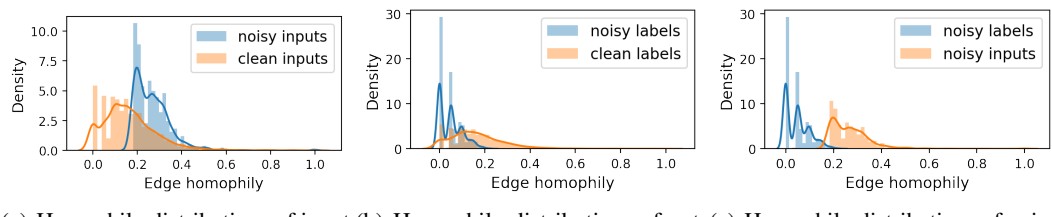

(a) Homophily distributions of input edges

(b) Homophily distributions of output edges

(c) Homophily distributions of noisy edges

Figure 15: Distributions of edge homophily for unilateral noise (case 2).

### D.4 Large-scale datasets

We conduct an empirical study on two large-scale datasets, *i.e.*, PPI (56,944 nodes and 1,612,348 edges) and Yelp (716,847 nodes and 13,954,819 edges). Here, we first evaluate the performance without any defenses (*i.e.*, standard training) with a 4-layer GCN that learns on different noisy cases. As shown in the Tab. 14 below, the added edge noise also degenerates the performance of GNN on these two large-scale datasets.

Table 14: Empirical results on PPI and Yelp dataset.

| dataset | clean | $\epsilon = 20\%$ | $\epsilon = 40\%$ | $\epsilon = 60\%$ |
|---|---|---|---|---|
| PPI (standard training) | .9076 | .8505 | .8486 | .8482 |
| Yelp (standard training) | .9143 | .8766 | .8708 | .8694 |

Next, we evaluate the effectiveness and efficiency of the proposed methods on these two datasets with $\epsilon = 40\%$ inherent noise. As shown below, RGIB can also bring promotions compared with standard training. Besides, RGIB indeed brings extra computing costs, and specifically, RGIB-SSL can be comparably more expensive than RGIB-REP. However, the extra computing costs are not so high and within an acceptable range.

Table 15: Empirical results on PPI and Yelp dataset.

| method | PPI (test AUC) | PPI (training time) | Yelp (test AUC) | Yelp (training time) |
|---|---|---|---|---|
| standard training | .8486 | 0.24h | .8708 | 2.40h |
| RGIB-SSL | .8613 | 0.25h | .8875 | 2.71h |
| RGIB-REP | .8565 | 0.24h | 8909 | 2.52h |

## D.5 Node classification task

We conduct experiments with a 2-layer GCN on Cora and Citeseer datasets with random label noise on nodes. As the empirical results are shown below, we justify that the RGIB framework can generalize to the node classification tasks with inherent label noise in nodes, where the two instantiations of RGIB also significantly outperform the standard training manner.

Table 16: Combating with label noise for node classification on the Cora dataset.

| method | clean | $\epsilon_y = 20\%$ | $\epsilon_y = 40\%$ | $\epsilon_y = 60\%$ |
|---|---|---|---|---|
| Standard training | .898 | .868 | .720 | .322 |
| RGIB-SSL | .900 | .876 | .786 | .388 |
| RGIB-REP | .894 | .862 | .760 | .312 |

Table 17: Combating with label noise for node classification on the Citeseer dataset.

| method | clean | $\epsilon_y = 20\%$ | $\epsilon_y = 40\%$ | $\epsilon_y = 60\%$ |
|---|---|---|---|---|
| Standard training | .776 | .746 | .608 | .278 |
| RGIB-SSL | .784 | .770 | .646 | .324 |
| RGIB-REP | .776 | .754 | .654 | .364 |

## D.6 Combating feature noise

Although our RGIB methods focus on edge noise, we conduct the following experiments to verify the point. Specifically, we compare standard training, RGIB-REP, and RGIB-SSL using a 4-layer GCN on six datasets with various ratios of feature noise. Here, the noisy feature is generated by adding $\epsilon_f\%$ random noise within the range of $[0, 1]$ to the normalized node feature.

The evaluation results of mean AUC are shown in the below Tab. 18 - Tab. 23. As can be seen, the noisy node feature also significantly degenerates the GNN's performance, and the degradation becomes severer as the noise ratio increases. Interestingly, compared with the standard training manner, RGIB-SSL and RGIB-REP can also promote the robustness of GNN against feature noise. For example, when learning with $\epsilon_f = 10\%$ feature noise, RGIB-SSL can bring $16.6\%$ and $6.5\%$ improvements in AUC on Cora and Citeseer datasets, respectively.

We speculate that as the graph representation $\boldsymbol{H}$ is encoded from the node feature $X$, regularizing $\boldsymbol{H}$ in RGIB objectives can also balance its dependence on $X$ and thus shows some potential robustness, even though we originally designed these objectives to handle edge noise. The above experiments show that our RGIB principle also has some merits in combating feature noise. We should note that this might be an initial verification, and a more comprehensive study will be conducted to have a rigorous conclusion in future explorations.

Table 18: Combating with feature noise on Cora dataset.

| method | $\epsilon_f = 2\%$ | $\epsilon_f = 4\%$ | $\epsilon_f = 6\%$ | $\epsilon_f = 8\%$ | $\epsilon_f = 10\%$ |
|---|---|---|---|---|---|
| Standard training | .8528 | .8629 | .8077 | .7220 | .7353 |
| RGIB-REP | .8681 | .8770 | .8679 | .8184 | .7812 |
| RGIB-SSL | **.9120** | **.9075** | **.8999** | **.8419** | **.8575** |

Table 19: Combating with feature noise on Citeseer dataset.

| method | $\epsilon_f = 2\%$ | $\epsilon_f = 4\%$ | $\epsilon_f = 6\%$ | $\epsilon_f = 8\%$ | $\epsilon_f = 10\%$ |
|---|---|---|---|---|---|
| Standard training | .8289 | .7630 | .7292 | .6786 | .6491 |
| RGIB-REP | .8543 | .7804 | .6832 | .6591 | .6455 |
| RGIB-SSL | **.9079** | **.8606** | **.7977** | **.7201** | **.6916** |

Table 20: Combating with feature noise on Pubmed dataset.

| method | $\epsilon_f = 2\%$ | $\epsilon_f = 4\%$ | $\epsilon_f = 6\%$ | $\epsilon_f = 8\%$ | $\epsilon_f = 10\%$ |
|---|---|---|---|---|---|
| Standard training | .9193 | .9064 | .9054 | .9008 | .8904 |
| RGIB-REP | .9374 | .9340 | .9252 | .9060 | .8929 |
| RGIB-SSL | **.9553** | **.9433** | **.9446** | **.9328** | **.9150** |

Table 21: Combating with feature noise on the Facebook dataset.

| method | $\epsilon_f = 2\%$ | $\epsilon_f = 4\%$ | $\epsilon_f = 6\%$ | $\epsilon_f = 8\%$ | $\epsilon_f = 10\%$ |
|---|---|---|---|---|---|
| Standard training | .9823 | .9804 | .9807 | .9738 | .9691 |
| RGIB-REP | **.9864** | **.9858** | **.9816** | .9756 | .9700 |
| RGIB-SSL | .9857 | .9840 | .9809 | **.9793** | **.9760** |

Table 22: Combating with feature noise on Chameleon dataset.

| method | $\epsilon_f = 2\%$ | $\epsilon_f = 4\%$ | $\epsilon_f = 6\%$ | $\epsilon_f = 8\%$ | $\epsilon_f = 10\%$ |
|---|---|---|---|---|---|
| Standard training | .9724 | .9642 | .9552 | .9493 | .9328 |
| RGIB-REP | **.9766** | .9712 | .9626 | .9547 | .9374 |
| RGIB-SSL | .9765 | **.9751** | **.9696** | **.9632** | **.9500** |

Table 23: Combating with feature noise on Squirrel dataset.

| method | $\epsilon_f = 2\%$ | $\epsilon_f = 4\%$ | $\epsilon_f = 6\%$ | $\epsilon_f = 8\%$ | $\epsilon_f = 10\%$ |
|---|---|---|---|---|---|
| Standard training | .9620 | .9548 | .9581 | .9465 | .9424 |
| RGIB-REP | **.9676** | .9638 | .9613 | .9584 | .9502 |
| RGIB-SSL | .9653 | **.9687** | **.9668** | **.9635** | **.9541** |

# E Full empirical results

## E.1 Further ablation studies on the trade-off parameters $\lambda$

We conduct an ablation study with the grid search of several hyper-parameters $\lambda$ in RGIB. For simplicity, we fix the weight of the supervision signal as one, *i.e.*, $\lambda_s = 1$. Then, the objective of RGIB can be formed as $\mathcal{L} = \mathcal{L}_{cls} + \lambda_1 \mathcal{R}_1 + \lambda_2 \mathcal{R}_2$, where the information regularization terms $\mathcal{R}_1/\mathcal{R}_2$ are alignment and uniformity for RGIB-SSL, while topology constraint and label constraint for RGIB-REP, respectively. As the heatmaps illustrated in Fig. 16, the $\lambda_1, \lambda_2$ are better in certain ranges. Neither too large nor too small value is not guaranteed to find a good solution.

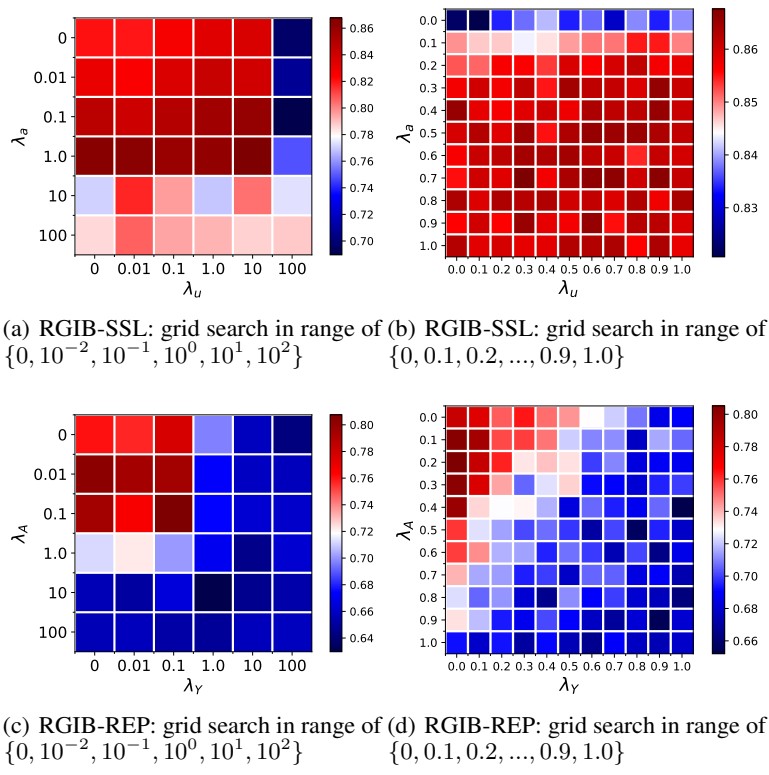

(a) RGIB-SSL: grid search in range of $\{0, 10^{-2}, 10^{-1}, 10^0, 10^1, 10^2\}$

(b) RGIB-SSL: grid search in range of $\{0, 0.1, 0.2, ..., 0.9, 1.0\}$

(c) RGIB-REP: grid search in range of $\{0, 10^{-2}, 10^{-1}, 10^0, 10^1, 10^2\}$

(d) RGIB-REP: grid search in range of $\{0, 0.1, 0.2, ..., 0.9, 1.0\}$

Figure 16: Grid search of hyper-parameters on Cora dataset ($\epsilon = 40\%$).

In addition to fixing the hyper-parameters $\lambda$ as constants, we also attempt dynamic tuning of these hyper-parameters within the training procedures, *i.e.*, using the optimization schedulers. We attempt 5 different schedulers in total, including constant, linear, sine, cosine, and exponential. As shown in Table 24, the selection of optimization schedulers greatly influences the final results, but there is no gold scheduler that consistently performs the best. Nonetheless, the constant and sine are generally better than the others.

Table 24: Comparison on different schedulers. Here, SSL/REP are short for RGIB-SSL/RGIB-REP. Experiments are performed with a 4-layer GAT and $\epsilon = 40\%$ mixed edge noise.

| dataset | Cora | | Citeseer | | Pubmed | | Facebook | | Chameleon | | Squirrel | |
|---|---|---|---|---|---|---|---|---|---|---|---|---|
| method | SSL | REP | SSL | REP | SSL | REP | SSL | REP | SSL | REP | SSL | REP |
| $constant$ | .8398 | **.7927** | **.8227** | **.7742** | .8596 | **.8416** | **.9694** | **.9778** | .9384 | **.9498** | .9293 | **.9320** |
| $linear(\cdot)$ | .8427 | .7653 | .8167 | .7559 | **.8645** | .8239 | .9669 | .9529 | .9434 | .9369 | .9316 | .9265 |
| $sin(\cdot)$ | **.8436** | .7924 | .8132 | .7680 | .8637 | .8275 | .9685 | .9594 | **.9447** | .9434 | **.9325** | .9282 |
| $cos(\cdot)$ | .8334 | .7833 | .8088 | .7647 | .8579 | .8372 | .9477 | .9629 | .9338 | .9444 | .9251 | .9303 |
| $exp(\cdot)$ | .8381 | .7815 | .8085 | .7569 | .8617 | .8177 | .9471 | .9613 | .9402 | .9418 | .9316 | .9299 |

## E.2 Combination of RGIB-REP and RGIB-SSL

We attempt to jointly combine and use RGIB-REP and RGIB-SSL simultaneously. Specifically, the "RGIB-REP+RGIB-SSL" integrates both data reparameterization technique and self-supervised learning method and the final objective for optimization equals minimizing Eq. 4 and Eq. 3 simultaneously. Then, we compare "RGIB-REP+RGIB-SSL" with standard training, RGIB-REP, and RGIB-SSL based on a 4-layer GCN on all six datasets with $\epsilon = 40\%$ bilateral noise.

As shown in the Tab. 25, the combination "RGIB-REP+RGIB-SSL" achieves considerable performances as RGIB-REP and RGIB-REP in most cases and outperforms these two instantiations on the Facebook dataset. Although we believe that a more careful finetuning of the hyper-parameters will bring further improvements to the "RGIB-REP+RGIB-SSL" combination, we suggest using one of the instantiations in practice to keep the learning objective simple and sufficient to combat the bilateral edge noise.

Table 25: Method comparison among (1) standard training, (2) RGIB-REP, (3) RGIB-SSL, and (4) the combination of RGIB-REP and RGIB-SSL.

| method | Cora | Citeseer | Pubmed | Facebook | Chameleon | Squirrel |
|---|---|---|---|---|---|---|
| Standard | .7419 | .7380 | .8748 | .9520 | .9496 | .9406 |
| RGIB-REP | .7966 | .7519 | .8834 | .9770 | .9621 | .9455 |
| RGIB-SSL | .8554 | .8427 | .8918 | .9711 | .9592 | .9426 |
| RGIB-REP+RGIB-SSL | .8351 | .8270 | .8880 | .9819 | .9570 | .9431 |

## E.3 Further case studies

**Alignment and uniformity of baseline methods.** The alignment of other methods is summarized in Tab. 26, while the uniformity is visualized in Fig. 17 and Fig. 18. Specifically, the mean alignments values on different datasets (*i.e.*, the x-axis) and various noise settings (*i.e.*, the y axis) from clean to extremely noisy. Besides, the x and y axes in the uniformity figures are the projection dimensions as the common TSNE plots. Here, we keep the same ranges of x and y axis from 0 to 128 in each plot to provide a fair and consistent overview of representation distributions under various noisy scenarios.

The intuition is that the alignment measures how the representations can be influenced when encountering topological perturbations, while uniformity describes how the representations distribute on the hyperplane. In other words, when taking the edge noise into consideration, the alignment actually measures the stability of GNN when the input graph is with extra edge noise in the testing phase, and the uniformity measures the denoising robustness of GNN when learning with inherent edge noise in the training phase.

As introduced in Sec. 3.2, desired representations should be invariant to the topological perturbations, *i.e.*, with a higher alignment. On the other hand, they should also be uniformly distributed on the hyperplane to preserve as much information as possible. Based on the concepts of alignment and uniformity, we then provide quantitive and qualitative analyses of the learned representations under edge noise in Sec. 3.2. As shown, with a higher ratio of edge noise, the alignment and uniformity can be damaged to a greater extent. Thus, these two metrics can help to comprehend the learned representations by GNN under edge noise. Thus, both metrics are indeed closely connected with the edge noise here, and we reveal that a severer edge noise brings a poorer alignment and a worse uniformity.

We have the following three observations. First, the sampling-based methods, *e.g.*, DropEdge, PTDNet, and NeuralSparse, can also promote alignment and uniformity due to their sampling mechanisms to defend the structural perturbations. Second, the contrastive methods, *e.g.*, SupCon and GRACE, are with much better alignment but much worse uniformity. The reason is that the learned representations are severely collapsed, which can be degenerated to single points seen from the uniformity plots but stay nearly unchanged when encountering structural perturbations. Third, the remaining methods are not observed with significant improvements in alignment or uniformity. When connecting the above observations with their empirical performances, we can draw a conclusion. That is, both alignment and uniformity are important to evaluate the robust methods from the perspective of representation learning. Besides, such a conclusion is in line with the previous study [35].

Table 26: Alignment comparison.

| method | Cora | | | | Citeseer | | | |
|---|---|---|---|---|---|---|---|---|
| | clean | $\varepsilon=20\%$ | $\varepsilon=40\%$ | $\varepsilon=60\%$ | clean | $\varepsilon=20\%$ | $\varepsilon=40\%$ | $\varepsilon=60\%$ |
| Standard | .616 | .687 | .695 | .732 | .445 | .586 | .689 | .696 |
| DropEdge | .606 | .670 | .712 | .740 | .463 | .557 | .603 | .637 |
| NeuralSparse | .561 | .620 | .691 | .692 | .455 | .469 | .583 | .594 |
| PTDNet | .335 | .427 | .490 | .522 | .330 | .397 | .459 | .499 |
| Co-teaching | .570 | .691 | .670 | .693 | .464 | .581 | .683 | .706 |
| Peer loss | .424 | .690 | .722 | .629 | .529 | .576 | .580 | .598 |
| Jaccard | .608 | .627 | .658 | .703 | .471 | .596 | .674 | .703 |
| GIB | .592 | .635 | .652 | .692 | .439 | .524 | .591 | .623 |
| SupCon | .060 | .045 | .033 | .024 | .086 | .077 | .060 | .065 |
| GRACE | .466 | .548 | .582 | .556 | .454 | .551 | .589 | .609 |
| **RGIB-REP** | .524 | .642 | .679 | .704 | .439 | .533 | .623 | .647 |
| **RGIB-SSL** | .475 | .543 | .578 | .615 | .418 | .505 | .533 | .542 |

**Learning curves of RGIB.** We draw the learning curves of RGIB with constant schedulers in Fig. 19, Fig. 20, Fig. 21, and Fig. 22. We normalize the values of each plotted line to $(0, 1)$ for better visualization and have the following observations.

For RGIB-SSL, the uniformity term, *i.e.*, $H(\boldsymbol{H})$, converges quickly and remains low after 200 epochs. Similarly, the alignment term, *i.e.*, $I(\boldsymbol{H}_1; \boldsymbol{H}_2)$ also converges in the early stages and keeps stable in the rest. At the same time, the supervised signal, *i.e.*, $I(\boldsymbol{H}; Y)$ gradually and steadily decreases as the training time moves forward. The learning processes are generally stable across different datasets.

As for RGIB-REP, we observe that the topology $I(\boldsymbol{Z}_A; \tilde{A})$ and label constraints $I(\boldsymbol{Z}_Y; \tilde{Y})$ can indeed adapt to noisy scenarios with different noise ratios. As can be seen, these two regularizations converge more significantly when learning on a more noisy case. That is, when noise ratio $\epsilon$ increases from 0 to 60%, these two regularizations react adaptively to the noisy data $\tilde{A}, \tilde{Y}$. Such a phenomenon shows that RGIB-REP with these two information constraints works as an effective information bottleneck to filter out the noisy signals.

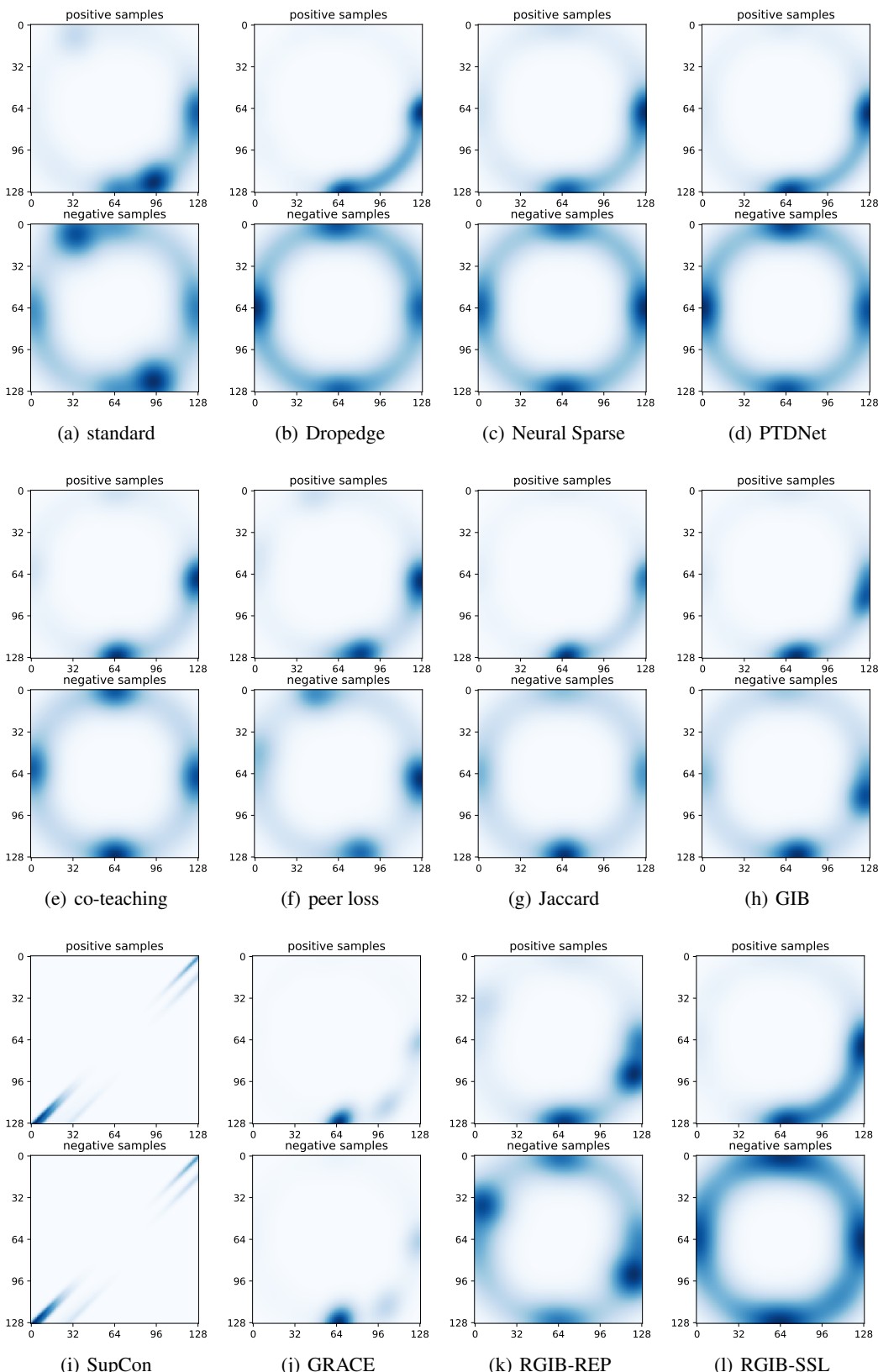

Figure 17: Uniformity on Cora with $\epsilon = 40\%$ bilateral noise.

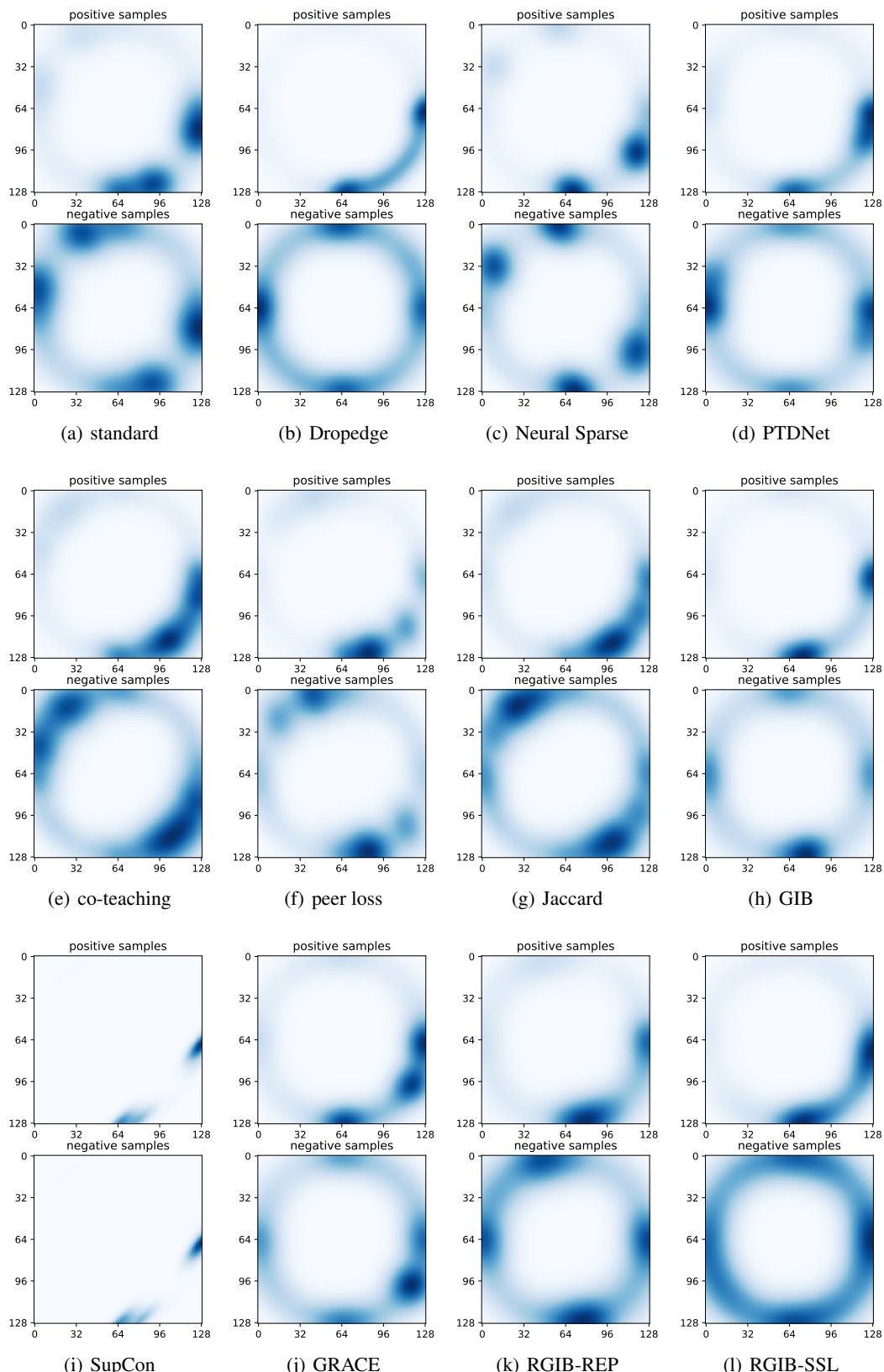

Figure 18: Uniformity on Citeseer with $\epsilon = 40\%$ bilateral noise.

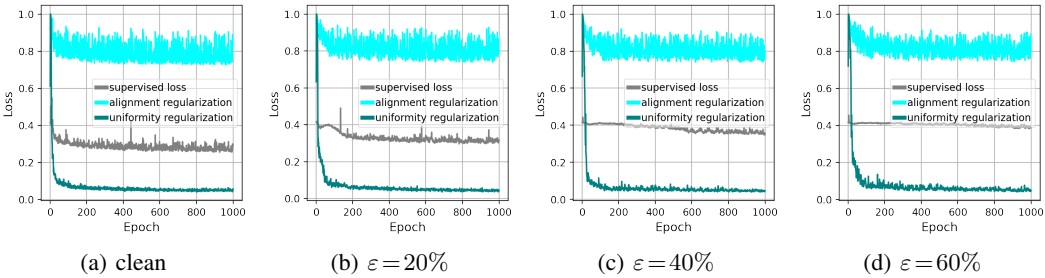

Figure 19: Learning curves of RGIB-SSL on Cora dataset.

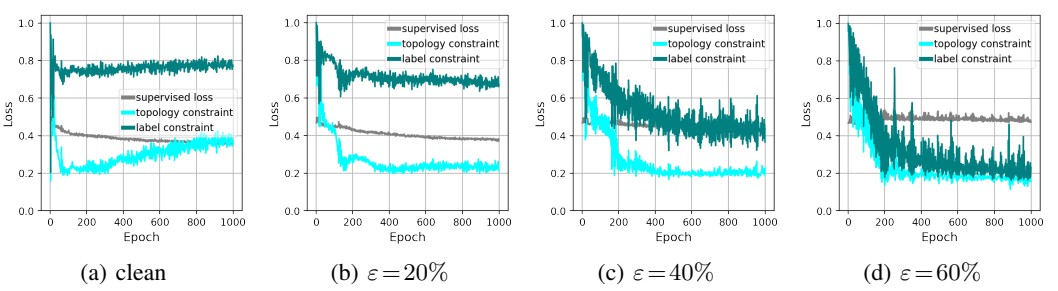

Figure 20: Learning curves of RGIB-REP on Cora dataset.

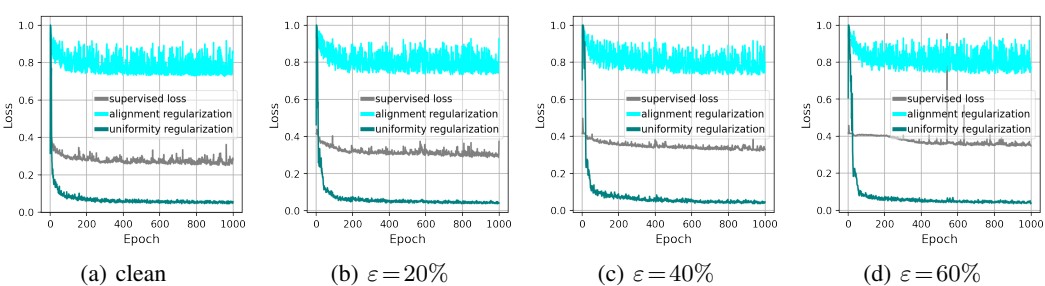

Figure 21: Learning curves of RGIB-SSL on Citeseer dataset.

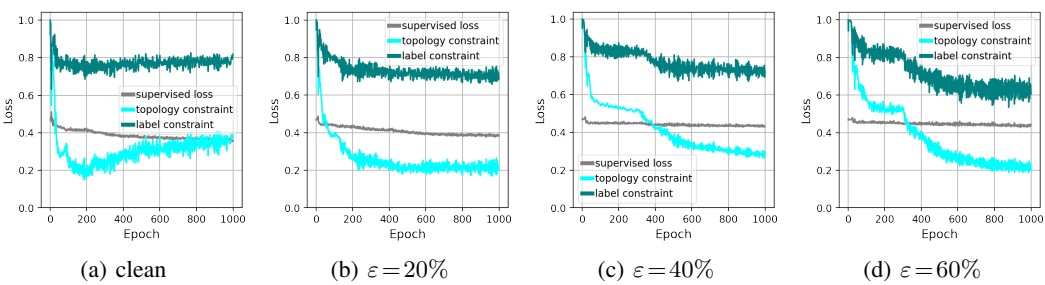

Figure 22: Learning curves of RGIB-REP on Citeseer dataset.

### E.4 Full baseline comparison with bilateral and unilateral noise

The entire evaluation with 10 baselines and two proposed methods are conducted keeping the same settings as in D.1.

Results on each dataset are summarized as follows.

- Tab. 30, Tab. 31, Tab. 32 : full results of GCN/GAT/SAGE on Cora dataset.
- Tab. 33, Tab. 34, Tab. 35 : full results of GCN/GAT/SAGE on CiteSeer dataset.
- Tab. 36, Tab. 37, Tab. 38 : full results of GCN/GAT/SAGE on PubMed dataset.
- Tab. 39, Tab. 40, Tab. 41 : full results of GCN/GAT/SAGE on Facebook dataset.
- Tab. 42, Tab. 43, Tab. 44 : full results of GCN/GAT/SAGE on Chameleon dataset.
- Tab. 45, Tab. 46, Tab. 47 : full results of GCN/GAT/SAGE on Squirrel dataset.

Table 27: Full results of GCN with edge noise.

| layers | clean | bilateral noise 20% | 40% | 60% | input noise 20% | 40% | 60% | label noise 20% | 40% | 60% |
|---|---|---|---|---|---|---|---|---|---|---|
| Cora | | | | | | | | | | |
| 2 | .9183±.0071 | .8870±.0108 | .8430±.0145 | .7959±.0190 | .9091±.0085 | .8943±.0085 | .8905±.0092 | .9020±.0041 | .8793±.0099 | .8723±.0103 |
| 4 | .8686±.0102 | .8111±.0213 | .7419±.0325 | .6970±.0377 | .8027±.0580 | .7856±.0526 | .7490±.0498 | .8281±.0133 | .8054±.0213 | .8060±.0172 |
| 6 | .8256±.0494 | .7940±.0133 | .7429±.0226 | .7177±.0207 | .8035±.0370 | .7973±.0166 | .7546±.0270 | .8249±.0112 | .7925±.0137 | .7901±.0145 |
| Citeseer | | | | | | | | | | |
| 2 | .8968±.0092 | .8651±.0084 | .8355±.0154 | .8254±.0066 | .8767±.0097 | .8615±.0090 | .8585±.0085 | .8834±.0155 | .8649±.0104 | .8674±.0122 |
| 4 | .8317±.0217 | .7864±.0145 | .7380±.0201 | .7085±.0146 | .8054±.0240 | .7708±.0325 | .7583±.0258 | .7965±.0124 | .7850±.0174 | .7659±.0240 |
| 6 | .8161±.0163 | .7355±.0387 | .7110±.0222 | .7106±.0168 | .7720±.0161 | .7460±.0401 | .7212±.0470 | .7900±.0131 | .7741±.0157 | .7648±.0229 |
| Pubmed | | | | | | | | | | |
| 2 | .9737±.0011 | .9473±.0019 | .9271±.0027 | .9141±.0041 | .9590±.0022 | .9468±.0022 | .9337±.0016 | .9646±.0024 | .9637±.0021 | .9597±.0022 |
| 4 | .9178±.0084 | .8870±.0041 | .8748±.0031 | .8641±.0041 | .8854±.0051 | .8759±.0031 | .8651±.0040 | .9030±.0082 | .9039±.0029 | .9070±.0062 |
| 6 | .9081±.0056 | .8870±.0056 | .8731±.0032 | .8640±.0036 | .8855±.0025 | .8742±.0029 | .8652±.0041 | .9050±.0083 | .9112±.0059 | .9063±.0036 |
| Facebook | | | | | | | | | | |
| 2 | .9887±.0008 | .9880±.0007 | .9866±.0007 | .9843±.0010 | .9878±.0008 | .9852±.0006 | .9834±.0011 | .9892±.0006 | .9888±.0008 | .9888±.0007 |
| 4 | .9870±.0009 | .9829±.0020 | .9520±.0424 | .9438±.0402 | .9819±.0015 | .9668±.0147 | .9622±.0154 | .9882±.0007 | .9880±.0007 | .9886±.0006 |
| 6 | .9849±.0009 | .9798±.0013 | .9609±.0138 | .9368±.0179 | .9764±.0034 | .9502±.0096 | .9469±.0160 | .9863±.0013 | .9865±.0012 | .9876±.0010 |
| Chameleon | | | | | | | | | | |
| 2 | .9823±.0027 | .9753±.0025 | .9696±.0022 | .9657±.0029 | .9784±.0017 | .9762±.0016 | .9754±.0023 | .9775±.0018 | .9769±.0018 | .9755±.0036 |
| 4 | .9797±.0021 | .9616±.0033 | .9496±.0190 | .9274±.0276 | .9608±.0038 | .9433±.0261 | .9368±.0271 | .9686±.0020 | .9580±.0021 | .9362±.0035 |
| 6 | .9752±.0036 | .9662±.0042 | .9511±.0079 | .9286±.0067 | .9656±.0045 | .9450±.0177 | .9276±.0229 | .9752±.0027 | .9766±.0035 | .9745±.0040 |
| Squirrel | | | | | | | | | | |
| 2 | .9761±.0011 | .9621±.0018 | .9519±.0020 | .9444±.0024 | .9610±.0028 | .9490±.0031 | .9401±.0036 | .9744±.0013 | .9731±.0010 | .9722±.0011 |
| 4 | .9725±.0011 | .9432±.0036 | .9406±.0031 | .9386±.0025 | .9416±.0042 | .9395±.0011 | .9411±.0040 | .9720±.0016 | .9720±.0016 | .9710±.0021 |
| 6 | .9694±.0028 | .9484±.0049 | .9429±.0038 | .9408±.0039 | .9489±.0057 | .9408±.0021 | .9386±.0022 | .9688±.0028 | .9675±.0027 | .9656±.0034 |

Table 28: Full results of GAT with edge noise.

| layers | clean | bilateral noise | | | input noise | | | label noise | | |
|---|---|---|---|---|---|---|---|---|---|---|
| | | 20% | 40% | 60% | 20% | 40% | 60% | 20% | 40% | 60% |
| | | | | | Cora | | | | | |
| 2 | .9076±.0070 | .8786±.0155 | .8489±.0131 | .8286±.0120 | .9014±.0070 | .8825±.0097 | .8880±.0077 | .8991±.0088 | .8841±.0097 | .8830±.0101 |
| 4 | .8783±.0103 | .8323±.0181 | .8005±.0212 | .7841±.0095 | .8616±.0107 | .8454±.0118 | .8377±.0146 | .8530±.0121 | .8357±.0082 | .8281±.0114 |
| 6 | .8650±.0157 | .8067±.0302 | .7514±.0389 | .7032±.0452 | .8414±.0127 | .7891±.0537 | .7678±.0641 | .8376±.0097 | .8154±.0069 | .8024±.0169 |
| | | | | | Citeseer | | | | | |
| 2 | .8911±.0090 | .8586±.0125 | .8338±.0127 | .8207±.0121 | .8689±.0096 | .8526±.0130 | .8512±.0174 | .8762±.0076 | .8650±.0102 | .8648±.0166 |
| 4 | .8386±.0138 | .8026±.0157 | .7775±.0248 | .7518±.0183 | .8191±.0092 | .8043±.0105 | .7912±.0073 | .8174±.0172 | .7998±.0143 | .7934±.0156 |
| 6 | .8299±.0098 | .7807±.0117 | .7373±.0270 | .7139±.0251 | .7970±.0134 | .7860±.0107 | .7741±.0126 | .7963±.0129 | .7883±.0162 | .7801±.0161 |
| | | | | | Pubmed | | | | | |
| 2 | .9406±.0032 | .9173±.0028 | .8984±.0030 | .8884±.0033 | .9255±.0024 | .9176±.0035 | .9102±.0030 | .9306±.0038 | .9271±.0030 | .9232±.0034 |
| 4 | .8960±.0068 | .8610±.0045 | .8434±.0042 | .8339±.0048 | .8668±.0040 | .8547±.0050 | .8476±.0037 | .8817±.0042 | .8772±.0040 | .8696±.0062 |
| 6 | .8631±.0072 | .8315±.0059 | .8116±.0073 | .8040±.0092 | .8374±.0036 | .8201±.0321 | .8067±.0306 | .8480±.0086 | .8414±.0116 | .8313±.0071 |
| | | | | | Facebook | | | | | |
| 2 | .9874±.0008 | .9869±.0005 | .9856±.0008 | .9836±.0007 | .9878±.0006 | .9861±.0006 | .9858±.0008 | .9872±.0006 | .9864±.0009 | .9857±.0008 |
| 4 | .9875±.0007 | .9857±.0015 | .9850±.0012 | .9820±.0014 | .9855±.0011 | .9827±.0019 | .9773±.0046 | .9873±.0010 | .9874±.0010 | .9874±.0005 |
| 6 | .9860±.0007 | .9805±.0025 | .9658±.0321 | .9577±.0314 | .9804±.0018 | .9738±.0044 | .9710±.0036 | .9854±.0015 | .9860±.0007 | .9867±.0012 |
| | | | | | Chameleon | | | | | |
| 2 | .9770±.0044 | .9725±.0027 | .9650±.0018 | .9625±.0018 | .9767±.0026 | .9747±.0020 | .9759±.0018 | .9746±.0023 | .9743±.0017 | .9711±.0041 |
| 4 | .9734±.0047 | .9721±.0035 | .9652±.0023 | .9605±.0031 | .9741±.0028 | .9686±.0030 | .9674±.0027 | .9740±.0037 | .9738±.0027 | .9712±.0047 |
| 6 | .9742±.0052 | .9659±.0029 | .9573±.0036 | .9482±.0054 | .9644±.0033 | .9543±.0075 | .9474±.0074 | .9722±.0043 | .9688±.0055 | .9698±.0065 |
| | | | | | Squirrel | | | | | |
| 2 | .9740±.0011 | .9680±.0007 | .9635±.0017 | .9588±.0025 | .9702±.0008 | .9690±.0010 | .9659±.0014 | .9719±.0018 | .9701±.0017 | .9686±.0012 |
| 4 | .9720±.0023 | .9581±.0046 | .9436±.0063 | .9335±.0062 | .9592±.0047 | .9455±.0075 | .9415±.0061 | .9682±.0030 | .9690±.0028 | .9686±.0021 |
| 6 | .9578±.0067 | .9507±.0050 | .9309±.0164 | .9254±.0089 | .9487±.0065 | .9419±.0041 | .9255±.0073 | .9585±.0097 | .9520±.0070 | .9507±.0162 |

Table 29: Full results of SAGE with edge noise.

| layers | clean | bilateral noise | | | input noise | | | label noise | | |
|---|---|---|---|---|---|---|---|---|---|---|
| | | 20% | 40% | 60% | 20% | 40% | 60% | 20% | 40% | 60% |
| | | | | | Cora | | | | | |
| 2 | .9045±.0066 | .8733±.0101 | .8520±.0137 | .8469±.0056 | .9006±.0074 | .8857±.0093 | .8917±.0063 | .8868±.0070 | .8695±.0106 | .8564±.0091 |
| 4 | .8664±.0109 | .8225±.0079 | .7833±.0093 | .7595±.0259 | .8607±.0110 | .8437±.0099 | .8387±.0183 | .8309±.0090 | .8046±.0074 | .7920±.0191 |
| 6 | .8426±.0207 | .7787±.0423 | .7420±.0251 | .7180±.0248 | .8256±.0222 | .7947±.0561 | .8005±.0421 | .8158±.0168 | .7707±.0235 | .7660±.0134 |
| | | | | | Citeseer | | | | | |
| 2 | .8648±.0098 | .8438±.0143 | .8404±.0113 | .8279±.0098 | .8644±.0109 | .8608±.0064 | .8647±.0161 | .8560±.0120 | .8533±.0112 | .8412±.0122 |
| 4 | .8329±.0093 | .7914±.0101 | .7686±.0213 | .7539±.0149 | .8171±.0173 | .8226±.0141 | .8157±.0203 | .8068±.0116 | .7891±.0136 | .7705±.0172 |
| 6 | .8390±.0187 | .7708±.0168 | .7223±.0614 | .7204±.0236 | .8086±.0145 | .7997±.0183 | .7872±.0145 | .7903±.0219 | .7690±.0201 | .7564±.0096 |
| | | | | | Pubmed | | | | | |
| 2 | .8995±.0044 | .9136±.0032 | .9094±.0035 | .9035±.0040 | .9295±.0025 | .9378±.0022 | .9410±.0023 | .8817±.0034 | .8793±.0043 | .8757±.0041 |
| 4 | .8446±.0058 | .8627±.0056 | .8663±.0061 | .8619±.0073 | .8715±.0080 | .8901±.0082 | .9033±.0057 | .8386±.0085 | .8330±.0104 | .8268±.0092 |
| 6 | .8360±.0224 | .8314±.0081 | .8105±.0279 | .8333±.0089 | .8224±.0335 | .8538±.0105 | .8566±.0199 | .8242±.0104 | .8161±.0071 | .8090±.0129 |
| | | | | | Facebook | | | | | |
| 2 | .9882±.0008 | .9858±.0008 | .9827±.0008 | .9788±.0012 | .9881±.0008 | .9862±.0007 | .9862±.0008 | .9867±.0006 | .9840±.0006 | .9825±.0014 |
| 4 | .9859±.0013 | .9824±.0015 | .9783±.0025 | .9698±.0040 | .9849±.0015 | .9815±.0024 | .9815±.0025 | .9844±.0015 | .9817±.0007 | .9809±.0009 |
| 6 | .9828±.0024 | .9751±.0055 | .9603±.0365 | .9616±.0091 | .9715±.0077 | .9672±.0057 | .9517±.0228 | .9761±.0136 | .9788±.0045 | .9749±.0151 |
| | | | | | Chameleon | | | | | |
| 2 | .9786±.0030 | .9683±.0036 | .9605±.0038 | .9536±.0028 | .9754±.0019 | .9734±.0015 | .9762±.0018 | .9700±.0019 | .9670±.0026 | .9603±.0049 |
| 4 | .9743±.0025 | .9645±.0035 | .9580±.0035 | .9505±.0010 | .9721±.0017 | .9721±.0021 | .9721±.0016 | .9678±.0026 | .9641±.0028 | .9605±.0051 |
| 6 | .9729±.0022 | .9606±.0046 | .9541±.0058 | .9470±.0035 | .9679±.0031 | .9658±.0041 | .9640±.0059 | .9673±.0026 | .9641±.0038 | .9625±.0042 |
| | | | | | Squirrel | | | | | |
| 2 | .9745±.0015 | .9680±.0015 | .9626±.0015 | .9570±.0016 | .9737±.0010 | .9736±.0010 | .9721±.0011 | .9692±.0015 | .9643±.0012 | .9606±.0017 |
| 4 | .9689±.0052 | .9584±.0107 | .9577±.0076 | .9541±.0021 | .9637±.0092 | .9630±.0079 | .9607±.0090 | .9663±.0020 | .9612±.0049 | .9612±.0020 |
| 6 | .9682±.0045 | .9555±.0065 | .9528±.0038 | .9461±.0054 | .9592±.0053 | .9600±.0036 | .9551±.0042 | .9574±.0192 | .9540±.0207 | .9583±.0023 |

Table 30: Full results on Cora dataset with GCN.

| | | bilateral noise | | | input noise | | | label noise | | |
|---|---|---|---|---|---|---|---|---|---|---|
| layers | method | 20% | 40% | 60% | 20% | 40% | 60% | 20% | 40% | 60% |
| L=2 | Standard | .8870±.0108 | .8430±.0145 | .7959±.0190 | .9091±.0085 | .8943±.0085 | .8905±.0092 | .9020±.0041 | .8793±.0099 | .8723±.0103 |
| | DropEdge | .8773±.0128 | .8305±.0175 | .8091±.0104 | .9016±.0086 | .8692±.0120 | .8635±.0099 | .9016±.0059 | .8791±.0089 | .8660±.0095 |
| | NeuralSparse | .8869±.0096 | .8497±.0103 | .8215±.0158 | .9118±.0063 | .8903±.0113 | .8849±.0086 | .8995±.0069 | .8797±.0073 | .8743±.0088 |
| | PTDNet | .8853±.0127 | .8451±.0168 | .8180±.0106 | .9095±.0108 | .8926±.0106 | .8825±.0096 | .9030±.0080 | .8801±.0067 | .8702±.0147 |
| | Co-teaching | .8857±.0202 | .8419±.0198 | .8026±.0237 | .9084±.0132 | .8959±.0107 | .8901±.0101 | .9021±.0203 | .8929±.0126 | .8699±.0115 |
| | Peer loss | .8867±.0115 | .8472±.0237 | .7970±.0182 | .9134±.0147 | .8993±.0171 | .8806±.0138 | .9015±.0021 | .8874±.0163 | .8813±.0291 |
| | Jaccard | .8912±.0190 | .8461±.0168 | .7964±.0182 | .9107±.0210 | .9016±.0270 | .8996±.0238 | .9094±.0107 | .8856±.0093 | .8721±.0135 |
| | GIB | .8857±.0296 | .8464±.0326 | .8037±.0309 | .9126±.0065 | .8973±.0141 | .8937±.0150 | .9083±.0136 | .8828±.0136 | .8807±.0146 |
| | SupCon | .8827±.0125 | .8451±.0157 | .8125±.0119 | .9006±.0074 | .8877±.0073 | .8846±.0124 | .8876±.0082 | .8668±.0088 | .8721±.0092 |
| | GRACE | .8588±.0158 | .8220±.0204 | .7987±.0076 | .8671±.0121 | .8102±.0126 | .8048±.0119 | .8990±.0092 | .8887±.0109 | .8896±.0074 |
| | RGIB-REP | .8915±.0091 | .8516±.0120 | .8358±.0077 | .9137±.0060 | .8958±.0048 | .8911±.0102 | .9046±.0107 | .8958±.0120 | .8935±.0087 |
| | RGIB-SSL | .9272±.0091 | .9001±.0145 | .8892±.0107 | .9342±.0049 | .9108±.0063 | .9014±.0107 | .9473±.0054 | .9427±.0051 | .9380±.0084 |
| L=4 | Standard | .8111±.0213 | .7419±.0325 | .6970±.0377 | .8027±.0580 | .7856±.0526 | .7490±.0498 | .8281±.0133 | .8054±.0213 | .8060±.0172 |
| | DropEdge | .8017±.0187 | .7423±.0335 | .7303±.0235 | .8338±.0132 | .7826±.0377 | .7454±.0425 | .8363±.0110 | .8273±.0106 | .8148±.0141 |
| | NeuralSparse | .8190±.0170 | .7318±.0379 | .7293±.0393 | .8534±.0250 | .7794±.0285 | .7637±.0529 | .8524±.0094 | .8246±.0111 | .8211±.0106 |
| | PTDNet | .8047±.0460 | .7559±.0246 | .7388±.0216 | .8433±.0443 | .8214±.0122 | .7770±.0624 | .8460±.0128 | .8214±.0166 | .8138±.0151 |
| | Co-teaching | .8197±.0236 | .7479±.0372 | .7030±.0475 | .8045±.0609 | .7871±.0564 | .7530±.0500 | .8446±.0219 | .8209±.0481 | .8157±.0246 |
| | Peer loss | .8185±.0226 | .7468±.0388 | .7018±.0473 | .8051±.0664 | .7866±.0623 | .7517±.0492 | .8325±.0201 | .8036±.0227 | .8069±.0193 |
| | Jaccard | .8143±.0218 | .7498±.0418 | .7024±.0403 | .8200±.0772 | .7838±.0558 | .7617±.0546 | .8289±.0177 | .8064±.0229 | .8148±.0227 |
| | GIB | .8198±.0331 | .7485±.0518 | .7148±.0455 | .8002±.0607 | .8099±.0566 | .7741±.0584 | .8337±.0133 | .8137±.0279 | .8157±.0270 |
| | SupCon | .8240±.0147 | .7819±.0261 | .7490±.0230 | .8349±.0124 | .8301±.0218 | .8025±.0210 | .8491±.0120 | .8275±.0115 | .8256±.0108 |
| | GRACE | .7872±.0207 | .6940±.0248 | .6929±.0140 | .7877±.0211 | .7107±.0188 | .6975±.0124 | .8531±.0166 | .8237±.0252 | .8193±.0246 |
| | RGIB-REP | .8313±.0098 | .7966±.0110 | .7591±.0142 | .8624±.0071 | .8313±.0136 | .8158±.0193 | .8554±.0149 | .8318±.0151 | .8297±.0150 |
| | RGIB-SSL | .8930±.0072 | .8554±.0167 | .8339±.0100 | .9024±.0097 | .8577±.0152 | .8421±.0156 | .9314±.0066 | .9224±.0100 | .9241±.0049 |
| L=6 | Standard | .7940±.0133 | .7429±.0226 | .7177±.0207 | .8035±.0370 | .7973±.0166 | .7546±.0270 | .8249±.0112 | .7925±.0137 | .7901±.0145 |
| | DropEdge | .7941±.0172 | .7353±.0130 | .6909±.0208 | .7900±.0560 | .7488±.0187 | .7170±.0187 | .8187±.0146 | .8187±.0076 | .7990±.0513 |
| | NeuralSparse | .7931±.0201 | .7324±.0434 | .7018±.0330 | .8082±.0418 | .7432±.0386 | .7393±.0243 | .8415±.0167 | .8160±.0181 | .8081±.0105 |
| | PTDNet | .8037±.0168 | .7601±.0225 | .7167±.0574 | .8129±.0158 | .7873±.0199 | .7570±.0511 | .8348±.0144 | .7933±.0538 | .7899±.0466 |
| | Co-teaching | .7930±.0132 | .7387±.0244 | .7151±.0262 | .8079±.0469 | .8058±.0170 | .7623±.0310 | .8298±.0365 | .8020±.0172 | .7976±.0375 |
| | Peer loss | .8015±.0211 | .7466±.0320 | .7196±.0286 | .8068±.0361 | .8031±.0169 | .7607±.0272 | .8409±.0146 | .7999±.0289 | .7918±.0240 |
| | Jaccard | .8002±.0159 | .7492±.0246 | .7211±.0250 | .8199±.0368 | .8041±.0204 | .7690±.0364 | .8277±.0138 | .8005±.0229 | .7904±.0230 |
| | GIB | .7961±.0163 | .7584±.0313 | .7201±.0300 | .8036±.0451 | .8067±.0331 | .7639±.0402 | .8347±.0111 | .8016±.0156 | .7947±.0147 |
| | SupCon | .8092±.0242 | .7365±.0227 | .6920±.0415 | .8021±.0251 | .7845±.0280 | .7434±.0257 | .8273±.0202 | .8181±.0234 | .8157±.0135 |
| | GRACE | .7576±.0148 | .7187±.0243 | .6860±.0213 | .7693±.0161 | .7171±.0223 | .6886±.0272 | .8209±.0347 | .8134±.0310 | .8102±.0234 |
| | RGIB-REP | .8103±.0137 | .7439±.0221 | .7040±.0192 | .8282±.0123 | .7857±.0142 | .7623±.0144 | .8365±.0163 | .8247±.0142 | .8240±.0119 |
| | RGIB-SSL | .8623±.0126 | .8080±.0240 | .7357±.0342 | .8632±.0187 | .7878±.0368 | .7310±.0483 | .9184±.0070 | .9120±.0108 | .9126±.0081 |

Table 31: Full results on Cora dataset with GAT.

| | | bilateral noise | | | input noise | | | label noise | | |
|---|---|---|---|---|---|---|---|---|---|---|
| layers | method | 20% | 40% | 60% | 20% | 40% | 60% | 20% | 40% | 60% |
| L=2 | Standard | .8786±.0155 | .8489±.0131 | .8286±.0120 | .9014±.0070 | .8825±.0097 | .8880±.0077 | .8991±.0088 | .8841±.0097 | .8830±.0101 |
| | DropEdge | .8741±.0114 | .8279±.0172 | .8101±.0141 | .8930±.0056 | .8720±.0063 | .8586±.0110 | .8954±.0093 | .8788±.0083 | .8717±.0106 |
| | NeuralSparse | .8820±.0134 | .8447±.0151 | .8248±.0112 | .9051±.0075 | .8884±.0121 | .8828±.0067 | .8982±.0060 | .8855±.0110 | .8761±.0129 |
| | PTDNet | .8799±.0152 | .8487±.0101 | .8314±.0075 | .9023±.0105 | .8838±.0116 | .8827±.0090 | .9019±.0086 | .8772±.0082 | .8726±.0060 |
| | Co-teaching | .8883±.0148 | .8571±.0192 | .8378±.0169 | .9069±.0134 | .8896±.0173 | .8939±.0125 | .9289±.0281 | .9100±.0358 | .8969±.0243 |
| | Peer loss | .8867±.0189 | .8562±.0206 | .8343±.0140 | .9026±.0135 | .8908±.0169 | .8937±.0073 | .9057±.0212 | .8848±.0194 | .8918±.0217 |
| | Jaccard | .8809±.0180 | .8492±.0161 | .8334±.0124 | .9066±.0142 | .8823±.0081 | .9065±.0215 | .9010±.0092 | .8919±.0186 | .8929±.0114 |
| | GIB | .8826±.0192 | .8564±.0218 | .8375±.0294 | .9260±.0330 | .9092±.0361 | .9162±.0213 | .9007±.0128 | .8928±.0195 | .8915±.0098 |
| | SupCon | .8709±.0119 | .8462±.0121 | .8300±.0132 | .8957±.0115 | .8827±.0131 | .8805±.0116 | .8881±.0065 | .8730±.0116 | .8652±.0143 |
| | GRACE | .8286±.0224 | .7564±.0229 | .7328±.0213 | .8238±.0215 | .7615±.0379 | .7309±.0297 | .8833±.0100 | .8805±.0125 | .8807±.0119 |
| | RGIB-REP | .8759±.0132 | .8374±.0104 | .8269±.0114 | .9006±.0118 | .8833±.0079 | .8798±.0131 | .8993±.0086 | .8838±.0115 | .8810±.0125 |
| | RGIB-SSL | .9142±.0092 | .8878±.0135 | .8777±.0118 | .9234±.0053 | .8973±.0067 | .8866±.0148 | .9389±.0053 | .9347±.0088 | .9311±.0057 |
| L=4 | Standard | .8323±.0181 | .8005±.0212 | .7841±.0095 | .8616±.0107 | .8454±.0118 | .8377±.0146 | .8530±.0121 | .8357±.0082 | .8281±.0114 |
| | DropEdge | .8237±.0157 | .7782±.0072 | .7515±.0050 | .8548±.0144 | .8205±.0090 | .8000±.0151 | .8516±.0100 | .8373±.0142 | .8374±.0157 |
| | NeuralSparse | .8309±.0154 | .7954±.0118 | .7769±.0135 | .8575±.0174 | .8450±.0110 | .8277±.0148 | .8503±.0123 | .8395±.0122 | .8348±.0121 |
| | PTDNet | .8364±.0147 | .8045±.0096 | .7890±.0081 | .8669±.0132 | .8445±.0155 | .8331±.0146 | .8507±.0113 | .8399±.0096 | .8370±.0117 |
| | Co-teaching | .8294±.0222 | .8001±.0300 | .7895±.0171 | .8696±.0196 | .8344±.0163 | .8423±.0180 | .8534±.0145 | .8374±.0313 | .8544±.0169 |
| | Peer loss | .8344±.0276 | .8067±.0254 | .7933±.0142 | .8520±.0202 | .8353±.0145 | .8376±.0216 | .8533±.0121 | .8527±.0075 | .8444±.0203 |
| | Jaccard | .8319±.0237 | .8001±.0235 | .7932±.0107 | .8634±.0130 | .8449±.0272 | .8406±.0141 | .8537±.0171 | .8402±.0134 | .8354±.0152 |
| | GIB | .8352±.0367 | .8111±.0244 | .7945±.0279 | .8860±.0099 | .8579±.0163 | .8493±.0404 | .8604±.0166 | .8434±.0094 | .8340±.0130 |
| | SupCon | .8324±.0102 | .8033±.0099 | .7776±.0145 | .8620±.0162 | .8441±.0098 | .8337±.0386 | .8514±.0138 | .8381±.0092 | .8318±.0121 |
| | GRACE | .7403±.0347 | .6711±.0695 | .6656±.0578 | .7707±.0267 | .7154±.0366 | .7146±.0237 | .8040±.0447 | .7988±.0292 | .8321±.0230 |
| | RGIB-REP | .8274±.0153 | .7882±.0134 | .7552±.0657 | .8652±.0138 | .8370±.0118 | .8154±.0147 | .8480±.0181 | .8332±.0129 | .8259±.0163 |
| | RGIB-SSL | .8760±.0112 | .8469±.0101 | .8304±.0176 | .8865±.0125 | .8553±.0127 | .8349±.0130 | .9163±.0090 | .9075±.0087 | .9036±.0087 |
| L=6 | Standard | .8067±.0302 | .7514±.0389 | .7032±.0452 | .8414±.0127 | .7891±.0537 | .7678±.0641 | .8376±.0097 | .8154±.0099 | .8024±.0169 |
| | DropEdge | .8051±.0111 | .7375±.0354 | .7110±.0388 | .8198±.0132 | .7514±.0615 | .7248±.0511 | .8499±.0135 | .8312±.0163 | .8112±.0079 |
| | NeuralSparse | .8169±.0130 | .7726±.0165 | .7149±.0596 | .8443±.0183 | .7997±.0149 | .7273±.0592 | .8460±.0113 | .8257±.0123 | .8149±.0224 |
| | PTDNet | .8207±.0166 | .7460±.0714 | .7145±.0621 | .8253±.0514 | .8209±.0231 | .7759±.0339 | .8464±.0106 | .8234±.0168 | .8159±.0148 |
| | Co-teaching | .8059±.0312 | .7576±.0386 | .7070±.0455 | .8496±.0165 | .7969±.0542 | .7717±.0691 | .8549±.0298 | .8204±.0284 | .8215±.0369 |
| | Peer loss | .8133±.0336 | .7572±.0411 | .7070±.0521 | .8455±.0214 | .7938±.0547 | .7679±.0737 | .8405±.0229 | .8318±.0197 | .8049±.0224 |
| | Jaccard | .8155±.0400 | .7537±.0466 | .7123±.0454 | .8495±.0142 | .7947±.0561 | .7762±.0771 | .8374±.0160 | .8245±.0067 | .8108±.0180 |
| | GIB | .8188±.0386 | .7509±.0439 | .7014±.0442 | .8452±.0392 | .8039±.0706 | .7923±.0718 | .8366±.0137 | .8220±.0108 | .8090±.0222 |
| | SupCon | .7586±.0629 | .6434±.0457 | .6115±.0607 | .7535±.0856 | .7102±.0655 | .6241±.0433 | .8088±.0568 | .8040±.0384 | .7869±.0392 |
| | GRACE | .5748±.0659 | .5949±.0650 | .5611±.0608 | .5675±.0788 | .6125±.0782 | .5537±.0607 | .5632±.0583 | .5588±.0692 | .6176±.0961 |
| | RGIB-REP | .8148±.0158 | .7553±.0179 | .6842±.0264 | .8404±.0129 | .8001±.0178 | .7433±.0663 | .8366±.0107 | .8274±.0151 | .8192±.0179 |
| | RGIB-SSL | .8613±.0107 | .8194±.0158 | .7858±.0133 | .8657±.0118 | .8213±.0189 | .8045±.0153 | .9049±.0059 | .8960±.0141 | .8985±.0117 |

Table 32: Full results on Cora dataset with SAGE.

| | | SAGE | | | | | | | | |
|---|---|---|---|---|---|---|---|---|---|---|
| | | bilateral noise | | | input noise | | | label noise | | |
| layers | method | 20% | 40% | 60% | 20% | 40% | 60% | 20% | 40% | 60% |
| L=2 | Standard | .8733±.0101 | .8520±.0137 | .8469±.0056 | .9006±.0074 | .8857±.0093 | .8917±.0063 | .8868±.0070 | .8695±.0106 | .8564±.0091 |
| | DropEdge | .8944±.0105 | .8600±.0083 | .8478±.0095 | .9013±.0080 | .9026±.0086 | .9050±.0084 | .8985±.0079 | .8725±.0107 | .8699±.0091 |
| | NeuralSparse | .8821±.0123 | .8570±.0090 | .8491±.0105 | .9020±.0090 | .8880±.0087 | .8949±.0098 | .8885±.0111 | .8681±.0110 | .8638±.0075 |
| | PTDNet | .8860±.0109 | .8536±.0101 | .8474±.0099 | .9040±.0101 | .8925±.0085 | .8947±.0070 | .8902±.0096 | .8721±.0068 | .8586±.0075 |
| | Co-teaching | .8794±.0097 | .8569±.0141 | .8557±.0080 | .8910±.0099 | .8891±.0100 | .8935±.0123 | .9071±.0216 | .8759±.0291 | .8632±.0156 |
| | Peer loss | .8817±.0102 | .8559±.0132 | .8480±.0058 | .9042±.0099 | .8866±.0174 | .8960±.0064 | .8876±.0111 | .8768±.0251 | .8739±.0149 |
| | Jaccard | .8828±.0170 | .8513±.0185 | .8474±.0081 | .9080±.0209 | .8975±.0210 | .8897±.0045 | .8922±.0112 | .8695±.0183 | .8596±.0133 |
| | GIB | .8765±.0196 | .8679±.0125 | .8546±.0246 | .9089±.0232 | .9004±.0098 | .8998±.0347 | .8938±.0082 | .8727±.0101 | .8614±.0125 |
| | SupCon | .8844±.0117 | .8507±.0135 | .8499±.0076 | .8974±.0075 | .8904±.0076 | .8943±.0122 | .8916±.0094 | .8742±.0092 | .8601±.0093 |
| | GRACE | .8123±.0113 | .7978±.0090 | .7944±.0137 | .8082±.0199 | .8010±.0160 | .7962±.0120 | .8280±.0092 | .8228±.0170 | .8261±.0102 |
| | RGIB-REP | .8748±.0094 | .8484±.0149 | .8380±.0102 | .9016±.0074 | .8876±.0075 | .8914±.0093 | .8863±.0078 | .8628±.0089 | .8449±.0077 |
| | RGIB-SSL | .9102±.0074 | .8967±.0114 | .8993±.0076 | .9196±.0072 | .9059±.0090 | .9082±.0084 | .9278±.0073 | .9174±.0089 | .9163±.0076 |
| L=4 | Standard | .8225±.0079 | .7833±.0093 | .7595±.0259 | .8607±.0110 | .8437±.0099 | .8387±.0183 | .8309±.0090 | .8046±.0074 | .7920±.0191 |
| | DropEdge | .8323±.0146 | .7944±.0130 | .7842±.0149 | .8734±.0082 | .8635±.0120 | .8628±.0146 | .8466±.0111 | .8199±.0155 | .8125±.0059 |
| | NeuralSparse | .8292±.0157 | .7930±.0108 | .7573±.0198 | .8703±.0108 | .8549±.0145 | .8596±.0061 | .8418±.0148 | .8025±.0148 | .8069±.0106 |
| | PTDNet | .8310±.0149 | .7847±.0174 | .7690±.0177 | .8626±.0141 | .8607±.0090 | .8630±.0154 | .8435±.0112 | .8125±.0117 | .7961±.0231 |
| | Co-teaching | .8237±.0139 | .7855±.0103 | .7658±.0343 | .8660±.0168 | .8533±.0149 | .8424±.0196 | .8580±.0345 | .8117±.0187 | .8088±.0275 |
| | Peer loss | .8230±.0107 | .7863±.0174 | .7626±.0294 | .8669±.0175 | .8534±.0179 | .8451±.0234 | .8363±.0142 | .8124±.0191 | .7904±.0231 |
| | Jaccard | .8261±.0153 | .7850±.0115 | .7602±.0348 | .8619±.0253 | .8465±.0227 | .8496±.0233 | .8378±.0106 | .8046±.0084 | .7976±.0288 |
| | GIB | .8286±.0059 | .7947±.0240 | .7729±.0378 | .8844±.0280 | .8437±.0158 | .8574±.0207 | .8374±.0185 | .8052±.0103 | .7947±.0229 |
| | SupCon | .8295±.0143 | .7809±.0176 | .7383±.0218 | .8568±.0115 | .8450±.0153 | .8445±.0187 | .8426±.0105 | .8150±.0170 | .7943±.0129 |
| | GRACE | .6242±.0245 | .6424±.0290 | .6711±.0452 | .6465±.0381 | .6172±.0329 | .6496±.0544 | .6434±.0384 | .6376±.0251 | .6438±.0449 |
| | RGIB-REP | .8274±.0112 | .7822±.0143 | .7692±.0202 | .8634±.0121 | .8470±.0144 | .8528±.0131 | .8367±.0149 | .8087±.0187 | .7991±.0120 |
| | RGIB-SSL | .8837±.0065 | .8728±.0116 | .8613±.0148 | .8960±.0109 | .8817±.0119 | .8825±.0113 | .9130±.0038 | .9041±.0075 | .9023±.0072 |
| L=6 | Standard | .7787±.0423 | .7420±.0251 | .7180±.0248 | .8256±.0222 | .7947±.0561 | .8005±.0421 | .8158±.0168 | .7707±.0235 | .7660±.0134 |
| | DropEdge | .8035±.0228 | .7398±.0560 | .7176±.0389 | .8262±.0159 | .8193±.0679 | .8089±.0260 | .8340±.0161 | .7993±.0091 | .7897±.0144 |
| | NeuralSparse | .7953±.0177 | .7378±.0180 | .7292±.0238 | .8384±.0120 | .8234±.0288 | .7980±.0701 | .8214±.0107 | .7908±.0136 | .7622±.0160 |
| | PTDNet | .7999±.0151 | .7604±.0169 | .7352±.0202 | .8311±.0143 | .8267±.0078 | .8109±.0140 | .8222±.0121 | .7823±.0078 | .7745±.0231 |
| | Co-teaching | .7817±.0477 | .7445±.0312 | .7212±.0332 | .8306±.0256 | .7991±.0595 | .8007±.0439 | .8324±.0256 | .7720±.0263 | .7687±.0266 |
| | Peer loss | .7781±.0451 | .7445±.0286 | .7192±.0277 | .8300±.0234 | .8020±.0624 | .8043±.0449 | .8309±.0149 | .7734±.0388 | .7652±.0262 |
| | Jaccard | .7779±.0437 | .7493±.0245 | .7277±.0238 | .8333±.0323 | .8075±.0605 | .8037±.0494 | .8148±.0186 | .7707±.0243 | .7709±.0204 |
| | GIB | .7814±.0493 | .7473±.0442 | .7349±.0437 | .8366±.0194 | .8106±.0689 | .8040±.0617 | .8172±.0258 | .7806±.0265 | .7689±.0180 |
| | SupCon | .7879±.0356 | .7019±.0285 | .6673±.0317 | .8219±.0469 | .7648±.0666 | .7159±.0717 | .8242±.0159 | .7880±.0152 | .7686±.0148 |
| | GRACE | .6866±.0160 | .6437±.0455 | .5967±.0248 | .6949±.0181 | .6536±.0365 | .6114±.0394 | .7239±.0231 | .7035±.0160 | .7014±.0111 |
| | RGIB-REP | .8049±.0146 | .7157±.0725 | .7099±.0473 | .8391±.0215 | .8149±.0234 | .7927±.0171 | .8358±.0100 | .7974±.0140 | .8046±.0135 |
| | RGIB-SSL | .8662±.0130 | .8430±.0178 | .8306±.0108 | .8746±.0091 | .8634±.0099 | .8603±.0156 | .8982±.0089 | .8930±.0108 | .8940±.0076 |

Table 33: Full results on Citeseer dataset with GCN.

| | | GCN | | | | | | | | |
|---|---|---|---|---|---|---|---|---|---|---|
| | | bilateral noise | | | input noise | | | label noise | | |
| layers | method | 20% | 40% | 60% | 20% | 40% | 60% | 20% | 40% | 60% |
| L=2 | Standard | .8651±.0084 | .8355±.0154 | .8254±.0066 | .8767±.0097 | .8615±.0090 | .8585±.0085 | .8834±.0155 | .8649±.0104 | .8674±.0122 |
| | DropEdge | .8613±.0112 | .8317±.0168 | .8112±.0158 | .8755±.0117 | .8557±.0122 | .8483±.0164 | .8862±.0084 | .8695±.0133 | .8688±.0143 |
| | NeuralSparse | .8605±.0119 | .8402±.0138 | .8239±.0069 | .8801±.0085 | .8634±.0109 | .8614±.0134 | .8827±.0094 | .8753±.0111 | .8720±.0156 |
| | PTDNet | .8646±.0155 | .8404±.0111 | .8223±.0142 | .8805±.0095 | .8647±.0112 | .8571±.0116 | .8813±.0074 | .8734±.0126 | .8708±.0097 |
| | Co-teaching | .8689±.0100 | .8416±.0215 | .8278±.0061 | .8849±.0167 | .8659±.0180 | .8659±.0090 | .8833±.0231 | .8744±.0281 | .8847±.0094 |
| | Peer loss | .8728±.0159 | .8374±.0182 | .8308±.0062 | .8783±.0089 | .8665±.0152 | .8638±.0131 | .8909±.0144 | .8653±.0276 | .8808±.0212 |
| | Jaccard | .8682±.0164 | .8406±.0230 | .8292±.0137 | .8725±.0090 | .8743±.0186 | .8696±.0158 | .8849±.0231 | .8718±.0105 | .8680±.0154 |
| | GIB | .8825±.0119 | .8476±.0331 | .8306±.0156 | .8819±.0381 | .8660±.0318 | .8688±.0136 | .8914±.0253 | .8681±.0128 | .8707±.0144 |
| | SupCon | .8344±.0101 | .8173±.0155 | .8140±.0121 | .8501±.0100 | .8325±.0129 | .8387±.0186 | .8436±.0128 | .8337±.0139 | .8269±.0131 |
| | GRACE | .8450±.0123 | .8225±.0169 | .7898±.0081 | .8529±.0074 | .8209±.0137 | .7967±.0171 | .8697±.0110 | .8646±.0122 | .8699±.0156 |
| | RGIB-REP | .8585±.0088 | .8347±.0213 | .8167±.0113 | .8751±.0074 | .8637±.0122 | .8600±.0119 | .8795±.0101 | .8638±.0175 | .8644±.0095 |
| | RGIB-SSL | .9199±.0091 | .8957±.0099 | .8759±.0087 | .9271±.0055 | .9018±.0090 | .8942±.0145 | .9495±.0059 | .9515±.0049 | .9480±.0062 |
| L=4 | Standard | .7864±.0145 | .7380±.0201 | .7085±.0146 | .8054±.0240 | .7708±.0325 | .7583±.0258 | .7965±.0124 | .7850±.0174 | .7659±.0240 |
| | DropEdge | .7635±.0106 | .7393±.0170 | .7094±.0190 | .8025±.0106 | .7730±.0147 | .7473±.0161 | .7937±.0175 | .7853±.0109 | .7632±.0142 |
| | NeuralSparse | .7765±.0123 | .7397±.0168 | .7148±.0237 | .8093±.0129 | .7809±.0160 | .7468±.0289 | .7968±.0198 | .7921±.0129 | .7752±.0209 |
| | PTDNet | .7795±.0131 | .7423±.0200 | .7283±.0130 | .8119±.0103 | .7811±.0191 | .7638±.0167 | .7968±.0141 | .7765±.0124 | .7622±.0215 |
| | Co-teaching | .7533±.0181 | .7238±.0245 | .7131±.0157 | .8059±.0263 | .7753±.0408 | .7668±.0272 | .7974±.0381 | .7877±.0291 | .7913±.0343 |
| | Peer loss | .7423±.0215 | .7345±.0213 | .7104±.0242 | .8106±.0250 | .7767±.0400 | .7653±.0315 | .7991±.0241 | .7990±.0289 | .7751±.0245 |
| | Jaccard | .7473±.0160 | .7324±.0204 | .7107±.0172 | .8176±.0283 | .7776±.0471 | .7725±.0432 | .8061±.0187 | .7887±.0196 | .7689±.0310 |
| | GIB | .7509±.0336 | .7388±.0240 | .7121±.0210 | .8070±.0398 | .7717±.0612 | .7798±.0421 | .7986±.0120 | .7852±.0186 | .7649±.0275 |
| | SupCon | .7554±.0196 | .7458±.0176 | .7299±.0218 | .8076±.0099 | .7767±.0111 | .7655±.0164 | .8024±.0108 | .7983±.0123 | .7807±.0166 |
| | GRACE | .7632±.0224 | .7242±.0219 | .6844±.0226 | .7615±.0152 | .7151±.0193 | .6830±.0232 | .7909±.0211 | .7630±.0196 | .7737±.0307 |
| | RGIB-REP | .7875±.0131 | .7519±.0181 | .7312±.0227 | .8299±.0134 | .7996±.0130 | .7771±.0178 | .8083±.0152 | .7846±.0234 | .7945±.0203 |
| | RGIB-SSL | .8694±.0108 | .8427±.0174 | .8137±.0170 | .8747±.0144 | .8461±.0109 | .8245±.0142 | .9204±.0085 | .9218±.0098 | .9250±.0071 |
| L=6 | Standard | .7355±.0387 | .7110±.0222 | .7106±.0168 | .7720±.0161 | .7460±.0401 | .7212±.0470 | .7900±.0131 | .7741±.0157 | .7648±.0229 |
| | DropEdge | .7653±.0139 | .7191±.0159 | .7006±.0304 | .7645±.0441 | .7442±.0160 | .7209±.0182 | .8007±.0152 | .7823±.0141 | .7656±.0196 |
| | NeuralSparse | .7644±.0180 | .7361±.0191 | .7036±.0387 | .7913±.0152 | .7476±.0279 | .7391±.0235 | .7793±.0391 | .7739±.0188 | .7688±.0256 |
| | PTDNet | .7661±.0153 | .7401±.0134 | .7072±.0385 | .7882±.0134 | .7728±.0164 | .7464±.0188 | .7913±.0122 | .7742±.0155 | .7582±.0151 |
| | Co-teaching | .7375±.0468 | .7171±.0299 | .7188±.0248 | .7813±.0185 | .7550±.0475 | .7310±.0496 | .8034±.0117 | .7885±.0275 | .7701±.0245 |
| | Peer loss | .7398±.0387 | .7105±.0288 | .7166±.0250 | .7783±.0158 | .7532±.0488 | .7301±.0509 | .7915±.0169 | .7748±.0299 | .7695±.0239 |
| | Jaccard | .7415±.0444 | .7162±.0307 | .7135±.0171 | .7793±.0263 | .7459±.0416 | .7369±.0473 | .7925±.0176 | .7825±.0245 | .7710±.0298 |
| | GIB | .7370±.0531 | .7226±.0268 | .7175±.0234 | .7743±.0435 | .7722±.0697 | .7276±.0534 | .7948±.0178 | .7803±.0221 | .7681±.0251 |
| | SupCon | .7714±.0122 | .7413±.0166 | .7205±.0231 | .7907±.0161 | .7842±.0109 | .7670±.0183 | .7745±.0231 | .7659±.0114 | .7573±.0180 |
| | GRACE | .6995±.0221 | .6901±.0210 | .6946±.0152 | .7041±.0240 | .7037±.0247 | .6818±.0169 | .7752±.0230 | .7739±.0151 | .7773±.0351 |
| | RGIB-REP | .7725±.0077 | .7429±.0202 | .7232±.0110 | .7781±.0218 | .7559±.0130 | .7415±.0221 | .7900±.0252 | .7797±.0181 | .7886±.0194 |
| | RGIB-SSL | .8417±.0169 | .7995±.0148 | .7673±.0137 | .8379±.0065 | .8026±.0173 | .7793±.0207 | .8984±.0144 | .9062±.0066 | .9060±.0098 |

Table 34: Full results on Citeseer dataset with GAT.

| | | GAT | | | | | | | | |
|---|---|---|---|---|---|---|---|---|---|---|
| layers | method | bilateral noise | | | input noise | | | label noise | | |
| | | 20% | 40% | 60% | 20% | 40% | 60% | 20% | 40% | 60% |
| L=2 | Standard | .8586±.0125 | .8338±.0127 | .8207±.0121 | .8689±.0096 | .8526±.0130 | .8512±.0174 | .8762±.0076 | .8650±.0102 | .8648±.0166 |
| | DropEdge | .8566±.0113 | .8333±.0183 | .8100±.0098 | .8750±.0079 | .8496±.0101 | .8512±.0121 | .8820±.0086 | .8679±.0112 | .8673±.0114 |
| | NeuralSparse | .8573±.0101 | .8431±.0151 | .8222±.0092 | .8743±.0117 | .8577±.0067 | .8580±.0135 | .8826±.0080 | .8724±.0076 | .8657±.0089 |
| | PTDNet | .8602±.0107 | .8381±.0137 | .8157±.0075 | .8755±.0090 | .8560±.0084 | .8574±.0154 | .8784±.0120 | .8693±.0098 | .8669±.0142 |
| | Co-teaching | .8628±.0220 | .8366±.0124 | .8199±.0194 | .8720±.0128 | .8521±.0139 | .8510±.0224 | .8924±.0122 | .8888±.0365 | .8919±.0305 |
| | Peer loss | .8637±.0125 | .8378±.0170 | .8235±.0120 | .8721±.0172 | .8529±.0173 | .8559±.0216 | .8878±.0185 | .8653±.0288 | .8631±.0258 |
| | Jaccard | .8615±.0197 | .8379±.0222 | .8223±.0124 | .8841±.0079 | .8556±.0119 | .8498±.0309 | .8843±.0143 | .8676±.0195 | .8661±.0256 |
| | GIB | .8610±.0230 | .8462±.0114 | .8324±.0316 | .8909±.0091 | .8823±.0188 | .8488±.0276 | .8781±.0135 | .8739±.0144 | .8741±.0156 |
| | SupCon | .8495±.0100 | .8138±.0174 | .8155±.0099 | .8611±.0086 | .8454±.0111 | .8393±.0172 | .8558±.0137 | .8459±.0170 | .8379±.0185 |
| | GRACE | .8092±.0221 | .7564±.0264 | .7479±.0278 | .8014±.0370 | .7628±.0240 | .7433±.0245 | .8788±.0146 | .8768±.0073 | .8654±.0172 |
| | RGIB-REP | .8545±.0108 | .8310±.0127 | .8137±.0091 | .8736±.0107 | .8566±.0097 | .8503±.0159 | .8778±.0093 | .8696±.0081 | .8614±.0084 |
| | RGIB-SSL | .9106±.0102 | .8829±.0058 | .8677±.0095 | .9172±.0083 | .8909±.0086 | .8785±.0121 | .9419±.0071 | .9410±.0087 | .9410±.0090 |
| L=4 | Standard | .8026±.0157 | .7775±.0248 | .7518±.0183 | .8191±.0092 | .8043±.0105 | .7912±.0073 | .8174±.0172 | .7998±.0143 | .7934±.0156 |
| | DropEdge | .8063±.0079 | .7624±.0211 | .7434±.0124 | .8171±.0132 | .7977±.0178 | .7814±.0162 | .8262±.0148 | .8103±.0178 | .8057±.0148 |
| | NeuralSparse | .7958±.0142 | .7761±.0172 | .7550±.0129 | .8282±.0130 | .8088±.0088 | .7911±.0174 | .8259±.0119 | .8135±.0092 | .7986±.0109 |
| | PTDNet | .8000±.0113 | .7734±.0198 | .7597±.0185 | .8254±.0105 | .8132±.0089 | .7950±.0143 | .8137±.0243 | .8082±.0094 | .8036±.0139 |
| | Co-teaching | .8016±.0184 | .7807±.0315 | .7521±.0267 | .8213±.0173 | .8068±.0156 | .7903±.0105 | .8402±.0220 | .8109±.0316 | .7947±.0350 |
| | Peer loss | .8064±.0178 | .7802±.0253 | .7544±.0191 | .8246±.0145 | .8108±.0122 | .7945±.0113 | .8160±.0329 | .8045±.0185 | .7925±.0207 |
| | Jaccard | .8098±.0222 | .7771±.0273 | .7517±.0186 | .8258±.0143 | .8083±.0138 | .7901±.0073 | .8206±.0168 | .8036±.0176 | .7999±.0215 |
| | GIB | .8170±.0230 | .7884±.0341 | .7645±.0247 | .8422±.0365 | .8112±.0212 | .7972±.0305 | .8192±.0249 | .8080±.0155 | .8010±.0177 |
| | SupCon | .7940±.0114 | .7728±.0125 | .7478±.0145 | .8137±.0115 | .8003±.0116 | .7777±.0409 | .8038±.0114 | .7972±.0198 | .7852±.0201 |
| | GRACE | .7319±.0433 | .6611±.0395 | .6449±.0579 | .7216±.0252 | .5947±.0660 | .6060±.0507 | .7775±.1040 | .7739±.0475 | .7882±.0328 |
| | RGIB-REP | .7991±.0107 | .7743±.0164 | .7418±.0121 | .8155±.0156 | .7905±.0157 | .7372±.0908 | .8108±.0118 | .7946±.0180 | .7935±.0131 |
| | RGIB-SSL | .8520±.0145 | .8306±.0149 | .8029±.0098 | .8592±.0120 | .8251±.0132 | .8145±.0110 | .9084±.0091 | .9101±.0076 | .9102±.0117 |
| L=6 | Standard | .7807±.0117 | .7373±.0270 | .7139±.0251 | .7970±.0134 | .7860±.0107 | .7741±.0126 | .7963±.0129 | .7883±.0162 | .7801±.0161 |
| | DropEdge | .7768±.0088 | .7477±.0195 | .7116±.0119 | .7854±.0232 | .7640±.0188 | .7425±.0362 | .8114±.0132 | .7840±.0217 | .7826±.0186 |
| | NeuralSparse | .7704±.0099 | .7462±.0170 | .7242±.0138 | .8047±.0101 | .7647±.0372 | .7248±.0596 | .8087±.0235 | .7855±.0176 | .7880±.0148 |
| | PTDNet | .7805±.0193 | .7503±.0223 | .7286±.0237 | .7927±.0287 | .7822±.0132 | .7579±.0355 | .8002±.0085 | .7977±.0134 | .7890±.0145 |
| | Co-teaching | .7819±.0141 | .7399±.0335 | .7236±.0292 | .7964±.0189 | .7809±.0183 | .7740±.0185 | .7933±.0406 | .7918±.0348 | .7979±.0245 |
| | Peer loss | .7846±.0214 | .7459±.0294 | .7187±.0259 | .7979±.0172 | .7955±.0168 | .7796±.0218 | .7957±.0273 | .7865±.0285 | .7912±.0148 |
| | Jaccard | .7902±.0117 | .7365±.0332 | .7157±.0248 | .8056±.0264 | .8038±.0226 | .7733±.0245 | .7964±.0226 | .7936±.0255 | .7847±.0218 |
| | GIB | .7818±.0230 | .7378±.0285 | .7137±.0416 | .8161±.0267 | .7995±.0183 | .7762±.0176 | .8002±.0155 | .7955±.0166 | .7794±.0244 |
| | SupCon | .7370±.0524 | .7160±.0462 | .6670±.0442 | .7667±.0402 | .7729±.0356 | .6999±.0597 | .7810±.0219 | .7752±.0119 | .7591±.0362 |
| | GRACE | .5068±.0128 | .5034±.0106 | .5108±.0319 | .5058±.0096 | .4956±.0069 | .5379±.0427 | .5181±.0547 | .5288±.0467 | .5068±.0178 |
| | RGIB-REP | .7817±.0129 | .7062±.0681 | .7254±.0188 | .7883±.0160 | .7769±.0168 | .7620±.0176 | .7981±.0092 | .7711±.0487 | .7817±.0164 |
| | RGIB-SSL | .8275±.0148 | .7989±.0136 | .7681±.0140 | .8261±.0096 | .8024±.0087 | .7806±.0174 | .8855±.0103 | .8918±.0143 | .8940±.0119 |

Table 35: Full results on Citeseer dataset with SAGE.

| | | SAGE | | | | | | | | |
|---|---|---|---|---|---|---|---|---|---|---|
| layers | method | bilateral noise | | | input noise | | | label noise | | |
| | | 20% | 40% | 60% | 20% | 40% | 60% | 20% | 40% | 60% |
| L=2 | Standard | .8438±.0143 | .8404±.0113 | .8279±.0098 | .8644±.0109 | .8608±.0064 | .8647±.0161 | .8560±.0120 | .8533±.0112 | .8412±.0122 |
| | DropEdge | .8654±.0118 | .8593±.0129 | .8503±.0125 | .8834±.0059 | .8791±.0108 | .8806±.0101 | .8809±.0085 | .8734±.0064 | .8710±.0170 |
| | NeuralSparse | .8658±.0079 | .8478±.0109 | .8437±.0069 | .8746±.0102 | .8674±.0086 | .8722±.0183 | .8670±.0120 | .8645±.0136 | .8485±.0105 |
| | PTDNet | .8620±.0125 | .8488±.0113 | .8477±.0078 | .8740±.0094 | .8675±.0120 | .8702±.0134 | .8657±.0138 | .8576±.0147 | .8530±.0073 |
| | Co-teaching | .8505±.0143 | .8436±.0179 | .8323±.0144 | .8652±.0133 | .8606±.0067 | .8650±.0247 | .8787±.0269 | .8660±.0101 | .8392±.0363 |
| | Peer loss | .8525±.0157 | .8440±.0120 | .8319±.0186 | .8661±.0199 | .8693±.0113 | .8735±.0151 | .8563±.0278 | .8535±.0107 | .8563±.0241 |
| | Jaccard | .8514±.0208 | .8492±.0143 | .8291±.0183 | .8691±.0282 | .8805±.0205 | .8715±.0152 | .8616±.0120 | .8629±.0210 | .8413±.0167 |
| | GIB | .8574±.0192 | .8577±.0134 | .8323±.0125 | .8693±.0186 | .8632±.0310 | .8739±.0405 | .8600±.0197 | .8576±.0182 | .8403±.0157 |
| | SupCon | .8344±.0106 | .8241±.0123 | .8168±.0111 | .8485±.0133 | .8493±.0147 | .8484±.0161 | .8487±.0109 | .8394±.0096 | .8323±.0113 |
| | GRACE | .8283±.0295 | .8319±.0192 | .8253±.0208 | .8361±.0190 | .8273±.0222 | .8351±.0134 | .8434±.0193 | .8422±.0119 | .8400±.0233 |
| | RGIB-REP | .8514±.0120 | .8359±.0093 | .8213±.0112 | .8614±.0118 | .8537±.0105 | .8664±.0153 | .8592±.0111 | .8533±.0134 | .8455±.0123 |
| | RGIB-SSL | .9003±.0104 | .8894±.0119 | .8916±.0078 | .9045±.0076 | .8945±.0097 | .8992±.0141 | .9143±.0082 | .9075±.0080 | .9087±.0108 |
| L=4 | Standard | .7914±.0101 | .7686±.0213 | .7539±.0149 | .8171±.0173 | .8226±.0141 | .8157±.0203 | .8068±.0116 | .7891±.0136 | .7705±.0172 |
| | DropEdge | .7889±.0182 | .7850±.0117 | .7678±.0206 | .8300±.0130 | .8032±.0953 | .8271±.0115 | .8058±.0118 | .7899±.0173 | .7826±.0282 |
| | NeuralSparse | .7934±.0115 | .7746±.0195 | .7639±.0175 | .8331±.0119 | .8242±.0176 | .8324±.0144 | .8111±.0146 | .7904±.0229 | .7747±.0242 |
| | PTDNet | .7972±.0105 | .7804±.0201 | .7563±.0230 | .8299±.0177 | .8259±.0119 | .8374±.0110 | .8121±.0168 | .7892±.0103 | .7845±.0166 |
| | Co-teaching | .7928±.0189 | .7679±.0230 | .7557±.0214 | .8226±.0172 | .8297±.0202 | .8256±.0263 | .8233±.0203 | .8083±.0218 | .7836±.0347 |
| | Peer loss | .7913±.0116 | .7710±.0307 | .7556±.0238 | .8197±.0264 | .8321±.0155 | .8239±.0275 | .8135±.0131 | .7940±.0194 | .7885±.0195 |
| | Jaccard | .7904±.0108 | .7691±.0269 | .7626±.0183 | .8359±.0251 | .8244±.0189 | .8290±.0264 | .8167±.0119 | .7972±.0157 | .7744±.0227 |
| | GIB | .7931±.0134 | .7739±.0254 | .7691±.0291 | .8199±.0278 | .8306±.0209 | .8217±.0273 | .8151±.0152 | .7947±.0170 | .7737±.0266 |
| | SupCon | .7870±.0130 | .7672±.0145 | .7641±.0125 | .8191±.0197 | .8081±.0138 | .8036±.0211 | .7982±.0137 | .7898±.0121 | .7783±.0172 |
| | GRACE | .6196±.0253 | .6404±.0258 | .6308±.0323 | .6286±.0287 | .6365±.0223 | .6198±.0157 | .6495±.0202 | .6431±.0210 | .6411±.0191 |
| | RGIB-REP | .7854±.0123 | .7703±.0123 | .7562±.0200 | .8195±.0140 | .8151±.0118 | .8134±.0221 | .8038±.0158 | .7863±.0140 | .7799±.0212 |
| | RGIB-SSL | .8545±.0157 | .8482±.0147 | .8352±.0127 | .8706±.0074 | .8525±.0118 | .8564±.0141 | .8867±.0088 | .8866±.0161 | .8903±.0124 |
| L=6 | Standard | .7708±.0168 | .7223±.0614 | .7204±.0236 | .8086±.0145 | .7997±.0183 | .7872±.0145 | .7903±.0219 | .7690±.0201 | .7564±.0096 |
| | DropEdge | .7756±.0143 | .7485±.0104 | .7290±.0261 | .8070±.0125 | .8060±.0165 | .8026±.0195 | .7861±.0174 | .7769±.0149 | .7534±.0212 |
| | NeuralSparse | .7757±.0175 | .7564±.0205 | .7306±.0173 | .8039±.0144 | .8008±.0157 | .8016±.0223 | .7902±.0189 | .7762±.0177 | .7588±.0160 |
| | PTDNet | .7844±.0116 | .7525±.0270 | .7435±.0108 | .8070±.0130 | .8128±.0108 | .8085±.0135 | .7948±.0165 | .7830±.0182 | .7634±.0210 |
| | Co-teaching | .7792±.0246 | .7248±.0642 | .7268±.0297 | .8087±.0136 | .8076±.0234 | .7864±.0147 | .8011±.0394 | .7671±.0406 | .7626±.0232 |
| | Peer loss | .7796±.0158 | .7213±.0628 | .7294±.0233 | .8135±.0225 | .8012±.0206 | .7914±.0208 | .8005±.0409 | .7710±.0316 | .7672±.0098 |
| | Jaccard | .7800±.0257 | .7267±.0606 | .7248±.0330 | .8166±.0328 | .8058±.0177 | .7980±.0250 | .7906±.0274 | .7749±.0269 | .7617±.0147 |
| | GIB | .7823±.0358 | .7352±.0808 | .7247±.0332 | .8180±.0388 | .8073±.0241 | .8049±.0378 | .7967±.0223 | .7710±.0267 | .7576±.0097 |
| | SupCon | .7649±.0176 | .7193±.0426 | .7040±.0231 | .7864±.0187 | .7802±.0219 | .7529±.0419 | .7775±.0174 | .7715±.0206 | .7517±.0235 |
| | GRACE | .6608±.0371 | .6767±.0441 | .6433±.0494 | .6591±.0344 | .6660±.0366 | .6670±.0292 | .6966±.0367 | .6678±.0682 | .6979±.0476 |
| | RGIB-REP | .7766±.0070 | .7479±.0150 | .7427±.0127 | .7981±.0104 | .7939±.0149 | .7894±.0238 | .7959±.0137 | .7865±.0163 | .7695±.0169 |
| | RGIB-SSL | .8372±.0169 | .8226±.0189 | .8184±.0135 | .8441±.0078 | .8252±.0129 | .8288±.0114 | .8795±.0112 | .8802±.0100 | .8749±.0178 |

Table 36: Full results on Pubmed dataset with GCN.

| layers | method | bilateral noise 20% | 40% | 60% | input noise 20% | 40% | 60% | label noise 20% | 40% | 60% |
|---|---|---|---|---|---|---|---|---|---|---|
| | | | | | GCN | | | | | |
| L=2 | Standard | .9473±.0019 | .9271±.0027 | .9141±.0041 | .9590±.0022 | .9468±.0022 | .9337±.0016 | .9646±.0024 | .9637±.0021 | .9597±.0022 |
| | DropEdge | .9394±.0025 | .9155±.0027 | .8994±.0036 | .9467±.0022 | .9302±.0021 | .9146±.0022 | .9594±.0026 | .9558±.0027 | .9519±.0019 |
| | NeuralSparse | .9479±.0021 | .9251±.0039 | .9120±.0029 | .9558±.0019 | .9315±.0033 | .9269±.0014 | .9654±.0015 | .9525±.0023 | .9588±.0017 |
| | PTDNet | .9467±.0018 | .9264±.0025 | .9111±.0032 | .9554±.0026 | .9320±.0035 | .9272±.0027 | .9651±.0018 | .9616±.0021 | .9584±.0021 |
| | Co-teaching | .9502±.0085 | .9335±.0104 | .9160±.0096 | .9510±.0098 | .9331±.0054 | .9255±.0113 | .9676±.0018 | .9560±.0127 | .9608±.0098 |
| | Peer loss | .9500±.0034 | .9339±.0044 | .9140±.0138 | .9558±.0068 | .9397±.0019 | .9283±.0090 | .9615±.0175 | .9521±.0178 | .9545±.0144 |
| | Jaccard | .9496±.0039 | .9325±.0087 | .9235±.0081 | .9554±.0019 | .9327±.0067 | .9230±.0159 | .9580±.0032 | .9532±.0086 | .9591±.0054 |
| | GIB | .9509±.0205 | .9268±.0074 | .9131±.0066 | .9599±.0303 | .9303±.0047 | .9228±.0234 | .9559±.0076 | .9573±.0040 | .9597±.0099 |
| | SupCon | .9345±.0020 | .9257±.0017 | .9118±.0031 | .9583±.0015 | .9345±.0030 | .9214±.0023 | .9625±.0021 | .9522±.0018 | .9506±.0010 |
| | GRACE | .9341±.0032 | .9319±.0027 | .9154±.0049 | .9409±.0046 | .9321±.0032 | .9225±.0078 | .9489±.0034 | .9516±.0027 | .9511±.0022 |
| | RGIB-REP | .9537±.0013 | .9368±.0018 | .9270±.0036 | .9579±.0022 | .9467±.0023 | .9365±.0016 | .9696±.0033 | .9690±.0023 | .9671±.0015 |
| | RGIB-SSL | .9585±.0022 | .9471±.0032 | .9301±.0021 | .9425±.0088 | .9305±.0098 | .9121±.0125 | .9719±.0020 | .9724±.0015 | .9711±.0012 |
| L=4 | Standard | .8870±.0041 | .8748±.0031 | .8641±.0041 | .8854±.0051 | .8759±.0031 | .8651±.0040 | .9030±.0082 | .9039±.0029 | .9070±.0062 |
| | DropEdge | .8711±.0149 | .8482±.0045 | .8354±.0062 | .8682±.0158 | .8456±.0036 | .8376±.0046 | .9313±.0071 | .9201±.0091 | .9240±.0077 |
| | NeuralSparse | .8908±.0080 | .8733±.0022 | .8630±.0049 | .8931±.0090 | .8720±.0043 | .8649±.0041 | .9272±.0108 | .9136±.0117 | .9089±.0084 |
| | PTDNet | .8872±.0071 | .8733±.0036 | .8623±.0050 | .8903±.0087 | .8776±.0078 | .8609±.0055 | .9219±.0122 | .9099±.0104 | .9093±.0101 |
| | Co-teaching | .8943±.0090 | .8760±.0117 | .8638±.0093 | .8931±.0045 | .8792±.0036 | .8606±.0083 | .9315±.0075 | .9291±.0327 | .9319±.0324 |
| | Peer loss | .8961±.0130 | .8815±.0099 | .8566±.0057 | .8917±.0076 | .8811±.0127 | .8643±.0129 | .9126±.0116 | .9101±.0046 | .9210±.0095 |
| | Jaccard | .8872±.0036 | .8803±.0060 | .8512±.0136 | .8987±.0221 | .8764±.0099 | .8639±.0073 | .9098±.0110 | .9135±.0116 | .9096±.0132 |
| | GIB | .8899±.0239 | .8729±.0205 | .8544±.0051 | .8932±.0256 | .8808±.0053 | .8618±.0317 | .9037±.0089 | .9114±.0065 | .9064±.0059 |
| | SupCon | .8853±.0061 | .8718±.0110 | .8525±.0108 | .8867±.0080 | .8739±.0033 | .8558±.0042 | .9131±.0068 | .9108±.0095 | .9162±.0125 |
| | GRACE | .8922±.0034 | .8749±.0098 | .8588±.0042 | .8810±.0034 | .8795±.0099 | .8593±.0040 | .9234±.0088 | .9252±.0052 | .9255±.0043 |
| | RGIB-REP | .9017±.0044 | .8834±.0082 | .8652±.0038 | .9008±.0033 | .8822±.0054 | .8687±.0056 | .9357±.0028 | .9343±.0062 | .9332±.0062 |
| | RGIB-SSL | .9225±.0125 | .8918±.0065 | .8697±.0053 | .9126±.0046 | .8889±.0052 | .8693±.0049 | .9594±.0026 | .9604±.0028 | .9613±.0023 |
| L=6 | Standard | .8870±.0056 | .8731±.0032 | .8640±.0036 | .8855±.0025 | .8742±.0029 | .8652±.0041 | .9050±.0083 | .9112±.0059 | .9063±.0036 |
| | DropEdge | .8623±.0039 | .8421±.0044 | .8342±.0058 | .8623±.0054 | .8407±.0051 | .8328±.0058 | .9140±.0054 | .9102±.0082 | .9092±.0072 |
| | NeuralSparse | .8814±.0049 | .8586±.0029 | .8603±.0060 | .8792±.0027 | .8695±.0031 | .8612±.0046 | .9140±.0065 | .9130±.0047 | .9080±.0077 |
| | PTDNet | .8807±.0053 | .8610±.0026 | .8518±.0040 | .8791±.0040 | .8708±.0032 | .8619±.0040 | .9123±.0074 | .9086±.0079 | .9105±.0072 |
| | Co-teaching | .8850±.0126 | .8698±.0101 | .8568±.0050 | .8773±.0023 | .8767±.0113 | .8691±.0095 | .9103±.0372 | .9311±.0290 | .9204±.0162 |
| | Peer loss | .8860±.0125 | .8633±.0079 | .8534±.0048 | .8734±.0023 | .8775±.0107 | .8668±.0042 | .9155±.0070 | .9172±.0089 | .9251±.0107 |
| | Jaccard | .8852±.0092 | .8658±.0085 | .8555±.0055 | .8711±.0031 | .8794±.0126 | .8670±.0200 | .9045±.0137 | .9118±.0108 | .9065±.0059 |
| | GIB | .8807±.0046 | .8738±.0140 | .8605±.0193 | .8718±.0070 | .8767±.0110 | .8623±.0114 | .9081±.0131 | .9136±.0147 | .9155±.0106 |
| | SupCon | .8716±.0090 | .8627±.0043 | .8533±.0045 | .8705±.0108 | .8733±.0026 | .8643±.0047 | .9232±.0081 | .9294±.0075 | .9218±.0092 |
| | GRACE | .8798±.0079 | .8664±.0031 | .8584±.0051 | .8806±.0043 | .8675±.0035 | .8591±.0045 | .9296±.0064 | .9190±.0052 | .9110±.0049 |
| | RGIB-REP | .8846±.0071 | .8715±.0035 | .8640±.0043 | .8818±.0054 | .8716±.0021 | .8646±.0047 | .9161±.0081 | .9156±.0075 | .9129±.0095 |
| | RGIB-SSL | .8915±.0062 | .8737±.0029 | .8633±.0036 | .8891±.0057 | .8732±.0034 | .8639±.0051 | .9450±.0042 | .9488±.0065 | .9484±.0027 |

Table 37: Full results on Pubmed dataset with GAT.

| layers | method | bilateral noise 20% | 40% | 60% | input noise 20% | 40% | 60% | label noise 20% | 40% | 60% |
|---|---|---|---|---|---|---|---|---|---|---|
| | | | | | GAT | | | | | |
| L=2 | Standard | .9173±.0028 | .8984±.0030 | .8884±.0033 | .9255±.0024 | .9176±.0035 | .9102±.0030 | .9306±.0038 | .9271±.0030 | .9232±.0034 |
| | DropEdge | .9102±.0023 | .8970±.0032 | .8837±.0034 | .9208±.0030 | .9177±.0028 | .9087±.0037 | .9374±.0031 | .9309±.0020 | .9267±.0019 |
| | NeuralSparse | .9106±.0021 | .8952±.0031 | .8840±.0029 | .9179±.0029 | .9210±.0032 | .9141±.0038 | .9310±.0039 | .9297±.0026 | .9252±.0021 |
| | PTDNet | .9119±.0015 | .8943±.0030 | .8844±.0037 | .9068±.0027 | .9210±.0026 | .9143±.0025 | .9311±.0031 | .9280±.0034 | .9267±.0024 |
| | Co-teaching | .9163±.0044 | .8924±.0050 | .8881±.0100 | .9058±.0089 | .9180±.0089 | .9136±.0068 | .9556±.0098 | .9364±.0207 | .9376±.0249 |
| | Peer loss | .9156±.0115 | .8968±.0083 | .8823±.0041 | .9147±.0055 | .9171±.0089 | .9173±.0070 | .9407±.0225 | .9296±.0084 | .9219±.0229 |
| | Jaccard | .9128±.0057 | .8940±.0094 | .8831±.0086 | .9042±.0092 | .9181±.0175 | .9228±.0214 | .9304±.0039 | .9355±.0127 | .9282±.0109 |
| | GIB | .9136±.0061 | .8965±.0110 | .8881±.0115 | .9052±.0313 | .9208±.0097 | .9383±.0268 | .9366±.0069 | .9291±.0065 | .9277±.0089 |
| | SupCon | .9072±.0036 | .8881±.0037 | .8763±.0027 | .9076±.0025 | .9080±.0033 | .9002±.0037 | .9202±.0032 | .9123±.0040 | .9059±.0027 |
| | GRACE | .8230±.0512 | .7882±.0693 | .7914±.0767 | .8273±.0671 | .8053±.0498 | .7993±.0695 | .8792±.0342 | .8926±.0257 | .8946±.0211 |
| | RGIB-REP | .9190±.0025 | .9034±.0017 | .8939±.0056 | .9250±.0033 | .9164±.0024 | .9099±.0029 | .9311±.0036 | .9306±.0025 | .9276±.0022 |
| | RGIB-SSL | .9223±.0027 | .9054±.0032 | .8960±.0027 | .9183±.0021 | .9071±.0023 | .8977±.0029 | .9405±.0027 | .9407±.0022 | .9378±.0018 |
| L=4 | Standard | .8610±.0045 | .8434±.0042 | .8339±.0048 | .8668±.0040 | .8547±.0050 | .8476±.0037 | .8817±.0042 | .8772±.0040 | .8696±.0062 |
| | DropEdge | .8600±.0034 | .8462±.0028 | .8325±.0044 | .8605±.0041 | .8557±.0057 | .8454±.0029 | .8852±.0052 | .8851±.0055 | .8871±.0034 |
| | NeuralSparse | .8691±.0034 | .8505±.0025 | .8402±.0055 | .8612±.0042 | .8572±.0048 | .8459±.0038 | .8906±.0059 | .8806±.0076 | .8796±.0075 |
| | PTDNet | .8614±.0038 | .8568±.0026 | .8372±.0338 | .8640±.0035 | .8541±.0050 | .8562±.0027 | .8909±.0052 | .8855±.0054 | .8785±.0064 |
| | Co-teaching | .8688±.0125 | .8450±.0065 | .8413±.0081 | .8666±.0066 | .8556±.0103 | .8468±.0118 | .8792±.0107 | .8946±.0335 | .8968±.0132 |
| | Peer loss | .8613±.0040 | .8493±.0105 | .8342±.0108 | .8652±.0061 | .8504±.0143 | .8471±.0059 | .8851±.0110 | .8839±.0085 | .8735±.0069 |
| | Jaccard | .8630±.0089 | .8502±.0082 | .8365±.0136 | .8677±.0198 | .8495±.0065 | .8485±.0116 | .8863±.0078 | .8846±.0060 | .8750±.0119 |
| | GIB | .8616±.0036 | .8534±.0143 | .8370±.0116 | .8814±.0189 | .8486±.0059 | .8421±.0220 | .8828±.0111 | .8827±.0066 | .8700±.0076 |
| | SupCon | .8534±.0061 | .8292±.0128 | .8048±.0164 | .8625±.0085 | .8447±.0069 | .8360±.0141 | .8768±.0057 | .8686±.0062 | .8629±.0065 |
| | GRACE | .8355±.0477 | .8202±.0387 | .8046±.0409 | .8439±.0344 | .8078±.0649 | .7822±.0686 | .7878±.0839 | .8185±.0702 | .8008±.0794 |
| | RGIB-REP | .8656±.0043 | .8443±.0057 | .8339±.0033 | .8674±.0058 | .8528±.0043 | .8392±.0036 | .8909±.0084 | .8925±.0051 | .8833±.0062 |
| | RGIB-SSL | .8891±.0037 | .8651±.0036 | .8480±.0068 | .8878±.0039 | .8650±.0044 | .8489±.0055 | .9276±.0042 | .9277±.0022 | .9264±.0024 |
| L=6 | Standard | .8315±.0059 | .8116±.0073 | .8040±.0092 | .8374±.0036 | .8201±.0321 | .8067±.0306 | .8480±.0086 | .8414±.0116 | .8313±.0071 |
| | DropEdge | .8468±.0407 | .8236±.0050 | .8004±.0071 | .8655±.0037 | .8303±.0044 | .7915±.0334 | .8501±.0058 | .8568±.0068 | .8523±.0381 |
| | NeuralSparse | .8426±.0035 | .8234±.0032 | .8098±.0067 | .8323±.0342 | .8258±.0090 | .7977±.0295 | .8622±.0062 | .8563±.0091 | .8577±.0075 |
| | PTDNet | .8384±.0319 | .8117±.0384 | .8031±.0380 | .8508±.0044 | .8251±.0284 | .8204±.0034 | .8635±.0064 | .8498±.0251 | .8452±.0065 |
| | Co-teaching | .8360±.0122 | .8111±.0122 | .8081±.0113 | .8391±.0101 | .8246±.0398 | .8071±.0312 | .8729±.0090 | .8647±.0107 | .8424±.0079 |
| | Peer loss | .8354±.0128 | .8186±.0111 | .8044±.0104 | .8416±.0122 | .8277±.0359 | .8133±.0336 | .8504±.0132 | .8563±.0215 | .8310±.0089 |
| | Jaccard | .8333±.0140 | .8191±.0073 | .8044±.0082 | .8408±.0221 | .8340±.0424 | .8216±.0497 | .8528±.0100 | .8507±.0210 | .8304±.0108 |
| | GIB | .8316±.0125 | .8266±.0055 | .8064±.0241 | .8359±.0136 | .8205±.0345 | .8274±.0430 | .8551±.0149 | .8493±.0134 | .8361±.0072 |
| | SupCon | .8218±.0236 | .7763±.0306 | .7113±.0241 | .8137±.0821 | .8025±.0286 | .7555±.0297 | .8543±.0139 | .8390±.0370 | .8540±.0071 |
| | GRACE | .7773±.0737 | .7729±.0344 | .6821±.1217 | .7015±.1340 | .6953±.0805 | .6643±.1009 | .5212±.0262 | .5455±.0459 | .5743±.0549 |
| | RGIB-REP | .8324±.0039 | .8162±.0088 | .7934±.0068 | .8248±.0070 | .8145±.0088 | .7892±.0107 | .8518±.0056 | .8496±.0083 | .8492±.0113 |
| | RGIB-SSL | .8738±.0051 | .8443±.0033 | .8124±.0084 | .8709±.0023 | .8432±.0046 | .8187±.0050 | .9208±.0043 | .9250±.0029 | .9235±.0031 |

Table 38: Full results on Pubmed dataset with SAGE.

| | | SAGE | | | | | | | | |
|---|---|---|---|---|---|---|---|---|---|---|
| layers | method | bilateral noise | | | input noise | | | label noise | | |
| | | 20% | 40% | 60% | 20% | 40% | 60% | 20% | 40% | 60% |
| L=2 | Standard | .9136±.0032 | .9094±.0035 | .9035±.0040 | .9295±.0025 | .9378±.0022 | .9410±.0023 | .8817±.0034 | .8793±.0043 | .8757±.0041 |
| | DropEdge | .9142±.0026 | .9188±.0029 | .9014±.0038 | .9179±.0020 | .9178±.0013 | .9165±.0013 | .9021±.0021 | .9003±.0018 | .9011±.0020 |
| | NeuralSparse | .9118±.0030 | .9137±.0031 | .9094±.0034 | .9271±.0019 | .9209±.0026 | .9234±.0017 | .9033±.0028 | .8925±.0030 | .8864±.0032 |
| | PTDNet | .9114±.0026 | .9138±.0030 | .9104±.0037 | .9267±.0020 | .9211±.0027 | .9229±.0024 | .9030±.0028 | .8941±.0040 | .8868±.0033 |
| | Co-teaching | .9164±.0031 | .9160±.0027 | .9092±.0132 | .9362±.0061 | .9258±.0072 | .9228±.0017 | .8827±.0026 | .9054±.0236 | .8856±.0113 |
| | Peer loss | .9103±.0104 | .9174±.0094 | .9044±.0087 | .9308±.0079 | .9238±.0051 | .9307±.0076 | .9005±.0118 | .8833±.0066 | .8909±.0058 |
| | Jaccard | .9108±.0123 | .9167±.0105 | .9109±.0031 | .9236±.0098 | .9259±.0175 | .9308±.0141 | .8861±.0052 | .8785±.0092 | .8842±.0048 |
| | GIB | .9173±.0049 | .9125±.0211 | .9096±.0020 | .9338±.0152 | .9294±.0125 | .9299±.0194 | .8880±.0125 | .8865±.0129 | .8853±.0058 |
| | SupCon | .9167±.0026 | .9148±.0031 | .9106±.0035 | .9259±.0034 | .9305±.0031 | .9230±.0033 | .9033±.0055 | .8929±.0051 | .8888±.0034 |
| | GRACE | .8317±.0063 | .8291±.0059 | .8336±.0068 | .8288±.0071 | .8344±.0074 | .8305±.0052 | .8320±.0046 | .8344±.0078 | .8328±.0075 |
| | RGIB-REP | .9192±.0022 | .9136±.0027 | .9119±.0037 | .9278±.0031 | .9322±.0031 | .9331±.0034 | .9055±.0033 | .9015±.0051 | .9000±.0043 |
| | RGIB-SSL | .9276±.0052 | .9285±.0030 | .9313±.0031 | .9273±.0056 | .9305±.0040 | .9334±.0025 | .9304±.0017 | .9294±.0049 | .9274±.0066 |
| L=4 | Standard | .8627±.0056 | .8663±.0061 | .8619±.0073 | .8715±.0080 | .8901±.0082 | .9033±.0057 | .8386±.0085 | .8330±.0104 | .8268±.0092 |
| | DropEdge | .8737±.0060 | .8740±.0065 | .8816±.0077 | .9072±.0120 | .9016±.0078 | .9087±.0019 | .9037±.0024 | .9036±.0048 | .9048±.0059 |
| | NeuralSparse | .8888±.0037 | .8825±.0046 | .8822±.0051 | .8982±.0067 | .9146±.0029 | .9190±.0041 | .8647±.0075 | .8535±.0091 | .8472±.0079 |
| | PTDNet | .8842±.0045 | .8853±.0059 | .8846±.0052 | .8885±.0055 | .9057±.0055 | .9127±.0058 | .8464±.0105 | .8458±.0066 | .8408±.0065 |
| | Co-teaching | .8678±.0104 | .8752±.0090 | .8667±.0122 | .8727±.0079 | .8971±.0159 | .9061±.0098 | .8601±.0240 | .8498±.0368 | .8514±.0388 |
| | Peer loss | .8683±.0136 | .8717±.0100 | .8665±.0117 | .8736±.0075 | .8997±.0079 | .9095±.0119 | .8510±.0151 | .8514±.0115 | .8423±.0128 |
| | Jaccard | .8646±.0124 | .8725±.0141 | .8625±.0140 | .8773±.0125 | .9017±.0205 | .9155±.0164 | .8378±.0137 | .8383±.0118 | .8275±.0083 |
| | GIB | .8816±.0203 | .8768±.0098 | .8641±.0238 | .8824±.0102 | .8918±.0095 | .9221±.0243 | .8479±.0103 | .8410±.0151 | .8330±.0120 |
| | SupCon | .8765±.0086 | .8682±.0100 | .8518±.0198 | .8907±.0054 | .8957±.0104 | .8879±.0113 | .8865±.0046 | .8811±.0052 | .8810±.0044 |
| | GRACE | .7142±.0451 | .6741±.0484 | .7227±.0446 | .6578±.0701 | .6816±.0536 | .7012±.0516 | .6498±.0896 | .6647±.0535 | .6540±.0424 |
| | RGIB-REP | .8746±.0049 | .8708±.0064 | .8758±.0071 | .8800±.0045 | .8904±.0048 | .8941±.0051 | .8504±.0078 | .8466±.0070 | .8533±.0118 |
| | RGIB-SSL | .9315±.0036 | .9237±.0035 | .9214±.0033 | .9276±.0046 | .9310±.0039 | .9286±.0035 | .9463±.0030 | .9448±.0037 | .9430±.0021 |
| L=6 | Standard | .8314±.0081 | .8105±.0279 | .8333±.0089 | .8224±.0335 | .8538±.0105 | .8566±.0199 | .8242±.0104 | .8161±.0071 | .8090±.0129 |
| | DropEdge | .8781±.0041 | .8665±.0076 | .8556±.0081 | .8800±.0050 | .8731±.0122 | .8628±.0152 | .8503±.0070 | .8698±.0074 | .8426±.0096 |
| | NeuralSparse | .8611±.0101 | .8589±.0081 | .8482±.0182 | .8694±.0083 | .8858±.0068 | .8889±.0057 | .8368±.0074 | .8344±.0090 | .8353±.0081 |
| | PTDNet | .8425±.0073 | .8511±.0067 | .8525±.0064 | .8430±.0107 | .8627±.0115 | .8739±.0103 | .8232±.0076 | .8195±.0071 | .8198±.0060 |
| | Co-teaching | .8385±.0114 | .8202±.0303 | .8349±.0147 | .8265±.0385 | .8607±.0164 | .8612±.0190 | .8299±.0352 | .8243±.0074 | .8143±.0409 |
| | Peer loss | .8339±.0162 | .8111±.0285 | .8335±.0084 | .8317±.0412 | .8610±.0135 | .8605±.0223 | .8277±.0224 | .8260±.0197 | .8204±.0316 |
| | Jaccard | .8355±.0110 | .8180±.0272 | .8347±.0124 | .8298±.0436 | .8564±.0107 | .8607±.0394 | .8260±.0138 | .8207±.0130 | .8140±.0192 |
| | GIB | .8309±.0062 | .8195±.0329 | .8336±.0206 | .8226±.0355 | .8527±.0208 | .8647±.0344 | .8310±.0142 | .8237±.0128 | .8141±.0165 |
| | SupCon | .8288±.0109 | .8276±.0351 | .8174±.0070 | .8617±.0122 | .8443±.0528 | .8365±.0350 | .8745±.0071 | .8692±.0132 | .8661±.0076 |
| | GRACE | .7293±.0133 | .7151±.0368 | .6737±.0522 | .7102±.0296 | .7392±.0257 | .6751±.0444 | .7120±.0166 | .7042±.0588 | .7230±.0286 |
| | RGIB-REP | .8493±.0057 | .8424±.0083 | .8373±.0079 | .8439±.0063 | .8527±.0092 | .8536±.0112 | .8275±.0218 | .8148±.0098 | .8211±.0145 |
| | RGIB-SSL | .8915±.0060 | .8971±.0051 | .8956±.0046 | .8948±.0079 | .9026±.0042 | .8995±.0040 | .9313±.0042 | .9311±.0036 | .9294±.0053 |

Table 39: Full results on Facebook dataset with GCN.

| | | GCN | | | | | | | | |
|---|---|---|---|---|---|---|---|---|---|---|
| layers | method | bilateral noise | | | input noise | | | label noise | | |
| | | 20% | 40% | 60% | 20% | 40% | 60% | 20% | 40% | 60% |
| L=2 | Standard | .9880±.0007 | .9866±.0007 | .9843±.0010 | .9878±.0008 | .9852±.0006 | .9834±.0011 | .9892±.0006 | .9888±.0008 | .9888±.0007 |
| | DropEdge | .9865±.0007 | .9845±.0008 | .9821±.0009 | .9764±.0010 | .9835±.0007 | .9708±.0012 | .9783±.0006 | .9879±.0008 | .9881±.0007 |
| | NeuralSparse | .9877±.0007 | .9861±.0008 | .9837±.0009 | .9772±.0007 | .9852±.0007 | .9732±.0010 | .9788±.0006 | .9885±.0007 | .9888±.0005 |
| | PTDNet | .9876±.0008 | .9862±.0008 | .9838±.0008 | .9776±.0007 | .9848±.0008 | .9728±.0011 | .9859±.0006 | .9786±.0009 | .9886±.0006 |
| | Co-teaching | .9871±.0052 | .9722±.0104 | .9876±.0009 | .9738±.0018 | .9848±.0077 | .9750±.0002 | .9716±.0266 | .9873±.0121 | .9825±.0009 |
| | Peer loss | .9755±.0095 | .9737±.0033 | .9895±.0096 | .9797±.0024 | .9855±.0051 | .9720±.0019 | .9763±.0001 | .9732±.0110 | .9774±.0034 |
| | Jaccard | .9759±.0043 | .9796±.0056 | .9707±.0006 | .9739±.0098 | .9807±.0040 | .9793±.0150 | .9729±.0030 | .9773±.0044 | .9729±.0042 |
| | GIB | .9752±.0009 | .9735±0.021 | .9720±.0154 | .9796±.0245 | .9820±.0173 | .9707±.0024 | .9724±.0017 | .9774±.1404 | .9717±.0086 |
| | SupCon | .9779±.0008 | .9748±.0014 | .9665±.0022 | .9724±.0009 | .9803±.0011 | .9783±.0010 | .9713±.0060 | .9569±.0160 | .9701±.0024 |
| | GRACE | .8883±.0120 | .8811±.0077 | .8412±.0224 | .8950±.0081 | .8792±.0119 | .8530±.0221 | .8864±.0077 | .8826±.0051 | .8784±.0104 |
| | RGIB-REP | .9881±.0007 | .9866±.0008 | .9846±.0009 | .9880±.0007 | .9854±.0005 | .9841±.0011 | .9895±.0007 | .9891±.0008 | .9894±.0008 |
| | RGIB-SSL | .9845±.0005 | .9810±.0005 | .9765±.0014 | .9840±.0011 | .9808±.0012 | .9787±.0009 | .9862±.0005 | .9859±.0010 | .9866±.0007 |
| L=4 | Standard | .9829±.0020 | .9520±.0424 | .9438±.0402 | .9819±.0015 | .9668±.0147 | .9622±.0154 | .9882±.0007 | .9880±.0007 | .9886±.0006 |
| | DropEdge | .9811±.0028 | .9682±.0096 | .9473±.0120 | .9803±.0016 | .9685±.0033 | .9531±.0112 | .9673±.0008 | .9771±.0008 | .9776±.0009 |
| | NeuralSparse | .9825±.0020 | .9638±.0039 | .9456±.0067 | .9712±.0022 | .9691±.0045 | .9583±.0071 | .9781±.0007 | .9781±.0008 | .9784±.0008 |
| | PTDNet | .9725±.0017 | .9674±.0131 | .9485±.0094 | .9725±.0014 | .9668±.0089 | .9493±.0267 | .9879±.0003 | .9880±.0008 | .9783±.0008 |
| | Co-teaching | .9820±.0113 | .9526±.0439 | .9480±.0450 | .9712±.0084 | .9707±.0141 | .9714±.0146 | .9762±.0070 | .9797±.0167 | .9638±.0154 |
| | Peer loss | .9807±.0016 | .9536±.0503 | .9430±.0437 | .9758±.0011 | .9703±.0204 | .9622±.0176 | .9769±.0085 | .9750±.0128 | .9734±.0137 |
| | Jaccard | .9794±.0031 | .9579±.0471 | .9428±.0452 | .9784±.0127 | .9702±.0295 | .9638±.0340 | .9702±.0016 | .9725±.0064 | .9758±.0035 |
| | GIB | .9773±.0182 | .9608±.0548 | .9417±.0441 | .9796±.0260 | .9647±.0200 | .9650±.0373 | .9742±.0073 | .9703±.0023 | .9771±.0030 |
| | SupCon | .9588±.0067 | .9508±.0100 | .9297±.0121 | .9647±.0088 | .9517±.0113 | .9401±.0135 | .9647±.0164 | .9567±.0149 | .9553±.0133 |
| | GRACE | .8899±.0100 | .8865±.0226 | .8315±.0108 | .9015±.0049 | .8833±.0285 | .8395±.0157 | .8913±.0036 | .8972±.0046 | .8887±.0123 |
| | RGIB-REP | .9832±.0026 | .9770±.0032 | .9519±.0063 | .9833±.0014 | .9723±.0062 | .9682±.0087 | .9884±.0006 | .9883±.0007 | .9889±.0007 |
| | RGIB-SSL | .9829±.0023 | .9711±.0025 | .9643±.0029 | .9821±.0019 | .9707±.0021 | .9668±.0040 | .9857±.0010 | .9881±.0008 | .9857±.0007 |
| L=6 | Standard | .9798±.0013 | .9609±.0138 | .9368±.0179 | .9764±.0034 | .9502±.0096 | .9469±.0160 | .9863±.0013 | .9865±.0012 | .9876±.0010 |
| | DropEdge | .9774±.0030 | .9635±.0063 | .9500±.0103 | .9540±.0034 | .9579±.0083 | .9411±.0084 | .9556±.0014 | .9642±.0006 | .9559±.0012 |
| | NeuralSparse | .9774±.0020 | .9596±.0067 | .9435±.0114 | .9449±.0029 | .9496±.0073 | .9490±.0087 | .9564±.0012 | .9562±.0011 | .9465±.0016 |
| | PTDNet | .9767±.0066 | .9606±.0131 | .9449±.0175 | .9555±.0034 | .9568±.0064 | .9416±.0145 | .9560±.0011 | .9460±.0014 | .9569±.0009 |
| | Co-teaching | .9638±.0051 | .9565±.0235 | .9426±.0261 | .9520±.0051 | .9568±.0148 | .9489±.0228 | .9474±.0068 | .9458±.0143 | .9561±.0169 |
| | Peer loss | .9699±.0045 | .9557±.0141 | .9362±.0221 | .9554±.0053 | .9494±.0135 | .9445±.0234 | .9498±.0063 | .9581±.0200 | .9650±.0021 |
| | Jaccard | .9641±.0032 | .9553±.0137 | .9425±.0178 | .9507±.0064 | .9616±.0125 | .9480±.0170 | .9528±.0060 | .9525±.0075 | .9423±.0057 |
| | GIB | .9649±.0145 | .9602±.0198 | .9416±.0236 | .9511±.0333 | .9552±.0261 | .9555±.0346 | .9556±.0099 | .9430±.0106 | .9578±.0052 |
| | SupCon | .9462±.0050 | .9377±.0054 | .9282±.0113 | .9460±.0073 | .9355±.0118 | .9253±.0110 | .9411±.0088 | .9550±.0087 | .9510±.0122 |
| | GRACE | .8939±.0118 | .8563±.0206 | .8259±.0019 | .9071±.0077 | .8582±.0228 | .8274±.0029 | .8962±.0093 | .9026±.0035 | .8977±.0100 |
| | RGIB-REP | .9774±.0031 | .9680±.0097 | .9501±.0134 | .9780±.0024 | .9619±.0069 | .9423±.0085 | .9868±.0008 | .9863±.0010 | .9872±.0011 |
| | RGIB-SSL | .9751±.0023 | .9641±.0037 | .9395±.0122 | .9763±.0034 | .9633±.0040 | .9451±.0084 | .9845±.0008 | .9843±.0012 | .9847±.0011 |

Table 40: Full results on Facebook dataset with GAT.

| | | GAT | | | | | | | | |
|---|---|---|---|---|---|---|---|---|---|---|
| | | bilateral noise | | | input noise | | | label noise | | |
| layers | method | 20% | 40% | 60% | 20% | 40% | 60% | 20% | 40% | 60% |
| L=2 | Standard | .9869±.0005 | .9856±.0008 | .9836±.0007 | .9878±.0006 | .9861±.0006 | .9858±.0008 | .9872±.0006 | .9864±.0009 | .9857±.0008 |
| | DropEdge | .9645±.0007 | .9532±.0010 | .9517±.0009 | .9652±.0007 | .9636±.0007 | .9733±.0008 | .9755±.0007 | .9743±.0010 | .9739±.0012 |
| | NeuralSparse | .9661±.0007 | .9651±.0009 | .9638±.0006 | .9765±.0008 | .9652±.0007 | .9744±.0006 | .9662±.0010 | .9657±.0006 | .9656±.0007 |
| | PTDNet | .9762±.0007 | .9551±.0008 | .9735±.0007 | .9766±.0007 | .9650±.0007 | .9645±.0009 | .9766±.0006 | .9759±.0009 | .9757±.0008 |
| | Co-teaching | .9723±.0006 | .9685±.0054 | .9762±.0039 | .9632±.0057 | .9720±.0001 | .9741±.0046 | .9703±0.001 | .9789±.0232 | .9792±.0288 |
| | Peer loss | .9762±.0032 | .9644±.0020 | .9725±.0076 | .9699±.0020 | .9720±.0104 | .9708±.0103 | .9632±.0194 | .9807±.0207 | .9713±.0078 |
| | Jaccard | .9771±.0038 | .9661±.0057 | .9736±.0078 | .9556±.0064 | .9758±.0180 | .9780±.0012 | .9720±.0100 | .9749±.0064 | .9701±.0087 |
| | GIB | .9879±.0075 | .9616±.0007 | .9709±.0199 | .9564±.0122 | .9746±.0004 | .9645±.0225 | .9758±.0101 | .9738±.0019 | .9799±.0094 |
| | SupCon | .9718±.0028 | .9646±.0085 | .9659±.0087 | .9753±.0008 | .9737±.0009 | .9629±.0009 | .9720±.0038 | .9456±.0038 | .9419±.0029 |
| | GRACE | .8377±.0309 | .8716±.0263 | .8519±.0264 | .8343±.0341 | .8502±.0271 | .8487±.0407 | .8768±.0457 | .8370±.0416 | .8794±.0319 |
| | RGIB-REP | .9865±.0006 | .9854±.0009 | .9837±.0006 | .9872±.0007 | .9855±.0007 | .9855±.0008 | .9871±.0007 | .9861±.0009 | .9859±.0006 |
| | RGIB-SSL | .9788±.0011 | .9764±.0017 | .9717±.0021 | .9784±.0012 | .9764±.0015 | .9734±.0021 | .9778±.0008 | .9784±.0009 | .9791±.0009 |
| L=4 | Standard | .9857±.0015 | .9850±.0012 | .9820±.0014 | .9855±.0011 | .9827±.0019 | .9773±.0046 | .9873±.0010 | .9874±.0010 | .9874±.0005 |
| | DropEdge | .9839±.0009 | .9791±.0032 | .9767±.0024 | .9836±.0009 | .9772±.0022 | .9705±.0030 | .9852±.0005 | .9845±.0009 | .9844±.0007 |
| | NeuralSparse | .9854±.0009 | .9812±.0019 | .9768±.0027 | .9839±.0012 | .9790±.0028 | .9736±.0030 | .9872±.0009 | .9869±.0009 | .9868±.0007 |
| | PTDNet | .9843±.0039 | .9830±.0018 | .9745±.0076 | .9844±.0012 | .9787±.0039 | .9752±.0027 | .9872±.0005 | .9868±.0010 | .9872±.0007 |
| | Co-teaching | .9789±.0039 | .9777±.0086 | .9717±.0111 | .9703±.0050 | .9718±.0071 | .9782±.0049 | .9645±.0105 | .9730±.0267 | .9646±.0014 |
| | Peer loss | .9944±.0029 | .9630±.0105 | .9687±.0046 | .9610±.0068 | .9610±.0064 | .9783±.0073 | .9677±.0053 | .9638±.0050 | .9691±.0203 |
| | Jaccard | .9757±.0079 | .9863±.0087 | .9883±.0103 | .9607±.0182 | .9616±.0184 | .9655±.0151 | .9689±.0025 | .9625±.0009 | .9604±.0036 |
| | GIB | .9809±.0134 | .9747±.0077 | .9709±.0204 | .9721±.0297 | .9741±.0199 | .9740±.0111 | .9747±.0038 | .9768±.0093 | .9799±.0016 |
| | SupCon | .9709±.0156 | .9464±.0254 | .9134±.0309 | .9824±.0008 | .9745±.0062 | .9677±.0113 | .9600±.0154 | .9580±.0183 | .9482±.0174 |
| | GRACE | .7803±.0649 | .7947±.0600 | .7618±.1058 | .8259±.0669 | .7588±.0963 | .7260±.1331 | .8440±.0370 | .8448±.0337 | .8122±.0403 |
| | RGIB-REP | .9855±.0014 | .9839±.0016 | .9783±.0034 | .9838±.0019 | .9806±.0013 | .9765±.0037 | .9872±.0008 | .9871±.0008 | .9867±.0007 |
| | RGIB-SSL | .9751±.0033 | .9707±.0061 | .9602±.0043 | .9767±.0030 | .9683±.0053 | .9603±.0062 | .9755±.0028 | .9772±.0032 | .9762±.0047 |
| L=6 | Standard | .9805±.0025 | .9658±.0321 | .9577±.0314 | .9804±.0018 | .9738±.0044 | .9710±.0036 | .9854±.0015 | .9860±.0007 | .9867±.0012 |
| | DropEdge | .9643±.0041 | .9638±.0054 | .9500±.0274 | .9736±.0042 | .9589±.0089 | .9606±.0055 | .9735±.0008 | .9726±.0012 | .9736±.0009 |
| | NeuralSparse | .9746±.0077 | .9667±.0092 | .9560±.0122 | .9740±.0047 | .9668±.0056 | .9652±.0043 | .9735±.0017 | .9742±.0018 | .9733±.0030 |
| | PTDNet | .9769±.0037 | .9679±.0071 | .9582±.0296 | .9754±.0027 | .9688±.0045 | .9645±.0039 | .9738±.0041 | .9741±.0022 | .9756±.0012 |
| | Co-teaching | .9745±.0034 | .9686±.0316 | .9625±.0408 | .9747±.0084 | .9833±.0108 | .9738±.0035 | .9622±.0024 | .9676±.0072 | .9721±.0215 |
| | Peer loss | .9690±.0114 | .9728±.0334 | .9588±.0309 | .9716±.0028 | .9776±.0075 | .9766±.0074 | .9871±.0207 | .9605±.0202 | .9768±.0063 |
| | Jaccard | .9789±.0084 | .9651±.0360 | .9620±.0337 | .9673±.0007 | .9809±.0110 | .9731±.0021 | .9652±.0064 | .9607±.0004 | .9751±.0030 |
| | GIB | .9839±.0037 | .9777±.0420 | .9584±.0374 | .9868±.0290 | .9760±.0135 | .9791±.0249 | .9648±.0054 | .9640±.0046 | .9774±.0029 |
| | SupCon | .9428±.0194 | .9097±.0361 | .8779±.0347 | .9653±.0171 | .9459±.0378 | .9312±.0344 | .9465±.0173 | .9484±.0194 | .9023±.0333 |
| | GRACE | .7234±.0687 | .6604±.1398 | .6075±.1089 | .7415±.1242 | .6176±.1345 | .6216±.1341 | .7613±.1157 | .7256±.1351 | .7062±.1042 |
| | RGIB-REP | .9779±.0031 | .9711±.0042 | .9433±.0392 | .9753±.0041 | .9666±.0087 | .9614±.0057 | .9847±.0017 | .9848±.0013 | .9856±.0009 |
| | RGIB-SSL | .9653±.0107 | .9491±.0105 | .9254±.0206 | .9676±.0057 | .9502±.0087 | .9384±.0067 | .9737±.0065 | .9675±.0052 | .9713±.0103 |

Table 41: Full results on Facebook dataset with SAGE.

| | | SAGE | | | | | | | | |
|---|---|---|---|---|---|---|---|---|---|---|
| | | bilateral noise | | | input noise | | | label noise | | |
| layers | method | 20% | 40% | 60% | 20% | 40% | 60% | 20% | 40% | 60% |
| L=2 | Standard | .9858±.0008 | .9827±.0008 | .9788±.0012 | .9881±.0008 | .9862±.0007 | .9862±.0008 | .9867±.0006 | .9840±.0006 | .9825±.0014 |
| | DropEdge | .9745±.0008 | .9807±.0011 | .9765±.0011 | .9766±.0007 | .9749±.0011 | .9752±.0008 | .9761±.0008 | .9739±.0010 | .9719±.0011 |
| | NeuralSparse | .9753±.0008 | .9619±.0011 | .9683±.0007 | .9675±.0008 | .9660±.0006 | .9659±.0011 | .9667±.0007 | .9640±.0007 | .9627±.0011 |
| | PTDNet | .9755±.0007 | .9723±.0009 | .9779±.0012 | .9775±.0007 | .9761±.0008 | .9754±.0007 | .9766±.0007 | .9743±.0007 | .9726±.0014 |
| | Co-teaching | .9759±.0101 | .9722±.0004 | .9728±.0100 | .9702±.0050 | .9742±.0023 | .9739±.0081 | .9764±.0125 | .9788±.0205 | .9700±.0282 |
| | Peer loss | .9765±.0052 | .9720±.0072 | .9790±.0103 | .9787±.0005 | .9781±.0016 | .9787±.0075 | .9721±.0019 | .9842±.0136 | .9793±.0031 |
| | Jaccard | .9747±.0018 | .9720±.0009 | .9721±.0096 | .9729±.0166 | .9760±.0139 | .9752±.0102 | .9733±.0053 | .9706±.0059 | .9716±.0098 |
| | GIB | .9798±.0089 | .9780±.0040 | .9765±.0020 | .9774±.0139 | .9767±.0287 | .9778±.0006 | .9780±.0012 | .9713±.0067 | .9772±.0050 |
| | SupCon | .9482±.0166 | .9288±.0046 | .9141±.0139 | .9793±.0043 | .9782±.0072 | .9786±.0078 | .9278±.0048 | .9252±.0074 | .9156±.0041 |
| | GRACE | .9113±.0116 | .8907±.0295 | .9077±.0089 | .9012±.0113 | .8885±.0177 | .9031±.0104 | .9053±.0126 | .9050±.0173 | .9067±.0137 |
| | RGIB-REP | .9860±.0009 | .9834±.0008 | .9802±.0014 | .9878±.0009 | .9863±.0007 | .9863±.0011 | .9875±.0006 | .9860±.0007 | .9855±.0011 |
| | RGIB-SSL | .9840±.0007 | .9829±.0010 | .9811±.0006 | .9838±.0007 | .9821±.0008 | .9823±.0004 | .9841±.0008 | .9839±.0007 | .9844±.0010 |
| L=4 | Standard | .9824±.0015 | .9783±.0025 | .9698±.0040 | .9849±.0015 | .9815±.0024 | .9815±.0025 | .9844±.0015 | .9817±.0007 | .9809±.0009 |
| | DropEdge | .9818±.0026 | .9764±.0034 | .9722±.0022 | .9625±.0015 | .9601±.0043 | .9600±.0023 | .9649±.0008 | .9627±.0017 | .9615±.0012 |
| | NeuralSparse | .9811±.0031 | .9776±.0021 | .9740±.0031 | .9636±.0023 | .9600±.0026 | .9615±.0016 | .9648±.0008 | .9628±.0014 | .9613±.0025 |
| | PTDNet | .9823±.0012 | .9757±.0041 | .9735±.0032 | .9623±.0041 | .9609±.0031 | .9607±.0022 | .9652±.0007 | .9632±.0009 | .9623±.0024 |
| | Co-teaching | .9660±.0009 | .9653±.0075 | .9659±.0114 | .9600±.0098 | .9634±.0091 | .9626±.0079 | .9654±.0253 | .9706±.0136 | .9628±.0263 |
| | Peer loss | .9668±.0105 | .9676±.0064 | .9716±.0030 | .9681±.0079 | .9641±.0044 | .9603±.0112 | .9671±.0204 | .9609±.0088 | .9660±.0032 |
| | Jaccard | .9621±.0019 | .9715±.0056 | .9772±.0068 | .9662±.0143 | .9675±.0094 | .9782±.0056 | .9725±.0107 | .9726±.0063 | .9705±9.276 |
| | GIB | .9774±0.000 | .9758±.0200 | .9732±.0025 | .9708±.0195 | .9626±.0286 | .9679±.0171 | .9765±.0101 | .9714±.0018 | .9710±.0107 |
| | SupCon | .9169±.0144 | .9096±.0191 | .9001±.0143 | .9324±.0208 | .9188±.0100 | .9185±.0192 | .9295±.0114 | .9328±.0158 | .9325±.0148 |
| | GRACE | .8683±.0292 | .8355±.0696 | .8545±.0415 | .8585±.0424 | .8721±.0261 | .8413±.0364 | .8108±.0213 | .8527±.0375 | .8376±.0330 |
| | RGIB-REP | .9828±.0016 | .9804±.0020 | .9750±.0024 | .9843±.0014 | .9811±.0030 | .9808±.0030 | .9857±.0009 | .9833±.0025 | .9830±.0019 |
| | RGIB-SSL | .9817±.0016 | .9797±.0014 | .9768±.0009 | .9828±.0018 | .9798±.0014 | .9797±.0013 | .9821±.0009 | .9820±.0011 | .9826±.0010 |
| L=6 | Standard | .9751±.0055 | .9603±.0365 | .9616±.0091 | .9715±.0077 | .9672±.0057 | .9517±.0228 | .9761±.0136 | .9788±.0045 | .9749±.0151 |
| | DropEdge | .9652±.0039 | .9676±.0088 | .9578±.0149 | .9734±.0050 | .9616±.0126 | .9665±.0073 | .9631±.0022 | .9606±.0046 | .9651±.0145 |
| | NeuralSparse | .9640±.0075 | .9577±.0222 | .9672±.0061 | .9647±.0048 | .9694±.0156 | .9673±.0036 | .9632±.0017 | .9618±.0037 | .9615±.0051 |
| | PTDNet | .9668±.0029 | .9633±.0142 | .9617±.0111 | .9774±.0028 | .9693±.0053 | .9642±.0055 | .9629±.0035 | .9736±.0010 | .9708±.0037 |
| | Co-teaching | .9644±.0116 | .9699±.0463 | .9658±.0182 | .9730±.0107 | .9738±.0076 | .9583±.0239 | .9742±.0138 | .9744±.0106 | .9720±.0202 |
| | Peer loss | .9644±.0141 | .9602±.0369 | .9681±.0085 | .9613±.0167 | .9728±.0053 | .9524±.0288 | .9655±.0220 | .9769±.0028 | .9650±.0256 |
| | Jaccard | .9750±.0119 | .9666±.0453 | .9686±.0094 | .9666±.0261 | .9612±.0056 | .9683±.0383 | .9634±.0170 | .9784±.0110 | .9781±.0178 |
| | GIB | .9739±.0197 | .9656±.0564 | .9749±.0125 | .9743±.0351 | .9717±.0034 | .9615±.0291 | .9607±.0153 | .9779±.0059 | .9743±.0222 |
| | SupCon | .9007±.0112 | .8859±.0189 | .8730±.0140 | .8997±.0142 | .8940±.0104 | .9006±.0232 | .9146±.0114 | .9166±.0143 | .9185±.0122 |
| | GRACE | .8202±.0365 | .7225±.0840 | .7901±.1065 | .8018±.0575 | .8224±.0547 | .7960±.0915 | .8061±.0367 | .8262±.0306 | .8103±.0351 |
| | RGIB-REP | .9770±.0040 | .9557±.0119 | .9574±.0149 | .9749±.0046 | .9683±.0072 | .9584±.0093 | .9838±.0020 | .9829±.0015 | .9814±.0029 |
| | RGIB-SSL | .9773±.0032 | .9720±.0031 | .9678±.0030 | .9781±.0027 | .9739±.0023 | .9692±.0050 | .9774±.0016 | .9786±.0012 | .9786±.0017 |

Table 42: Full results on Chameleon dataset with GCN.

| | | GCN | | | | | | | | |
|---|---|---|---|---|---|---|---|---|---|---|
| layers | method | bilateral noise | | | input noise | | | label noise | | |
| | | 20% | 40% | 60% | 20% | 40% | 60% | 20% | 40% | 60% |
| L=2 | Standard | .9753±.0025 | .9696±.0022 | .9657±.0029 | .9784±.0017 | .9762±.0016 | .9754±.0023 | .9775±.0018 | .9769±.0018 | .9755±.0036 |
| | DropEdge | .9716±.0030 | .9640±.0025 | .9579±.0029 | .9646±.0026 | .9602±.0024 | .9661±.0024 | .9686±.0018 | .9664±.0029 | .9651±.0031 |
| | NeuralSparse | .9741±.0032 | .9692±.0030 | .9633±.0030 | .9682±.0026 | .9646±.0023 | .9627±.0022 | .9679±.0017 | .9669±.0029 | .9651±.0039 |
| | PTDNet | .9748±.0026 | .9682±.0032 | .9637±.0031 | .9679±.0021 | .9649±.0019 | .9627±.0025 | .9690±.0014 | .9674±.0021 | .9648±.0033 |
| | Co-teaching | .9679±.0059 | .9641±.0086 | .9676±.0088 | .9767±.0096 | .9712±.0084 | .9709±.0106 | .9790±.0218 | .9729±.0015 | .9745±.0272 |
| | Peer loss | .9647±.0057 | .9662±.0106 | .9658±.0059 | .9781±.0043 | .9784±.0014 | .9759±.0095 | .9762±.0159 | .9695±.0090 | .9736±.0036 |
| | Jaccard | .9684±.0103 | .9657±.0118 | .9660±.0030 | .9611±.0190 | .9799±.0005 | .9774±.0128 | .9711±.0010 | .9754±.0091 | .9750±.0086 |
| | GIB | .9649±.0052 | .9711±.0050 | .9634±.0177 | .9709±.0106 | .9714±.0257 | .9786±.0012 | .9674±.0066 | .9660±.0070 | .9778±.0089 |
| | SupCon | .9625±.0024 | .9677±.0024 | .9613±.0032 | .9757±.0018 | .9718±.0023 | .9697±.0034 | .9749±.0013 | .9755±.0024 | .9750±.0030 |
| | GRACE | .9145±.0105 | .8978±.0110 | .8915±.0033 | .9081±.0121 | .9005±.0062 | .8931±.0042 | .9171±.0042 | .9184±.0054 | .9196±.0058 |
| | RGIB-REP | .9754±.0026 | .9698±.0026 | .9652±.0027 | .9797±.0023 | .9757±.0016 | .9744±.0028 | .9804±.0023 | .9798±.0016 | .9782±.0028 |
| | RGIB-SSL | .9719±.0023 | .9658±.0025 | .9595±.0026 | .9709±.0024 | .9664±.0024 | .9638±.0025 | .9786±.0016 | .9797±.0022 | .9789±.0034 |
| L=4 | Standard | .9616±.0033 | .9496±.0190 | .9274±.0276 | .9608±.0038 | .9433±.0261 | .9368±.0271 | .9686±.0020 | .9580±.0021 | .9362±.0035 |
| | DropEdge | .9568±.0044 | .9548±.0058 | .9407±.0050 | .9567±.0039 | .9433±.0055 | .9432±.0088 | .9580±.0020 | .9579±.0033 | .9578±.0033 |
| | NeuralSparse | .9599±.0026 | .9497±.0032 | .9402±.0057 | .9609±.0027 | .9540±.0178 | .9348±.0169 | .9583±.0017 | .9583±.0030 | .9571±.0031 |
| | PTDNet | .9607±.0030 | .9514±.0036 | .9424±.0097 | .9610±.0024 | .9457±.0194 | .9360±.0090 | .9585±.0022 | .9576±.0030 | .9665±.0035 |
| | Co-teaching | .9595±.0112 | .9516±.0199 | .9483±.0374 | .9524±.0123 | .9446±.0288 | .9447±.0286 | .9642±.0079 | .9650±.0058 | .9533±.0050 |
| | Peer loss | .9543±.0090 | .9533±.0192 | .9267±.0320 | .9558±.0037 | .9482±.0280 | .9412±.0269 | .9621±.0161 | .9501±.0055 | .9569±.0227 |
| | Jaccard | .9503±.0061 | .9538±.0223 | .9344±.0364 | .9507±.0056 | .9436±.0286 | .9364±.0385 | .9603±.0051 | .9659±.0020 | .9557±.0070 |
| | GIB | .9554±.0036 | .9561±.0292 | .9321±.0267 | .9605±.0069 | .9521±.0231 | .9416±.0512 | .9651±.0061 | .9582±.0100 | .9489±.0029 |
| | SupCon | .9561±.0046 | .9531±.0043 | .9467±.0045 | .9606±.0031 | .9536±.0031 | .9468±.0075 | .9584±.0015 | .9580±.0051 | .9477±.0038 |
| | GRACE | .8978±.0145 | .8987±.0050 | .8949±.0030 | .8994±.0141 | .9007±.0045 | .8964±.0043 | .9053±.0051 | .9074±.0136 | .9075±.0113 |
| | RGIB-REP | .9723±.0035 | .9621±.0036 | .9519±.0052 | .9705±.0027 | .9604±.0052 | .9480±.0053 | .9785±.0031 | .9797±.0017 | .9785±.0032 |
| | RGIB-SSL | .9655±.0022 | .9592±.0043 | .9500±.0053 | .9658±.0023 | .9570±.0047 | .9486±.0029 | .9730±.0017 | .9752±.0030 | .9744±.0032 |
| L=6 | Standard | .9662±.0042 | .9511±.0079 | .9286±.0067 | .9656±.0045 | .9450±.0177 | .9276±.0229 | .9752±.0027 | .9766±.0035 | .9745±.0040 |
| | DropEdge | .9601±.0032 | .9484±.0053 | .9331±.0066 | .9608±.0020 | .9481±.0129 | .9331±.0073 | .9537±.0022 | .9559±.0027 | .9547±.0024 |
| | NeuralSparse | .9644±.0035 | .9550±.0033 | .9411±.0046 | .9538±.0035 | .9468±.0038 | .9387±.0053 | .9525±.0033 | .9541±.0031 | .9535±.0037 |
| | PTDNet | .9655±.0043 | .9585±.0033 | .9384±.0088 | .9657±.0024 | .9553±.0051 | .9391±.0044 | .9537±.0037 | .9559±.0025 | .9543±.0032 |
| | Co-teaching | .9654±.0058 | .9582±.0139 | .9319±.0163 | .9505±.0121 | .9534±.0222 | .9271±.0308 | .9679±.0198 | .9557±.0112 | .9557±.0282 |
| | Peer loss | .9658±.0120 | .9551±.0174 | .9308±.0099 | .9552±.0040 | .9441±.0226 | .9358±.0328 | .9575±.0164 | .9667±.0230 | .9593±.0236 |
| | Jaccard | .9754±.0063 | .9549±.0088 | .9292±.0104 | .9637±.0035 | .9495±.0213 | .9457±.0350 | .9500±.0040 | .9550±.0128 | .9433±.0066 |
| | GIB | .9800±.0203 | .9621±.0246 | .9329±.0238 | .9544±.0212 | .9542±.0395 | .9456±.0219 | .9619±.0072 | .9505±.0112 | .9536±.0124 |
| | SupCon | .9537±.0054 | .9452±.0073 | .9352±.0046 | .9534±.0068 | .9454±.0068 | .9375±.0023 | .9618±.0036 | .9627±.0032 | .9557±.0067 |
| | GRACE | .9024±.0098 | .8995±.0030 | .8936±.0029 | .9025±.0097 | .8983±.0055 | .8952±.0042 | .9154±.0092 | .9166±.0092 | .9151±.0082 |
| | RGIB-REP | .9666±.0028 | .9580±.0074 | .9467±.0067 | .9662±.0027 | .9606±.0041 | .9453±.0049 | .9740±.0033 | .9776±.0021 | .9753±.0031 |
| | RGIB-SSL | .9633±.0032 | .9416±.0036 | .9177±.0082 | .9627±.0037 | .9482±.0037 | .9276±.0093 | .9712±.0017 | .9742±.0025 | .9726±.0032 |

Table 43: Full results on Chameleon dataset with GAT.

| | | GAT | | | | | | | | |
|---|---|---|---|---|---|---|---|---|---|---|
| layers | method | bilateral noise | | | input noise | | | label noise | | |
| | | 20% | 40% | 60% | 20% | 40% | 60% | 20% | 40% | 60% |
| L=2 | Standard | .9725±.0027 | .9650±.0018 | .9625±.0018 | .9767±.0026 | .9747±.0020 | .9759±.0018 | .9746±.0023 | .9743±.0017 | .9711±.0041 |
| | DropEdge | .9716±.0023 | .9630±.0036 | .9563±.0023 | .9747±.0026 | .9717±.0030 | .9708±.0025 | .9752±.0018 | .9726±.0026 | .9698±.0035 |
| | NeuralSparse | .9733±.0029 | .9667±.0028 | .9615±.0028 | .9762±.0028 | .9748±.0028 | .9741±.0015 | .9762±.0022 | .9742±.0033 | .9704±.0042 |
| | PTDNet | .9736±.0028 | .9657±.0024 | .9620±.0027 | .9768±.0023 | .9735±.0001 | .9739±.0022 | .9670±.0021 | .9631±.0028 | .9611±.0058 |
| | Co-teaching | .9617±.0071 | .9629±.0047 | .9626±.0051 | .9624±.0084 | .9645±.0027 | .9647±.0104 | .9684±.0279 | .9610±.0139 | .9688±.0092 |
| | Peer loss | .9675±.0064 | .9646±.0101 | .9683±.0037 | .9602±.0046 | .9621±.0112 | .9611±.0078 | .9638±.0045 | .9626±.0074 | .9620±.0089 |
| | Jaccard | .9662±.0063 | .9665±.0062 | .9697±.0073 | .9652±.0130 | .9649±.0051 | .9616±.0123 | .9608±.0076 | .9612±.0047 | .9603±.0134 |
| | GIB | .9628±.0018 | .9639±.0115 | .9625±.0208 | .9681±.0091 | .9643±.0257 | .9601±.0285 | .9644±.0119 | .9688±.0096 | .9631±.0113 |
| | SupCon | .9709±.0059 | .9611±.0195 | .9607±.0032 | .9752±.0011 | .9721±.0026 | .9706±.0022 | .9692±.0035 | .9701±.0030 | .9689±.0057 |
| | GRACE | .7908±.0348 | .7829±.0494 | .7598±.0693 | .7992±.0371 | .7960±.0371 | .7913±.0418 | .7905±.0261 | .7985±.0322 | .7894±.0340 |
| | RGIB-REP | .9736±.0031 | .9668±.0021 | .9629±.0026 | .9769±.0018 | .9752±.0016 | .9752±.0018 | .9766±.0021 | .9753±.0029 | .9732±.0041 |
| | RGIB-SSL | .9635±.0037 | .9593±.0037 | .9536±.0041 | .9640±.0062 | .9587±.0048 | .9545±.0035 | .9709±.0015 | .9728±.0028 | .9719±.0046 |
| L=4 | Standard | .9721±.0035 | .9652±.0023 | .9605±.0031 | .9741±.0028 | .9686±.0030 | .9674±.0027 | .9740±.0037 | .9738±.0027 | .9712±.0047 |
| | DropEdge | .9666±.0031 | .9572±.0034 | .9469±.0043 | .9659±.0028 | .9598±.0024 | .9496±.0040 | .9749±.0024 | .9747±.0016 | .9718±.0031 |
| | NeuralSparse | .9687±.0027 | .9603±.0026 | .9536±.0033 | .9699±.0025 | .9600±.0036 | .9502±.0037 | .9748±.0022 | .9756±.0022 | .9727±.0035 |
| | PTDNet | .9707±.0041 | .9640±.0031 | .9549±.0034 | .9708±.0027 | .9602±.0025 | .9548±.0058 | .9754±.0025 | .9761±.0021 | .9709±.0041 |
| | Co-teaching | .9726±.0096 | .9648±.0030 | .9639±.0088 | .9551±.0037 | .9503±.0027 | .9506±.0028 | .9691±.0039 | .9664±.7565 | .9622±.0242 |
| | Peer loss | .9745±.0092 | .9714±.0088 | .9668±.0103 | .9637±.0041 | .9602±.0088 | .9658±.0064 | .9549±.0073 | .9531±.0096 | .9587±.0068 |
| | Jaccard | .9776±.0089 | .9692±.0098 | .9663±.0112 | .9729±.0024 | .9677±.0214 | .9744±.0012 | .9587±.0078 | .9519±.0092 | .9573±.0137 |
| | GIB | .9600±.0148 | .9646±.0041 | .9641±.0089 | .9639±.0197 | .9642±.0189 | .9636±.0047 | .9646±.0044 | .9624±.0103 | .9609±.0064 |
| | SupCon | .9644±.0033 | .9537±.0106 | .9381±.0125 | .9675±.0028 | .9631±.0042 | .9555±.0068 | .9594±.0131 | .9442±.0136 | .9386±.0102 |
| | GRACE | .5429±.0465 | .6085±.1132 | .6273±.1475 | .6178±.0940 | .5839±.1022 | .6513±.1544 | .5534±.0439 | .6195±.1055 | .6333±.0818 |
| | RGIB-REP | .9704±.0035 | .9645±.0027 | .9591±.0036 | .9713±.0023 | .9654±.0046 | .9572±.0047 | .9741±.0040 | .9745±.0028 | .9718±.0039 |
| | RGIB-SSL | .9507±.0034 | .9432±.0034 | .9389±.0059 | .9526±.0037 | .9451±.0045 | .9360±.0083 | .9619±.0037 | .9645±.0042 | .9643±.0038 |
| L=6 | Standard | .9659±.0029 | .9573±.0036 | .9482±.0054 | .9644±.0033 | .9543±.0075 | .9474±.0074 | .9722±.0043 | .9688±.0055 | .9698±.0065 |
| | DropEdge | .9461±.0082 | .9295±.0063 | .9062±.0218 | .9469±.0059 | .9293±.0087 | .9182±.0063 | .9698±.0036 | .9689±.0046 | .9694±.0037 |
| | NeuralSparse | .9563±.0079 | .9391±.0057 | .9185±.0157 | .9473±.0149 | .9338±.0043 | .9250±.0037 | .9688±.0037 | .9705±.0026 | .9696±.0063 |
| | PTDNet | .9508±.0075 | .9359±.0222 | .9288±.0153 | .9535±.0082 | .9424±.0053 | .9332±.0043 | .9723±.0033 | .9717±.0036 | .9697±.0046 |
| | Co-teaching | .9699±.0033 | .9627±.0129 | .9501±.0141 | .9709±.0129 | .9569±.0115 | .9482±.0155 | .9673±.0285 | .9698±.0260 | .9631±.0176 |
| | Peer loss | .9736±.0116 | .9570±.0133 | .9485±.0076 | .9659±.0060 | .9555±.0157 | .9469±.0086 | .9704±.0136 | .9701±.0196 | .9682±.0078 |
| | Jaccard | .9680±.0057 | .9607±.0055 | .9511±.0106 | .9815±.0185 | .9563±.0145 | .9468±.0100 | .9750±.0124 | .9732±.0093 | .9772±.0120 |
| | GIB | .9652±.0138 | .9666±.0113 | .9612±.0165 | .9776±.0272 | .9610±.0322 | .9624±.0365 | .9587±.0079 | .9514±.0078 | .9585±.0138 |
| | SupCon | .9481±.0185 | .9310±.0171 | .9015±.0169 | .9455±.0267 | .9490±.0157 | .9194±.0239 | .9413±.0206 | .9370±.0174 | .9241±.0144 |
| | GRACE | .5681±.0785 | .5107±.0287 | .5147±.0361 | .5599±.0776 | .5046±.0124 | .5167±.0502 | .5850±.0385 | .5864±.0795 | .5518±.0404 |
| | RGIB-REP | .9612±.0042 | .9410±.0069 | .9258±.0169 | .9531±.0054 | .9417±.0058 | .9313±.0092 | .9716±.0049 | .9747±.0036 | .9707±.0056 |
| | RGIB-SSL | .9367±.0058 | .9233±.0046 | .9199±.0053 | .9418±.0085 | .9236±.0056 | .9205±.0050 | .9576±.0066 | .9562±.0079 | .9557±.0083 |

Table 44: Full results on Chameleon dataset with SAGE.

| | | SAGE | | | | | | | | |
|---|---|---|---|---|---|---|---|---|---|---|
| | | bilateral noise | | | input noise | | | label noise | | |
| layers | method | 20% | 40% | 60% | 20% | 40% | 60% | 20% | 40% | 60% |
| L=2 | Standard | .9683±.0036 | .9605±.0038 | .9536±.0028 | .9754±.0019 | .9734±.0015 | .9762±.0018 | .9700±.0019 | .9670±.0026 | .9603±.0049 |
| | DropEdge | .9604±.0025 | .9614±.0028 | .9556±.0022 | .9677±.0022 | .9659±.0030 | .9673±.0026 | .9630±.0016 | .9678±.0033 | .9625±.0047 |
| | NeuralSparse | .9669±.0031 | .9584±.0029 | .9550±.0029 | .9751±.0025 | .9735±.0022 | .9745±.0021 | .9705±.0014 | .9661±.0031 | .9606±.0051 |
| | PTDNet | .9674±.0035 | .9594±.0028 | .9538±.0019 | .9752±.0022 | .9637±.0033 | .9651±.0029 | .9603±.0016 | .9658±.0036 | .9598±.0042 |
| | Co-teaching | .9581±.0040 | .9549±.0110 | .9543±.0093 | .9453±.0044 | .9425±.0083 | .9469±.0089 | .9585±.0006 | .9595±.0039 | .9525±.0309 |
| | Peer loss | .9638±.0096 | .9581±.0098 | .9512±.0021 | .9541±.0091 | .9561±.0075 | .9557±.0104 | .9625±.0146 | .9506±.0155 | .9587±.0227 |
| | Jaccard | .9672±.0074 | .9650±.0123 | .9589±.0056 | .9503±.0007 | .9540±.0190 | .9517±.0015 | .9592±.0010 | .9522±.0027 | .9545±.0118 |
| | GIB | .9650±.0066 | .9593±.0214 | .9530±.0021 | .9671±.0007 | .9606±0.000 | .9639±.0275 | .9537±.0055 | .9523±.0048 | .9586±.0068 |
| | SupCon | .9636±.0031 | .9656±.0017 | .9599±.0041 | .9506±.0022 | .9585±.0020 | .9593±.0019 | .9567±.0024 | .9530±.0027 | .9516±.0037 |
| | GRACE | .8215±.0332 | .8327±.0178 | .8372±.0245 | .8369±.0146 | .8251±.0441 | .8295±.0422 | .8610±.0055 | .8583±.0114 | .8641±.0137 |
| | RGIB-REP | .9699±.0028 | .9626±.0031 | .9577±.0034 | .9765±.0022 | .9741±.0018 | .9770±.0014 | .9733±.0017 | .9715±.0025 | .9670±.0037 |
| | RGIB-SSL | .9740±.0030 | .9709±.0027 | .9686±.0025 | .9752±.0020 | .9737±.0031 | .9741±.0035 | .9789±.0016 | .9791±.0028 | .9781±.0029 |
| L=4 | Standard | .9645±.0035 | .9580±.0035 | .9505±.0010 | .9721±.0017 | .9721±.0021 | .9721±.0016 | .9678±.0026 | .9641±.0028 | .9605±.0051 |
| | DropEdge | .9674±.0041 | .9585±.0026 | .9516±.0024 | .9652±.0029 | .9638±.0039 | .9652±.0019 | .9602±.0021 | .9663±.0030 | .9614±.0038 |
| | NeuralSparse | .9649±.0022 | .9576±.0035 | .9495±.0041 | .9619±.0032 | .9602±.0022 | .9620±.0031 | .9675±.0017 | .9642±.0031 | .9603±.0046 |
| | PTDNet | .9651±.0036 | .9574±.0034 | .9506±.0037 | .9624±.0027 | .9622±.0015 | .9630±.0035 | .9687±.0024 | .9646±.0025 | .9596±.0056 |
| | Co-teaching | .9722±.0090 | .9665±.0040 | .9582±.0007 | .9683±.0083 | .9633±.0101 | .9613±.0067 | .9617±.0255 | .9638±.0070 | .9637±.0066 |
| | Peer loss | .9543±.0099 | .9663±.0044 | .9496±.0096 | .9610±.0110 | .9667±.0086 | .9601±.0060 | .9657±.0035 | .9679±.0026 | .9661±.0180 |
| | Jaccard | .9438±.0076 | .9485±.0047 | .9444±.0069 | .9653±.0115 | .9683±.0218 | .9697±.0157 | .9696±.0043 | .9537±.0048 | .9685±.0088 |
| | GIB | .9557±.0177 | .9574±.0043 | .9538±.0117 | .9644±.0214 | .9614±.0204 | .9559±.0038 | .9576±.0026 | .9464±.0080 | .9303±.0065 |
| | SupCon | .9494±.0130 | .9351±.0093 | .9263±.0065 | .9592±.0122 | .9538±.0122 | .9512±.0125 | .9478±.0099 | .9501±.0126 | .9449±.0099 |
| | GRACE | .8040±.0364 | .8542±.0415 | .8266±.0443 | .8362±.0272 | .8251±.0484 | .8642±.0383 | .7462±.0412 | .7970±.0306 | .7892±.0341 |
| | RGIB-REP | .9665±.0031 | .9582±.0035 | .9520±.0035 | .9740±.0018 | .9719±.0028 | .9722±.0022 | .9690±.0021 | .9696±.0023 | .9671±.0058 |
| | RGIB-SSL | .9725±.0027 | .9682±.0028 | .9655±.0049 | .9740±.0021 | .9707±.0040 | .9721±.0044 | .9748±.0023 | .9762±.0031 | .9754±.0040 |
| L=6 | Standard | .9606±.0046 | .9541±.0058 | .9470±.0035 | .9679±.0031 | .9658±.0041 | .9640±.0059 | .9673±.0026 | .9641±.0038 | .9625±.0042 |
| | DropEdge | .9532±.0028 | .9540±.0024 | .9487±.0043 | .9592±.0028 | .9582±.0035 | .9580±.0299 | .9500±.0014 | .9561±.0030 | .9537±.0024 |
| | NeuralSparse | .9525±.0043 | .9383±.0365 | .9480±.0036 | .9597±.0027 | .9569±.0037 | .9530±.0057 | .9589±.0024 | .9552±.0025 | .9515±.0034 |
| | PTDNet | .9536±.0042 | .9557±.0038 | .9483±.0047 | .9706±.0035 | .9581±.0040 | .9459±.0074 | .9580±.0024 | .9559±.0021 | .9429±.0045 |
| | Co-teaching | .9503±.0070 | .9466±.0113 | .9355±.0039 | .9492±.0031 | .9457±.0059 | .9403±.0118 | .9445±.0009 | .9465±.0055 | .9447±.0115 |
| | Peer loss | .9553±.0064 | .9409±.0086 | .9330±.0076 | .9697±.0084 | .9656±.0110 | .9652±.0066 | .9521±.0210 | .9595±.0195 | .9589±.0047 |
| | Jaccard | .9416±.0113 | .9535±.0116 | .9535±.0111 | .9610±.0042 | .9568±.0210 | .9463±.0085 | .9518±.0057 | .9421±.0032 | .9433±.0137 |
| | GIB | .9609±.0173 | .9579±.0050 | .9568±.0070 | .9691±.0315 | .9698±.0037 | .9592±.0279 | .9433±.0065 | .9366±.0074 | .9347±.0115 |
| | SupCon | .9310±.0107 | .9134±.0087 | .9178±.0101 | .9291±.0114 | .9244±.0271 | .9181±.0281 | .9342±.0132 | .9373±.0116 | .9313±.0087 |
| | GRACE | .7479±.0742 | .7756±.0708 | .7718±.0880 | .7275±.0906 | .7914±.0755 | .8109±.0655 | .7754±.0258 | .7629±.0508 | .7686±.0437 |
| | RGIB-REP | .9636±.0022 | .9558±.0032 | .9455±.0038 | .9703±.0022 | .9641±.0075 | .9607±.0050 | .9683±.0013 | .9681±.0033 | .9664±.0042 |
| | RGIB-SSL | .9665±.0030 | .9601±.0045 | .9563±.0036 | .9662±.0052 | .9654±.0040 | .9610±.0032 | .9699±.0022 | .9716±.0033 | .9698±.0031 |

Table 45: Full results on Squirrel dataset with GCN.

| | | GCN | | | | | | | | |
|---|---|---|---|---|---|---|---|---|---|---|
| | | bilateral noise | | | input noise | | | label noise | | |
| layers | method | 20% | 40% | 60% | 20% | 40% | 60% | 20% | 40% | 60% |
| L=2 | Standard | .9621±.0018 | .9519±.0020 | .9444±.0024 | .9610±.0028 | .9490±.0031 | .9401±.0036 | .9744±.0013 | .9731±.0010 | .9722±.0011 |
| | DropEdge | .9571±.0015 | .9480±.0032 | .9370±.0028 | .9587±.0011 | .9486±.0026 | .9349±.0026 | .9623±.0015 | .9612±.0008 | .9604±.0014 |
| | NeuralSparse | .9611±.0019 | .9528±.0027 | .9409±.0023 | .9621±.0017 | .9495±.0016 | .9364±.0022 | .9741±.0013 | .9724±.0018 | .9722±.0010 |
| | PTDNet | .9612±.0018 | .9513±.0026 | .9424±.0024 | .9619±.0016 | .9488±.0035 | .9363±.0022 | .9744±.0011 | .9723±.0015 | .9718±.0009 |
| | Co-teaching | .9701±.0076 | .9535±.0044 | .9465±.0059 | .9599±.0097 | .9525±.0120 | .9454±.0109 | .9629±.0047 | .9666±.0304 | .9631±.0004 |
| | Peer loss | .9610±.0015 | .9608±.0057 | .9451±.0097 | .9636±.0107 | .9491±.0125 | .9436±.0091 | .9694±.0012 | .9628±.0134 | .9676±.0143 |
| | Jaccard | .9613±.0027 | .9568±.0046 | .9509±.0014 | .9622±.0159 | .9594±.0111 | .9397±.0119 | .9528±.0017 | .9405±.0089 | .9401±.0010 |
| | GIB | .9522±.0195 | .9478±.0094 | .9489±.0159 | .9494±.0260 | .9461±.0091 | .9432±.0068 | .9457±.0099 | .9462±.0017 | .9492±.0060 |
| | SupCon | .9494±.0013 | .9439±.0009 | .9458±.0024 | .9421±.0008 | .9549±.0013 | .9489±.0030 | .9444±.0011 | .9423±.0015 | .9318±.0009 |
| | GRACE | .9292±.0011 | .9316±.0010 | .9330±.0013 | .9290±.0012 | .9317±.0010 | .9327±.0016 | .9054±.0018 | .9058±.0015 | .9052±.0018 |
| | RGIB-REP | .9614±.0019 | .9554±.0026 | .9471±.0035 | .9611±.0021 | .9512±.0032 | .9392±.0043 | .9761±.0015 | .9753±.0011 | .9745±.0015 |
| | RGIB-SSL | .9571±.0019 | .9476±.0029 | .9402±.0020 | .9534±.0016 | .9486±.0027 | .9451±.0031 | .9752±.0023 | .9749±.0019 | .9738±.0023 |
| L=4 | Standard | .9432±.0036 | .9406±.0031 | .9386±.0025 | .9416±.0042 | .9395±.0011 | .9411±.0040 | .9720±.0016 | .9720±.0016 | .9710±.0021 |
| | DropEdge | .9439±.0063 | .9377±.0024 | .9365±.0045 | .9426±.0042 | .9376±.0033 | .9358±.0023 | .9608±.0008 | .9603±.0013 | .9698±.0016 |
| | NeuralSparse | .9494±.0065 | .9309±.0028 | .9297±.0033 | .9469±.0047 | .9403±.0028 | .9417±.0062 | .9633±.0009 | .9626±.0016 | .9625±.0016 |
| | PTDNet | .9485±.0056 | .9326±.0046 | .9304±.0048 | .9469±.0048 | .9400±.0030 | .9379±.0026 | .9633±.0018 | .9623±.0012 | .9626±.0012 |
| | Co-teaching | .9461±.0099 | .9352±.0064 | .9374±.0051 | .9462±.0101 | .9425±.0032 | .9306±.0053 | .9675±.0292 | .9641±.0115 | .9655±.0264 |
| | Peer loss | .9457±.0117 | .9345±.0117 | .9286±.0044 | .9362±.0038 | .9386±.0058 | .9336±.0043 | .9636±.0012 | .9694±.0033 | .9696±.0006 |
| | Jaccard | .9443±.0047 | .9327±.0079 | .9244±.0054 | .9388±.0179 | .9345±.0146 | .9240±.0055 | .9529±.0029 | .9512±.0089 | .9501±.0120 |
| | GIB | .9472±.0127 | .9329±.0155 | .9302±.0202 | .9390±.0045 | .9406±.0196 | .9397±.0249 | .9641±.0025 | .9628±.0032 | .9601±.0042 |
| | SupCon | .9473±.0010 | .9348±.0017 | .9301±.0029 | .9372±.0010 | .9343±.0037 | .9305±.0033 | .9516±.0042 | .9595±.0011 | .9511±.0040 |
| | GRACE | .9394±.0008 | .9380±.0007 | .9363±.0016 | .9392±.0011 | .9378±.0008 | .9363±.0017 | .9171±.0019 | .9174±.0014 | .9166±.0013 |
| | RGIB-REP | .9509±.0046 | .9455±.0071 | .9434±.0059 | .9495±.0053 | .9432±.0059 | .9405±.0050 | .9735±.0013 | .9733±.0015 | .9737±.0009 |
| | RGIB-SSL | .9499±.0040 | .9426±.0019 | .9425±.0021 | .9479±.0051 | .9429±.0017 | .9429±.0024 | .9727±.0015 | .9729±.0019 | .9726±.0011 |
| L=6 | Standard | .9484±.0049 | .9429±.0038 | .9408±.0039 | .9489±.0057 | .9408±.0021 | .9386±.0022 | .9688±.0028 | .9675±.0027 | .9656±.0034 |
| | DropEdge | .9460±.0063 | .9375±.0028 | .9349±.0017 | .9474±.0057 | .9409±.0030 | .9365±.0022 | .9669±.0022 | .9652±.0031 | .9638±.0038 |
| | NeuralSparse | .9508±.0049 | .9429±.0047 | .9391±.0029 | .9532±.0060 | .9412±.0044 | .9388±.0036 | .9681±.0020 | .9647±.0018 | .9671±.0019 |
| | PTDNet | .9535±.0047 | .9451±.0050 | .9395±.0033 | .9514±.0041 | .9432±.0068 | .9380±.0020 | .9690±.0023 | .9667±.0035 | .9649±.0025 |
| | Co-teaching | .9480±.0144 | .9454±.0074 | .9463±.0076 | .9538±.0051 | .9424±.0048 | .9379±.0012 | .9672±.0179 | .9694±.0283 | .9654±.0195 |
| | Peer loss | .9539±.0051 | .9420±.0127 | .9405±.0060 | .9445±.0139 | .9446±.0012 | .9462±.0081 | .9684±.0182 | .9639±.0188 | .9624±.0189 |
| | Jaccard | .9519±.0138 | .9459±.0100 | .9398±.0115 | .9483±.0070 | .9429±.0203 | .9428±.0105 | .9697±.0074 | .9687±.0062 | .9673±.0077 |
| | GIB | .9447±.0103 | .9459±.0026 | .9410±.0108 | .9431±.0289 | .9413±.0301 | .9386±.0238 | .9633±.0041 | .9619±.0114 | .9612±.0064 |
| | SupCon | .9590±.0023 | .9507±.0019 | .9463±.0028 | .9580±.0024 | .9519±.0018 | .9464±.0026 | .9638±.0051 | .9524±.0048 | .9419±.0012 |
| | GRACE | .9399±.0007 | .9372±.0006 | .9348±.0017 | .9397±.0012 | .9370±.0009 | .9349±.0017 | .9213±.0017 | .9218±.0013 | .9211±.0014 |
| | RGIB-REP | .9556±.0028 | .9492±.0060 | .9463±.0063 | .9570±.0018 | .9496±.0060 | .9420±.0056 | .9677±.0029 | .9676±.0031 | .9662±.0018 |
| | RGIB-SSL | .9462±.0029 | .9471±.0020 | .9448±.0032 | .9509±.0032 | .9474±.0017 | .9449±.0030 | .9642±.0012 | .9659±.0027 | .9647±.0010 |

Table 46: Full results on Squirrel dataset with GAT.

| | | GAT | | | | | | | | |
|---|---|---|---|---|---|---|---|---|---|---|
| | | bilateral noise | | | input noise | | | label noise | | |
| layers | method | 20% | 40% | 60% | 20% | 40% | 60% | 20% | 40% | 60% |
| L=2 | Standard | .9680±.0007 | .9635±.0017 | .9588±.0025 | .9702±.0008 | .9690±.0010 | .9659±.0014 | .9719±.0018 | .9701±.0017 | .9686±.0012 |
| | DropEdge | .9619±.0014 | .9567±.0020 | .9505±.0022 | .9627±.0008 | .9593±.0014 | .9551±.0012 | .9684±.0015 | .9662±.0026 | .9661±.0018 |
| | NeuralSparse | .9656±.0009 | .9602±.0012 | .9533±.0019 | .9662±.0011 | .9629±.0010 | .9584±.0017 | .9709±.0012 | .9692±.0014 | .9685±.0009 |
| | PTDNet | .9655±.0011 | .9602±.0012 | .9546±.0024 | .9668±.0010 | .9630±.0015 | .9603±.0021 | .9610±.0015 | .9698±.0012 | .9689±.0016 |
| | Co-teaching | .9566±.0034 | .9562±.0066 | .9516±.0026 | .9505±.0045 | .9509±.0092 | .9509±.0046 | .9591±.0293 | .9880±.0174 | .9360±.0089 |
| | Peer loss | .9720±.0097 | .9726±.0039 | .9682±.0106 | .9706±6.649 | .9715±.0064 | .9704±.0107 | .9681±.0121 | .9635±.0069 | .9683±.0116 |
| | Jaccard | .9715±.0098 | .9651±.0053 | .9648±.0094 | .9807±.0202 | .9768±.0041 | .9645±.0174 | .9637±.0081 | .9668±.0104 | .9627±.0002 |
| | GIB | .9727±.0019 | .9820±.0058 | .9610±.0069 | .9780±0.001 | .9965±0.000 | .9838±.0123 | .9660±.0067 | .9611±.0022 | .9679±.0061 |
| | SupCon | .9471±.0019 | .9386±.0022 | .9353±.0026 | .9539±.0032 | .9463±.0026 | .9411±.0034 | .9629±.0028 | .9631±.0019 | .9636±.0021 |
| | GRACE | .7300±.1193 | .7583±.1135 | .7015±.1383 | .7494±.1101 | .6903±.1807 | .7032±.1659 | .8109±.0382 | .7780±.1044 | .7608±.0620 |
| | RGIB-REP | .9675±.0014 | .9631±.0012 | .9587±.0023 | .9692±.0007 | .9660±.0013 | .9632±.0016 | .9735±.0012 | .9719±.0013 | .9715±.0010 |
| | RGIB-SSL | .9453±.0020 | .9421±.0017 | .9385±.0022 | .9418±.0024 | .9394±.0033 | .9362±.0023 | .9726±.0007 | .9705±.0024 | .9695±.0024 |
| L=4 | Standard | .9581±.0046 | .9436±.0063 | .9335±.0062 | .9592±.0047 | .9455±.0075 | .9415±.0061 | .9682±.0030 | .9690±.0028 | .9686±.0021 |
| | DropEdge | .9501±.0041 | .9340±.0046 | .9234±.0059 | .9476±.0054 | .9354±.0067 | .9292±.0039 | .9529±.0047 | .9513±.0054 | .9510±.0037 |
| | NeuralSparse | .9524±.0041 | .9383±.0046 | .9320±.0044 | .9401±.0071 | .9377±.0045 | .9334±.0041 | .9559±.0031 | .9555±.0032 | .9556±.0022 |
| | PTDNet | .9416±.0054 | .9310±.0059 | .9302±.0032 | .9447±.0031 | .9393±.0053 | .9305±.0131 | .9574±.0039 | .9557±.0025 | .9552±.0042 |
| | Co-teaching | .9324±.0109 | .9024±.0120 | .9029±.0126 | .9421±.0074 | .9345±.0126 | .9371±.0091 | .9576±.0124 | .9592±.0064 | .9551±.0087 |
| | Peer loss | .9520±.0071 | .9506±.0145 | .9370±.0140 | .9560±.0063 | .9451±.0086 | .9448±.0142 | .9548±.0128 | .9500±.0088 | .9593±.0134 |
| | Jaccard | .9561±.0133 | .9423±.0160 | .9396±.0117 | .9578±.0185 | .9583±.0256 | .9558±.0050 | .9572±.0074 | .9572±.0083 | .9554±.0089 |
| | GIB | .9353±.0080 | .9271±.0093 | .9098±.0079 | .9545±.0156 | .9504±.0240 | .9530±.0348 | .9544±.0023 | .9549±.0073 | .9524±.0060 |
| | SupCon | .9397±.0044 | .9322±.0038 | .9266±.0047 | .9365±.0039 | .9340±.0021 | .9314±.0029 | .9574±.0019 | .9557±.0065 | .9556±.0049 |
| | GRACE | .5710±.1110 | .5658±.1143 | .6203±.1324 | .5694±.1202 | .5771±.0945 | .6021±.1203 | .5291±.0694 | .5097±.0139 | .5029±.0072 |
| | RGIB-REP | .9541±.0059 | .9432±.0059 | .9316±.0063 | .9526±.0037 | .9401±.0072 | .9389±.0056 | .9681±.0028 | .9672±.0029 | .9660±.0048 |
| | RGIB-SSL | .9347±.0038 | .9321±.0031 | .9260±.0032 | .9321±.0022 | .9327±.0020 | .9270±.0028 | .9592±.0031 | .9651±.0014 | .9641±.0021 |
| L=6 | Standard | .9507±.0050 | .9309±.0164 | .9254±.0089 | .9487±.0065 | .9419±.0041 | .9255±.0073 | .9585±.0097 | .9520±.0070 | .9507±.0162 |
| | DropEdge | .9347±.0049 | .9279±.0118 | .9181±.0078 | .9368±.0082 | .9262±.0076 | .9175±.0127 | .9412±.0092 | .9431±.0076 | .9444±.0090 |
| | NeuralSparse | .9399±.0088 | .9279±.0167 | .9224±.0051 | .9423±.0063 | .9342±.0023 | .9213±.0115 | .9468±.0118 | .9474±.0102 | .9504±.0051 |
| | PTDNet | .9433±.0105 | .9348±.0072 | .9213±.0100 | .9460±.0048 | .9340±.0052 | .9273±.0094 | .9482±.0088 | .9541±.0092 | .9453±.0077 |
| | Co-teaching | .9219±.0059 | .9329±.0231 | .9229±.0185 | .9392±.0093 | .9352±.0098 | .9261±.0100 | .9496±.0383 | .9445±.0272 | .9490±.0284 |
| | Peer loss | .9300±.0136 | .9316±.0220 | .9349±.0147 | .9271±.0119 | .9170±.0117 | .9057±.0127 | .9230±.0221 | .9341±.0113 | .9187±.0325 |
| | Jaccard | .9577±.0112 | .9374±.0225 | .9297±.0158 | .9469±.0111 | .9406±.0111 | .9309±.0168 | .9393±.0153 | .9390±.0151 | .9314±.0223 |
| | GIB | .9504±.0239 | .9475±.0221 | .9445±.0127 | .9484±.0211 | .9487±.0139 | .9438±.0307 | .9421±.0133 | .9484±.0101 | .9486±.0206 |
| | SupCon | .9269±.0089 | .9174±.0092 | .9107±.0052 | .9278±.0057 | .9223±.0054 | .9210±.0038 | .9140±.0072 | .9173±.0085 | .9182±.0081 |
| | GRACE | .5003±.0009 | .5427±.0961 | .6004±.1210 | .5657±.1221 | .6023±.0931 | .5342±.0684 | .5064±.0145 | .5017±.0031 | .5074±.0190 |
| | RGIB-REP | .9404±.0067 | .9248±.0161 | .9218±.0051 | .9437±.0084 | .9299±.0070 | .9238±.0063 | .9473±.0071 | .9486±.0073 | .9397±.0045 |
| | RGIB-SSL | .9286±.0033 | .9241±.0055 | .9165±.0055 | .9276±.0023 | .9245±.0015 | .9257±.0037 | .9593±.0043 | .9586±.0027 | .9499±.0067 |

Table 47: Full results on Squirrel dataset with SAGE.

| | | SAGE | | | | | | | | |
|---|---|---|---|---|---|---|---|---|---|---|
| | | bilateral noise | | | input noise | | | label noise | | |
| layers | method | 20% | 40% | 60% | 20% | 40% | 60% | 20% | 40% | 60% |
| L=2 | Standard | .9680±.0015 | .9626±.0015 | .9570±.0016 | .9737±.0010 | .9736±.0010 | .9721±.0011 | .9692±.0015 | .9643±.0012 | .9606±.0017 |
| | DropEdge | .9657±.0011 | .9602±.0015 | .9535±.0020 | .9622±.0010 | .9625±.0009 | .9609±.0020 | .9593±.0009 | .9530±.0009 | .9593±.0007 |
| | NeuralSparse | .9572±.0008 | .9517±.0012 | .9562±.0019 | .9534±.0007 | .9534±.0010 | .9521±.0016 | .9589±.0012 | .9534±.0009 | .9591±.0010 |
| | PTDNet | .9578±.0009 | .9513±.0013 | .9559±.0021 | .9630±.0010 | .9504±.0007 | .9524±.0010 | .9596±.0014 | .9529±.0010 | .9596±.0019 |
| | Co-teaching | .9526±.0031 | .9560±.0071 | .9526±.0029 | .9403±.0050 | .9585±.0061 | .9585±.0101 | .9533±.0022 | .9533±.0306 | .9539±.0026 |
| | Peer loss | .9552±.0079 | .9507±.0050 | .9588±.0066 | .9512±.0052 | .9534±.0019 | .9599±.0013 | .9525±.0105 | .9555±.0178 | .9510±.0187 |
| | Jaccard | .9658±.0111 | .9619±.0068 | .9630±.0027 | .9600±.0189 | .9649±.0173 | .9617±.0171 | .9620±.0105 | .9608±.0068 | .9603±.0044 |
| | GIB | .9656±.0079 | .9660±.0077 | .9661±.0060 | .9632±.0260 | .9517±.0213 | .9565±.0010 | .9565±.0009 | .9593±.0002 | .9548±.0111 |
| | SupCon | .9628±.0015 | .9541±.0019 | .9409±.0040 | .9585±.0061 | .9585±.0010 | .9533±.0022 | .9533±.0306 | .9539±.0026 | .9400±.0021 |
| | GRACE | .8543±.0076 | .8572±.0094 | .8528±.0166 | .8563±.0068 | .8601±.0087 | .8528±.0075 | .8477±.0119 | .8459±.0105 | .8452±.0141 |
| | RGIB-REP | .9701±.0010 | .9647±.0014 | .9597±.0018 | .9757±.0009 | .9747±.0007 | .9737±.0011 | .9735±.0010 | .9707±.0007 | .9682±.0016 |
| | RGIB-SSL | .9536±.0016 | .9480±.0027 | .9448±.0030 | .9734±.0014 | .9739±.0015 | .9803±.0024 | .9765±.0021 | .9702±.0015 | .9694±.0005 |
| L=4 | Standard | .9584±.0107 | .9577±.0076 | .9541±.0021 | .9637±.0092 | .9630±.0079 | .9607±.0090 | .9663±.0020 | .9612±.0049 | .9612±.0020 |
| | DropEdge | .9518±.0030 | .9577±.0021 | .9515±.0033 | .9656±.0051 | .9612±.0093 | .9573±.0111 | .9683±.0020 | .9628±.0014 | .9608±.0012 |
| | NeuralSparse | .9604±.0083 | .9570±.0040 | .9513±.0040 | .9644±.0066 | .9599±.0084 | .9605±.0061 | .9652±.0035 | .9632±.0023 | .9605±.0021 |
| | PTDNet | .9624±.0050 | .9581±.0039 | .9536±.0033 | .9667±.0045 | .9668±.0057 | .9649±.0052 | .9675±.0043 | .9635±.0040 | .9607±.0026 |
| | Co-teaching | .9518±.0181 | .9468±.0093 | .9479±.0024 | .9525±.0086 | .9577±.0159 | .9528±.0137 | .9625±.0244 | .9610±.0220 | .9675±.0097 |
| | Peer loss | .9632±.0196 | .9662±.0085 | .9602±.0088 | .9674±.0145 | .9626±.0120 | .9643±.0145 | .9637±.0004 | .9617±.0170 | .9691±.0016 |
| | Jaccard | .9622±.0135 | .9606±.0157 | .9587±.0105 | .9647±.0084 | .9630±.0253 | .9640±.0278 | .9505±.0026 | .9508±.0063 | .9599±.0039 |
| | GIB | .9527±.0122 | .9538±.0275 | .9531±.0174 | .9603±.0081 | .9620±.0115 | .9580±.0330 | .9598±.0031 | .9523±.0064 | .9559±.0091 |
| | SupCon | .9478±.0042 | .9412±.0059 | .9342±.0039 | .9267±.0045 | .9368±.0028 | .9249±.0026 | .9375±.0034 | .9335±.0012 | .9317±.0021 |
| | GRACE | .8859±.0115 | .8751±.0085 | .8609±.0212 | .8824±.0113 | .8805±.0261 | .8633±.0178 | .8386±.0130 | .8466±.0103 | .8347±.0180 |
| | RGIB-REP | .9611±.0058 | .9578±.0035 | .9536±.0037 | .9682±.0049 | .9667±.0051 | .9584±.0074 | .9689±.0014 | .9664±.0027 | .9635±.0022 |
| | RGIB-SSL | .9498±.0032 | .9443±.0047 | .9422±.0044 | .9531±.0071 | .9599±.0023 | .9551±.0019 | .9623±.0021 | .9603±.0087 | .9601±.0091 |
| L=6 | Standard | .9555±.0065 | .9528±.0038 | .9461±.0054 | .9592±.0053 | .9600±.0036 | .9551±.0042 | .9574±.0192 | .9540±.0207 | .9583±.0023 |
| | DropEdge | .9544±.0069 | .9411±.0200 | .9463±.0041 | .9566±.0070 | .9473±.0172 | .9519±.0052 | .9464±.0023 | .9402±.0084 | .9396±.0025 |
| | NeuralSparse | .9563±.0060 | .9515±.0036 | .9457±.0054 | .9591±.0046 | .9460±.0052 | .9415±.0061 | .9371±.0025 | .9344±.0017 | .9319±.0033 |
| | PTDNet | .9565±.0060 | .9524±.0050 | .9463±.0056 | .9593±.0045 | .9584±.0044 | .9492±.0038 | .9384±.0023 | .9342±.0018 | .9236±.0018 |
| | Co-teaching | .9423±.0071 | .9396±.0126 | .9251±.0137 | .9382±.0063 | .9433±.0054 | .9388±.0088 | .9344±.0232 | .9496±.0204 | .9374±.0083 |
| | Peer loss | .9473±.0108 | .9371±.0040 | .9333±.0084 | .9339±.0053 | .9312±.0058 | .9331±.0066 | .9396±.0211 | .9375±.0195 | .9335±.0035 |
| | Jaccard | .9335±.0058 | .9305±.0057 | .9225±.0146 | .9309±.0059 | .9386±.0084 | .9433±.0147 | .9348±.0200 | .9210±.0285 | .9273±.0029 |
| | GIB | .9437±.0258 | .9323±.0155 | .9204±.0182 | .9492±.0239 | .9406±.0137 | .9322±.0335 | .9407±.0269 | .9364±.0274 | .9394±.0079 |
| | SupCon | .9392±.0049 | .9330±.0098 | .9221±.0138 | .9266±.0024 | .9313±.0102 | .9309±.0052 | .9264±.0021 | .9202±.0084 | .9196±.0021 |
| | GRACE | .8418±.0983 | .8470±.0823 | .8732±.0151 | .8747±.0151 | .8126±.1348 | .8716±.0148 | .8385±.0097 | .8370±.0295 | .8487±.0121 |
| | RGIB-REP | .9558±.0047 | .9474±.0066 | .9455±.0052 | .9583±.0044 | .9566±.0035 | .9447±.0030 | .9662±.0057 | .9646±.0021 | .9634±.0030 |
| | RGIB-SSL | .9437±.0026 | .9391±.0036 | .9366±.0042 | .9441±.0064 | .9549±.0012 | .9469±.0012 | .9678±.0031 | .9613±.0076 | .9601±.0041 |

