# OpenReview forum: "Combating Bilateral Edge Noise for Robust Link Prediction"
_NeurIPS.cc/2023/Conference — NeurIPS 2023 poster_

### Official Review · Reviewer_4XLr · 2023-06-27

**Soundness:** 3 good
**Presentation:** 3 good
**Contribution:** 3 good
**Rating:** 7
**Confidence:** 4

**Summary:**

This paper focuses on the robustness of graph neural networks (GNNs) in the presence of edge noise during link prediction on graphs. The authors empirically investigate the impact of edge noise on both the input topology and target labels, revealing significant performance degradation and representation collapse. To address this challenge, they propose a novel principle called Robust Graph Information Bottleneck (RGIB), which leverages information theory to extract reliable supervision signals and prevent representation collapse. The paper explores two instantiations of RGIB, RGIB-SSL and RGIB-REP, which utilize self-supervised learning and data reparametrization techniques, respectively, for implicit and explicit data denoising.


**Strengths:**

Overall, this paper makes a valuable contribution to understanding and enhancing the robustness of GNNs in link prediction tasks affected by edge noise.

The main novelty is based on systematic study of Bilateral Edge Noise including its empirical influence, visualization, and two instantiations of Robust Graph Information Bottleneck with data reparameterization and data augmentation.

This paper also provides theoretical analysis for noisy dataset convergence relaxations.



**Weaknesses:**

This paper provides a novel method and analysis for the edge noise issue for the GNN, through the lens of information theory. The method is clearly explained, and the performance is supported with grounded evaluations with visualizations. I did not carefully check the theory proofs, but from the main paper, there is no obvious limitation.

**Questions:**

If the SSL and REP versions are parallel methods, could the authors discuss the guidelines on method selection? Are they be simultaneously used while still improve? It would be better to add a section of such discussions with supportive case evidence.


**Limitations:**

See above.

---

> ### Author Rebuttal · Authors · 2023-08-09
>
> We thank the reviewer 4XLr for the valuable feedback. We addressed all the comments. Please find the point-to-point responses below. Any further comments and discussions are welcomed!
>
> **Q1**. *If the SSL and REP versions are parallel methods, could the authors discuss the guidelines on method selection? Are they be simultaneously used while still improve? It would be better to add a section of such discussions with supportive case evidence.*
>
> **Reply:** We appreciate Reviewer 4XLr’s insightful comment. We would like to clarify the comparison of the two instantiations of RGIB in the following five folds.
>
> **(1)** **From the theoretical perspective**, RGIB-SSL **explicitly** optimizes the representation $H$ with self-supervised regularizations, i.e., alignment $I(H_1; H_2)$ and uniformity $H(H_1)$, $H(H_2)$. By contrast, RGIB-REP **implicitly** optimizes $H$ by purifying the noisy $\tilde{A}$ and $\tilde{Y}$ with the reparameterization mechanism to extract clean signals in the forms of latent variables $Z_{Y}$ and $Z_{A}$. The information constraints $I(Z_{A}; \tilde{A})$, $I(Z_{Y}; \tilde{Y})$ are directly acting on $Z_{Y}$ and $Z_{A}$ and indirectly regularizing the representation $H$.
>
> **(2)** **Besides, from the methodology perspective, both instantiations are equipped with adaptive designs for obtaining an effective information bottleneck.** RGIB-SSL utilizes the automatically augmented views $\tilde{A_1}$, $\tilde{A_2}$ in a contrastive manner to be resistant to input noise. RGIB-SSL is intrinsically robust to label noise due to its self-supervised nature. Besides, RGIB-REP explicitly purifies the input graph’s topology and target labels with the jointly reparameterized $Z_{Y}$ and $Z_{A}$. It enables to model the mutual patterns of edge noise from both input and label spaces.
>
> In addition, we conduct a further comparison and analysis of the two instantiations that are summarized as follows.
>
> | Instantiation | methodology              | advantages                                                   | disadvantages                                                |
> | ------------- | ------------------------ | ------------------------------------------------------------ | ------------------------------------------------------------ |
> | RGIB-SSL      | self-supervised learning | automated graph augmentation; good effectiveness; can be applied in entirely self-supervised settings without labels. | with expensive calculation for the contrastive objectives, especially the uniformity; requires extra graph augmentation operations. |
> | RGIB-REP      | data reparametrization   | no needs to do data augmentation; good efficiency; the input/output constraints do not require extra annotations for supervision and can be easily controlled. | sensitive to the hyper-parameters $\lambda$; less effective in extremely noisy cases; only applicable in fully supervised settings. |
>
> **(3)** **In addition, RGIB-SSL assumes that the learned representation can be improved with higher uniformity and alignment**. As RGIB-SSL directly acts on the graph representation, it is more suitable for recovering the distribution of representation, especially when encountering representation collapse due to the severe edge noise. By contrast, RGIB-REP explicitly purifies the input graph’s topology and target labels with the jointly reparameterized $Z_A$ and $Z_Y$. Here, the latent variables $Z_A$,$Z_Y$ are expected to be more clean and more informative than the noisy $\tilde{A}$,$\tilde{Y}$. **RGIB-REP assumes that the GNN model can identify these latent variables and further benefit its learning procedure against noise**.
>
> **(4)** **Empirically, we observe that the two instantiations of RGIB can be generalized to different scenarios with their own priority according to the intrinsic graph properties.** RGIB-SSL is more adaptive to sparser graphs, e.g., Cora and Citeseer, where the edge noise results in severer representation collapse. RGIB-REP can be more suitable for denser graphs, e.g., Facebook and Chameleon, where the latent variables of edge data are extracted by RGIB-REP. More importantly, the two RGIB instantiations can be complementary to each other with flexible options in practical applications, and we have summarized such a point in Remark 5.1 of Section 5.1.
>
> **(5)** **Finally, we attempt to jointly combine and use RGIB-REP and RGIB-SSL simultaneously, as you suggest.** Specifically, the "RGIB-REP+RGIB-SSL" integrates both data reparametrization technique and self-supervised learning method, and the final objective for optimization equals minimizing Equation 3 and Equation 4 simultaneously. Then, we compare "RGIB-REP+RGIB-SSL" with standard training, RGIB-REP, and RGIB-SSL based on a 4-layer GCN on all six datasets with $\epsilon=40\\%$ bilateral noise.
>
> **As shown in the below table, the "RGIB-REP+RGIB-SSL" achieves considerable performances as RGIB-REP and RGIB-REP in most cases and outperforms these two instantiations on the Facebook dataset.** Although we believe that a more careful finetuning of the hyper-parameters will bring further improvements to the "RGIB-REP+RGIB-SSL" combination, we suggest using one of the instantiations in practice to keep the learning objective simple and sufficient to combat the bilateral edge noise. The above clarifications and evaluation results will be added to our draft.
>
> | dataset           | Cora  | Citeseer | Pubmed | Facebook | Chameleon | Squirrel |
> | ----------------- | ----- | -------- | ------ | -------- | --------- | -------- |
> | Standard training | .7419 | .7380    | .8748  | .9520    | .9496     | .9406    |
> | RGIB-REP          | .7966 | .7519    | .8834  | .9770    | .9621     | .9455    |
> | RGIB-SSL          | .8554 | .8427    | .8918  | .9711    | .9592     | .9426    |
> | RGIB-REP+RGIB-SSL | .8351 | .8270    | .8880  | .9819    | .9570     | .9431    |

---

### Official Review · Reviewer_Z7rx · 2023-06-30

**Soundness:** 4 excellent
**Presentation:** 3 good
**Contribution:** 3 good
**Rating:** 7
**Confidence:** 4

**Summary:**

This paper proposes to tackle the bilateral edge noise via mutual information. The authors start from empirical observations that existing GNNs are vulnerable to bilateral edge noises. To tackle this issue, the authors propose a robust graph information bottleneck which is information-theory guided. In practice, a self-supervised regularization and a purification mechanism are proposed. The extensive evaluation on various datasets as well as models demonstrates the effectiveness of proposed algorithm.

**Strengths:**

1. The paper is well-written and well-organized.
2. The introduced bilateral edge noise is an interesting and challenging problem in GNNs.
3. The proposed robust graph information bottleneck is well-motivated by both sufficient empirical observations as well as theoretical analysis.
4. The experiments are extensive, including various datasets and GNNs. Various ablation studies demonstrate the effectiveness of RGIB-SSL and RGIB-REP.


**Weaknesses:**

I have several concerns:
1. In RGIB-SSL, the authors introduce hybrid graph augmentation which shows superiority over other contrastive learning methods. However, how the proposed augmentation outperforms other graph contrastive methods needs more clarification to provide some insights of this design.
2. It is recommended to demonstrate the robustness of the proposed algorithm under attacks, such as [4, 8, 46].


**Questions:**

1. Please clarify the motivation of hybrid graph augmentation and the highlight the contribution compared to those in [18, 47].
2. Please include the evaluation under attacks [4, 8, 46] if possible, or discuss the generalization of the proposed algorithm against adversarial attacks.


**Limitations:**

The authors have not discussed the limitations.

---

> ### Author Rebuttal · Authors · 2023-08-09
>
> We thank the reviewer Z7rx for the valuable feedback. We addressed all the comments. Please find the point-to-point responses below. Any further comments and discussions are welcomed!
>
> **Q1**. *In RGIB-SSL, the authors introduce hybrid graph augmentation which shows superiority over other contrastive learning methods. However, how the proposed augmentation outperforms other graph contrastive methods needs more clarification to provide some insights of this design.*
>
> **Reply:** We appreciate the reviewer’s insightful advice. Regarding the three information terms of RGIB-SSL in Equation 3, we elaborate a detailed clarification by comparing it with other contrastive learning methods in the following three folds.
>
> **(1)** supervision term $I(H_1;\tilde{Y}) + I(H_2;\tilde{Y})$.
>
> **Compared with the common manner of self-supervised contrastive learning, RGIB-SSL also considers utilizing noisy labels $\tilde{Y}$.** We show that, although being potentially noisy, the labels $\tilde{Y}$ can also benefit learning. Besides, our experiments show that both the supervised contrastive learning method (e.g., SupCon) and the self-supervised contrastive learning method (e.g., GRACE) perform poorly when learning with the bilateral edge noise. An intuitive explanation is that SupCon entirely trusts and utilizes the noisy $\tilde{Y}$ while GRACE does not adopt $\tilde{Y}$.
>
> **From the data perspective, RGIB-SSL performs a hybrid mode of supervised and self-supervised learning.** It effectively extracts and utilizes the informative signals in $\tilde{Y}$ without entirely absorbing its noisy signals. This is achieved with the synergy of two self-supervision terms as follows.
>
> **(2)** self-supervision alignment term $I(H_1;H_2)$.
>
> **Here, the main differences with common contrastive learning methods are data augmentation and loss function.** Specifically, RGIB-SSL utilizes the hybrid augmentation algorithm and self-adversarial loss. Compared with the fixed augmentation manner of other contrastive learning methods [18, 47], i.e., performs random perturbation on node feature and edges, we propose a hybrid augmentation method with four augmentation operations (please refer to Appendix C for details).
>
> The motivation here is to encourage more diverse views with lower MI $I(A_1; A_2)$ and to avoid manual selection of augmentation operations. Besides, Proposition 4.2 provides a theoretical analysis of hybrid augmentation from information-theoretical perspectives.
>
> Empirically, compared with fixed augmentation, the hybrid brings a 3.0% average AUC promotion on Cora (please refer to Table 5). This means that a stronger augmentation scheme with more diverse views can help better deal with severer edge noise. Besides, the self-adversarial loss further enhances high-quality pairs and decreases low-quality counterparts. It refines the signal and brings up to 2.1% promotion.
>
> **(3)** self-supervision uniformity term $H(H_1) + H(H_2)$.
>
> As for the uniformity term, the understanding part in Section 3.2 shows that a severer edge noise brings a worse uniformity. **To learn a more robust representation, we add this uniformity term to form the final loss of RGIB-SSL and adopt the Gaussian potential kernel for implementation, which is usually not considered in other contrastive learning methods.**
>
> The ablation studies in Section 5.2 also illustrate that the uniformity term is essential, especially in dealing with label noise. Besides, the uniformity of learned representation is also enhanced (see Figure 6), and the various query edges tend to be more uniformly distributed on the unit circle, especially for the negative edges.
>
> In the revision, we will refine and highlight these explanations for the hybrid graph augmentation.
>
> **Q2**. *It is recommended to demonstrate the robustness of the proposed algorithm under attacks, such as [4, 8, 46].*
>
> **Reply:** Thanks for the nice suggestion. As an adversarial attack, the Nettack [49] is considered in all the three works [4, 8, 46] you mentioned. In Appendix D.4 of our work, we conduct experiments based on Nettack that perturbs the graph structure, which is the same as you suggested.
>
> As the Test AUC shown in the tables below, the adversarial attack that adds noisy edges to the input graph also significantly degenerates the GNN's performance. **Crucially, it is observed that RGIB-SSL and RGIB-REP can also promote the robustness of GNN against adversarial attacks on graph structure.** RGIB-SSL and RGIB-REP respectively achieve $4.1\\%$ and $5.5\\%$ improvements of Test AUC on the Cora dataset with $\epsilon_{adv}=20\\%$ adversarial perturbations, demonstrating the robustness of RGIB against adversarial attacks.
>
> Besides, the broader impact of this work and the general robustness of GNNs are also discussed in Appendix. B.4 and B.5.
>
> | Cora dataset (Table 11) | clean     | $\epsilon_{adv}=20\\%$ | $\epsilon_{adv}=40\\%$ | $\epsilon_{adv}=60\\%$ |
> | ----------------------- | --------- | --------------------- | --------------------- | --------------------- |
> | standard training       | .8686     | .7971                 | .7671                 | .7014                 |
> | RGIB-SSL                | **.9260** | .8296                 | **.8095**             | **.8052**             |
> | RGIB-REP                | .8758     | **.8408**             | .7918                 | .7611                 |
>
> | Citeseer dataset (Table 12) | clean     | $\epsilon_{adv}=20\\%$ | $\epsilon_{adv}=40\\%$ | $\epsilon_{adv}=60\\%$ |
> | --------------------------- | --------- | --------------------- | --------------------- | --------------------- |
> | standard training           | .8317     | .8139                 | .7736                 | .7481                 |
> | RGIB-SSL                    | **.9148** | **.8656**             | **.8347**             | **.8022**             |
> | RGIB-REP                    | .8415     | .8382                 | .8107                 | .7893                 |

---

### Official Review · Reviewer_cjox · 2023-07-04

**Soundness:** 3 good
**Presentation:** 3 good
**Contribution:** 3 good
**Rating:** 7
**Confidence:** 3

**Summary:**

This paper focuses on the robustness of GNNs under the edge noise. The authors disclose the influence of bilateral edge noise and the corresponding robustness issue via a series of empirical studies on edge noise. Based on the observations of bilateral noise, the authors propose an information-theory-guided principle, Robust Graph Information Bottleneck (RGIB) and its two instantiations, RGIB-SSL and RGIB-REP. Experimental results verify the effectiveness of RGIB instantiations.

**Strengths:**

- The paper is well written and structured.
- The emprical studies and relative experiments are clear and convincing.
- Experiments are comprehensive and complete.
- The proposed RGIB principle is simple yet effective.

**Weaknesses:**

- There is a lack of more comprehensive studies on other graph representation learning tasks, such as node classification.
- RGIB is proposed under the assumption of edge noise, however the motivation statement is not very convincing. The relationship between GIB and bilateral noise should be further explained.

**Questions:**

- I wonder how RGIB improves other graph representation learning tasks, like node classification?
- Why the model performance on original graphs are not reported? Since the graphs without additional noises are not considered as completely clean, RGIB should be able to provide performance gain as well.

**Limitations:**

Please refer to the weaknesses.

---

> ### Author Rebuttal · Authors · 2023-08-09
>
> We thank the reviewer cjox for the valuable feedback. We addressed all the comments. Please find the point-to-point responses below. Any further comments and discussions are welcomed!
>
> **Q1**. *There is a lack of more comprehensive studies on other graph representation learning tasks, such as node classification. I wonder how RGIB improves other graph representation learning tasks, like node classification?*
>
> **Reply:** The answer is yes. Actually, we have conducted experiments on the node classification task with random label noise on nodes, which is the same as you suggested. Please refer to Appendix D.6 for details. **As shown in below Tables 15 and 16, we justify that the RGIB framework can generalize to the node classification tasks with label noise on nodes, where the two instantiations of RGIB also significantly outperform the standard training manner.**
>
> Besides, please refer to the broader impact and the general robustness of GNNs discussed in the Appendix. B.4 and B.5, respectively. These contents are also relevant to your question.
>
> | Cora (Table 15)   | clean    | $\epsilon=20\\%$ | $\epsilon=40\\%$ | $\epsilon=60\\%$ |
> | ----------------- | -------- | --------------- | --------------- | --------------- |
> | standard training | **.898** | .868            | .720            | .322            |
> | RGIB-SSL          | .900     | **.876**        | **.786**        | **.388**        |
> | RGIB-REP          | .894     | .862            | .760            | .312            |
>
> | Citeseer (Table 16) | clean    | $\epsilon=20\\%$ | $\epsilon=40\\%$ | $\epsilon=60\\%$ |
> | ------------------- | -------- | --------------- | --------------- | --------------- |
> | standard training   | .776     | .746            | .608            | .278            |
> | RGIB-SSL            | **.784** | **.770**        | .646            | .324            |
> | RGIB-REP            | .776     | .754            | **.654**        | **.364**        |
>
> **Q2**. *RGIB is proposed under the assumption of edge noise, however the motivation statement is not very convincing. The relationship between GIB and bilateral noise should be further explained.*
>
> **Reply:** We would like to further explain the motivation of RGIB and the relationship between GIB and bilateral noise in the following three folds.
>
> **(1)** **Conceptually, we would clarify that GIB (Equation 1) is intrinsically susceptive to label noise since it entirely preserves the label supervision with maximizing $I(H;\tilde{Y})$.** As illustrated in Figure 2, the GIB decreases $I(H;\tilde{A}|\tilde{Y})$ by directly constraining $I(H;\tilde{A})$ to handle the input noise. Symmetrically, the label noise can be hidden in the area of $I(H; \tilde{Y} | \tilde{A})$, but trivially constraining $I(H; \tilde{Y})$ to regularize $I(H;\tilde{Y} | \tilde{A})$ is not ideal, since it will conflict with Equation 1. Besides, it cannot tackle the noise within $I(\tilde{A}; \tilde{Y})$, where the two kinds of noise can share similar patterns as the random split manner does not change their distributions in expectation. Thus, GIB cannot provide an ideal solution to the bilateral edge noise investigated in this work.
>
> **(2)** **Thus, it is crucial to further decouple the mutual dependence among $\tilde{A}$, $\tilde{Y}$, and $H$.** Based on the detailed analysis elaborated in Section 4.1 and Appendix B.2, we derive the RGIB principle that balances the three important information terms $H({H})$, $I( H; \tilde{Y} | \tilde{A})$ and $I( H; \tilde{A} | \tilde{Y})$. It works as an information bottleneck to filter out the noisy signals in both $\tilde{A}$ and $\tilde{Y}$, utilizing the supervision signals $I( H; \tilde{Y})$ at the same time. Analytically, GIB only indirectly regularizes the MI term $I( H;\tilde{A}|\tilde{Y})$, as we introduce section 4.1, which can only solve partial noisy information. By contrast, RGIB takes all the related MI terms, i.e., $H({H})$, $I( H; \tilde{Y} | \tilde{A})$, and $I( H; \tilde{A} | \tilde{Y})$, based on which to provide a solution that balances these MI terms as a more strict information bottleneck.
>
> **(3)** **We provide two instantiations for implementing the RGIB principle, i.e., RGIB-SSL and RGIB-REP.** These two instantiations benefit from different methodologies, i.e., self-supervised learning and data reparametrization, for implicit and explicit data denoising, respectively. Note that these two methodologies are not considered in the original GIB. Besides, the GIB is highly coupled with the GAT. By contrast, RGIB does not require any modifications to GNN architecture. It can be seamlessly integrated with various GNNs and promote their robustness against bilateral noise.
>
> In a nutshell, RGIB generalizes the GIB with improvements in both theories and methodologies to learn a robust representation more resistant to the bilateral edge noise. We will follow the reviewer's advice by refining the corresponding description to make it clearer.
>
> **Q3**. *Why the model performance on original graphs are not reported? Since the graphs without additional noises are not considered as completely clean, RGIB should be able to provide performance gain as well.*
>
> **Reply:** We have conducted this supplement experiment that evaluates all the baseline methods on clean datasets in Appendix E.1. As the below Table selected from Table 17, **the proposed two instantiations of RGIB can also boost the predicting performance when learning on clean graphs and significantly outperform other baselines in most cases.**
>
> | Table 17          | Cora      | Citeseer  | Pubmed    | Facebook  | Chameleon | Squirrel  |
> | ----------------- | --------- | --------- | --------- | --------- | --------- | --------- |
> | standard training | .8686     | .8317     | .9178     | .9870     | .9788     | .**9725** |
> | RGIB-SSL          | .8758     | .8415     | .9408     | **.9875** | **.9792** | .9680     |
> | RGIB-REP          | **.9260** | **.9148** | **.9593** | .9845     | .9740     | .9646     |

---

> > ### Comment · Reviewer_cjox · 2023-08-16
> >
> > Thank you for the detailed clarifications and experimental results. I have no further comment.

---

> > > ### Author Response · Authors · 2023-08-17
> > > **Many thanks for your positive support and constructive comments!**
> > >
> > > Hi Reviewer cjox,
> > >
> > > Thank you so much for your comments and appreciation! We really value your constructive feedback, as it helps us improve our work. We will carefully incorporate the discussions and experiments into our submission.
> > >
> > > Please feel free to interact with us if you have any further questions.

---

### Official Review · Reviewer_4Q8x · 2023-07-05

**Soundness:** 3 good
**Presentation:** 3 good
**Contribution:** 3 good
**Rating:** 6
**Confidence:** 3

**Summary:**

The authors extend the Graph Information Bottleneck (GIB) to "bilateral" structural noise and label noise. That is, both the adjacency matrix and the labels are being randomly perturbed. The authors observe that the bilateral noise leads to "poorer alignment and a worse uniformity". To handle this noise, the authors decompose the term used in GIB and, due to its intractability, propose two efficient, practical instantiations.

**Strengths:**

1. The authors propose two methods that significantly and consistently outperform the baselines
1. The approach is principled due to its roots in the powerful concept of information bottleneck
1. The experiments are extensive and are convincing that the method has some merit for certain applications

The paper is generally well-written and logically structured.

**Weaknesses:**

1. The method seems to rely heavily on the assumption that the node features are clean. It would be good to study how sensitive the model is to feature noise.
1. The authors could elaborate more on the assumptions etc., that make their instantiations tractable and when these assumptions are met.

Minor: margins between lines 215 & 216 seem violated.

**Questions:**

1. Would it make sense / be possible also to evaluate the method using an attack that optimizes for $\tilde{A}$ or $\tilde{Y}$, e.g., using first-order optimization?
1. As you do not provide code, I wonder if the clean labels/adjacency are used in any way during training?

**Limitations:**

The authors should elaborate more and more prominently on what the computational requirements of their method are and how it compares to the other baselines.

---

> ### Author Rebuttal · Authors · 2023-08-09
>
> We thank the reviewer 4Q8x for the valuable feedback. We addressed all the comments. Please find the point-to-point responses below. Any further comments and discussions are welcomed!
>
> **Q1**. *The method seems to rely heavily on the assumption that the node features are clean. It would be good to study how sensitive the model is to feature noise.*
>
> **Reply:** We appreciate the reviewer's question about the node feature. **Although our RGIB methods focus on edge noise, we conduct the following experiments to verify the point.** Specifically, we compare standard training, RGIB-REP, and RGIB-SSL using a 4-layer GCN on six datasets with various ratios of feature noise. Here, the noisy feature is generated by adding random noise within the range of $[0,1]$ to the normalized node feature.
>
> The evaluation results of mean AUC are reported in the additional one-page PDF file of our general response.
>
> As can be seen, the noisy node feature also significantly degenerates the GNN's performance, and the degradation becomes severer as the noise ratio increases. **Interestingly, compared with the standard training manner, RGIB-SSL and RGIB-REP can also promote the robustness of GNN against feature noise.** For example, when learning with $\epsilon_f=10\\%$ feature noise, RGIB-SSL can bring $16.6\\%$ and $6.5\\%$ improvements in AUC on Cora and Citeseer datasets, respectively.
>
> We speculate that as the graph representation $H$ is encoded from the node feature $X$, regularizing $H$ in RGIB objectives can also balance its dependence on $X$ and thus shows some potential robustness, even though we originally designed these objectives to handle edge noise.
>
> **The above experiments show that our RGIB principle also has some merits in combating feature noise.** We should note that this might be an initial verification, and a more comprehensive study will be conducted to have a rigorous conclusion in future explorations. We will add the above discussions and evaluation results to the submission.
>
> **Q2**. *The authors could elaborate more on the assumptions etc., that make their instantiations tractable and when these assumptions are met.*
>
> **Reply:** Thanks for the constructive advice. The primary assumption of our work is that the edges of collected graph data can be potentially noisy. As in the submission, we present two ways of realizing the RGIB principle, which actually corresponds to some assumptions in building the objectives with the proper approximation.
>
> **RGIB-SSL assumes that the learned representation can be improved with higher uniformity and alignment.** As RGIB-SSL directly acts on the graph representation, it is more suitable for recovering the distribution of representation, especially when encountering representation collapse due to the severe edge noise.
>
> RGIB-REP explicitly purifies the input graph’s topology and target labels with the jointly reparameterized $Z_A$ and $Z_Y$. Here, the latent variables $Z_A$,$Z_Y$ are expected to be more clean and informative than the noisy $\tilde{A}$,$\tilde{Y}$. **RGIB-REP assumes that the GNN model can identify these latent variables and further benefit its learning procedure against noise.**
>
> Empirically, RGIB-SSL is more adaptive to sparser graphs, e.g., Cora and Citeseer, where the edge noise results in severer representation collapse. RGIB-REP can be more suitable for denser graphs, e.g., Facebook and Chameleon, where the latent variables of edge data are extracted by RGIB-REP. More importantly, the two RGIB instantiations be complementary to each other with flexible options in practical applications.
>
> **Q3**. *Margins between lines 215 & 216 seem violated.*
>
> **Reply:** Thanks for this comment. The small margin is due to the automatic typesetting of the LaTex compiler, and we did not manually modify this margin in our draft. We will rearrange this part to be clearer.
>
> **Q4**. *Would it make sense / be possible also to evaluate the method using an attack that optimizes for $\tilde{A}$ or $\tilde{Y}$, e.g., using first-order optimization?*
>
> **Reply:** Yes, it is reasonable. In Appendix D.4, we conduct the adversarial attacks on $\tilde{A}$, which is the same as you suggested. As shown in Tables 11 and 12, the adversarial attack that adds noisy edges to the input graph also significantly degenerates the GNN's performance. **Importantly, it is observed that the two instantiations of RGIB can also promote the robustness of GNN against adversarial attacks on graph structure.** RGIB-SSL and RGIB-REP achieve $4.1\\%$ and $5.5\\%$ improvements in Test AUC on Cora dataset with $\epsilon_{adv}=20\\%$ adversarial perturbations.
>
> **Q5**. *As you do not provide code, I wonder if the clean labels/adjacency are used in any way during training?*
>
> **Reply:** We would like to kindly point out that we have provided an anonymous link to our source code. Please refer to line 732 in Appendix C.2.
>
> In addition, any clean labels and adjacency are not used in training. On the contrary, the model directly learns from the noisy adjacency $\tilde{A}$ and label $\tilde{Y}$, which is practical as the collected data is potentially noisy in real-world applications.
>
> **Q6**. *The authors should elaborate more and more prominently on what the computational requirements of their method are and how it compares to the other baselines.*
>
> **Reply:** Thanks for the advice. We provide a detailed explanation in the following two folds.
>
> **(1) Noise information.** RGIB and all the baselines are run without any noise priors, e.g., noise type or noise ratio. The only required information for training is adjacency $\tilde{A}$, node feature $X$, and edge labels $\tilde{Y}$, not including any additional heuristics or assumptions.
>
> **(2) Training time.** Further, we evaluate the effectiveness and efficiency of the proposed methods on two large-scale datasets with bilateral noise. As shown in Table 14 of Appendix D.5, the extra computing costs of RGIB are within an acceptable range.

---

> > ### Comment · Reviewer_4Q8x · 2023-08-17
> > **Thank you for the rebuttal**
> >
> > All my comments have been addressed.
> >
> > I think it would help the presentation if the adversarial attack experiments from Appendix D.4 were moved to the main part in a revision of the submitted paper.

---

> > > ### Author Response · Authors · 2023-08-17
> > >
> > > We sincerely appreciate your confirmation and the further suggestion. We will move the experiments about adversarial attack in Appendix D.4 to the main part in the revision, and all other advices will be considered and followed to improve the corresponding parts.
> > >
> > > Best,
> > >
> > > The author of submission4728

---

### Official Review · Reviewer_Fboz · 2023-07-07

**Soundness:** 4 excellent
**Presentation:** 3 good
**Contribution:** 4 excellent
**Rating:** 7
**Confidence:** 3

**Summary:**

This paper tackles the challenge of link prediction on graphs in the presence of edge noise, a topic that has seen little exploration despite the advancements in graph neural networks (GNNs). Through an empirical study, the authors reveal that edge noise can adversely affect both input topology and target labels, leading to performance degradation and representation collapse. In response, the paper introduces an information-theory-guided principle named Robust Graph Information Bottleneck (RGIB), which aims to extract reliable supervision signals and prevent representation collapse. RGIB achieves this by decoupling and balancing the mutual dependencies among graph topology, target labels, and representation, which creates new learning objectives for building robust representations in the face of bilateral noise. The authors present two specific implementations of RGIB, namely RGIB-SSL (which employs self-supervised learning) and RGIB-REP (which uses data reparametrization), for implicit and explicit data denoising respectively. The effectiveness of the proposed RGIB methods is validated through extensive experiments on six datasets and three GNNs under various noisy conditions.

**Strengths:**

The paper is well-written with clear motivation and structure. Overall interesting problem; good mathematical exposition; solid theoretical results and analysis. This paper has a very comprehensive experimental analysis including the experiments in the appendix.

**Weaknesses:**

This paper is overall good.

**Questions:**

Most of my concerns are addressed in the appendix. I have no further questions or suggestions for this work.

**Limitations:**

 Board impact is discussed in the appendix.

---

> ### Author Rebuttal · Authors · 2023-08-09
>
> We thank Reviewer Fboz for the valuable feedback and the positive support of our work.
>
> Any further comments and discussions are welcomed!

---

### Author Rebuttal · Authors · 2023-08-09

### A General Response by Authors

**We would like to thank all the reviewers for their valuable comments on our work.**

**We have received five reviews with positive ratings 7,6,7,7,7. We appreciate that all the reviewers have good impressions on our work**, including **(1)** interesting problem and powerful solution (Fboz, 4Q8x, cjox, Z7rx, 4XLr); **(2)** comprehensive and convincing experiments (Fboz, 4Q8x, cjox, Z7rx, 4XLr); **(3)** sufficient theoretical supports (Fboz, Z7rx, 4XLr); and **(4)** well-written and good presentation (Fboz, 4Q8x, cjox, Z7rx).

**In the rebuttal period, we have provided detailed responses to all the comments and questions point-by-point.** Specifically, we further clarify the assumption (Q2 for 4Q8x), motivation (Q2 for cjox), method (Q1 for Z7rx; Q1 for 4XLr), extension scenarios (Q4 for 4Q8x; Q1,Q3 for cjox; Q2 for Z7rx) and training details (Q5,Q6 for 4Q8x) of our work. Besides, we add new empirical evaluations with the extension on feature noise (Q1 for 4Q8x) and the integration of the two instantiations of RGIB (Q1 for 4XLr). The attached one-page PDF file contains the evaluations on feature noise (Q1 for 4Q8x).

Lastly, we would appreciate all reviewers’ time again. Would you mind checking our response and confirming whether you have any further questions? **We are anticipating your post-rebuttal feedback!**

---

### Decision · Program_Chairs · 2023-09-21

**Decision:**

Accept (poster)

**Comment:**

This paper proposes a novel principle called Robust Graph Information Bottleneck (RGIB) to extract reliable supervision signals and avoid representation collapse in graph neural networks under edge noise. The authors also explore two instantiations of RGIB, namely RGIB-SSL, and RGIB-REP, which leverage different methodologies for data denoising. The work provides empirical evidence to support the effectiveness of RGIB in enhancing the uniformity of learned representation and improving the alignment and uniformity of edge representation. Additionally, the authors propose several informative terms and selection mechanisms to further enhance the performance of RGIB.

Initially, reviewers raised several concerns regarding concept clarity, assumption validity, robustness evaluation, and other task evaluation (such as node classification). Those were well addressed by authors during the rebuttal, leading the unanimously positive reviews post-rebuttal: (7,7,7,7,6). AC sides with the reviewer consensus and considers this paper a clear accept case.